# Benefits of over-parameterization with EM

**Ji Xu**
Columbia University
jixu@cs.columbia.edu

**Daniel Hsu**
Columbia University
djhsu@cs.columbia.edu

**Arian Maleki**
Columbia University
arian@stat.columbia.edu

## Abstract

Expectation Maximization (EM) is among the most popular algorithms for maximum likelihood estimation, but it is generally only guaranteed to find its stationary points of the log-likelihood objective. The goal of this article is to present theoretical and empirical evidence that over-parameterization can help EM avoid spurious local optima in the log-likelihood. We consider the problem of estimating the mean vectors of a Gaussian mixture model in a scenario where the mixing weights are known. Our study shows that the global behavior of EM, when one uses an over-parameterized model in which the mixing weights are treated as unknown, is better than that when one uses the (correct) model with the mixing weights fixed to the known values. For symmetric Gaussians mixtures with two components, we prove that introducing the (statistically redundant) weight parameters enables EM to find the global maximizer of the log-likelihood starting from almost any initial mean parameters, whereas EM without this over-parameterization may very often fail. For other Gaussian mixtures, we provide empirical evidence that shows similar behavior. Our results corroborate the value of over-parameterization in solving non-convex optimization problems, previously observed in other domains.

## 1 Introduction

In a Gaussian mixture model (GMM), the observed data $\mathcal{Y} = \{\boldsymbol{y}_1, \boldsymbol{y}_2, \ldots, \boldsymbol{y}_n\} \subset \mathbb{R}^d$ comprise an i.i.d. sample from a mixture of $k$ Gaussians:

$$\boldsymbol{y}_1, \ldots, \boldsymbol{y}_n \overset{\text{i.i.d.}}{\sim} \sum_{i=1}^{k} w_i^* \, \mathcal{N}(\boldsymbol{\theta}_i^*, \boldsymbol{\Sigma}_i^*) \tag{1}$$

where $(w_i^*, \boldsymbol{\theta}_i^*, \boldsymbol{\Sigma}_i^*)$ denote the weight, mean, and covariance matrix of the $i^{\text{th}}$ mixture component. Parameters of the GMM are often estimated using the Expectation Maximization (EM) algorithm, which aims to find the maximizer of the log-likelihood objective. However, the log-likelihood function is not concave, so EM is only guaranteed to find its stationary points. This leads to the following natural and fundamental question in the study of EM and non-convex optimization: How can EM escape spurious local maxima and saddle points to reach the maximum likelihood estimate (MLE)? In this work, we give theoretical and empirical evidence that over-parameterizing the mixture model can help EM achieve this objective.

Our evidence is based on models in (1) where the mixture components share a known, common covariance, i.e., we fix $\boldsymbol{\Sigma}_i^* = \boldsymbol{\Sigma}^*$ for all $i$. First, we assume that the mixing weights $w_i$ are also fixed to known values. Under this model, which we call *Model 1*, EM finds a stationary point of the log-likelihood function in the parameter space of component means $(\boldsymbol{\theta}_1, \ldots, \boldsymbol{\theta}_k)$. Next, we over-parameterize Model 1 as follows. Despite the fact that the weights fixed in Model 1, we now pretend that they are not fixed. This gives a second model, which we call *Model 2*. Parameter estimation for Model 2 requires EM to estimate the mixing weights in addition to the component means. Finding the global maximizer of the log-likelihood over this enlarged parameter space is

seemingly more difficult for Model 2 than it is for Model 1, and perhaps needlessly so. However, in this paper we present theoretical and empirical evidence to the contrary.

1. For mixtures of two symmetric Gaussians (i.e., $k = 2$ and $\boldsymbol{\theta}_1^* = -\boldsymbol{\theta}_2^*$), we prove that EM for Model 2 converges to the global maximizer of the log-likelihood objective with almost any initialization of the mean parameters, while EM for Model 1 will fail to do so for many choices of $(w_1^*, w_2^*)$. These results are established for idealized executions of EM in an infinite sample size limit, which we complement with finite sample results.

2. We prove that the spurious local maxima in the (population) log-likelihood objective for Model 1 are eliminated in the objective for Model 2.

3. We present an empirical study to show that for more general mixtures of Gaussians, with a variety of model parameters and sample sizes, EM for Model 2 has higher probability to find the MLE than Model 1 under random initializations.

**Related work.** Since Dempster's 1977 paper [Dempster et al., 1977], the EM algorithm has become one of the most popular algorithms to find the MLE for mixture models. Due to its popularity, the convergence analysis of EM has attracted researchers' attention for years. Local convergence of EM has been shown by Wu [1983], Xu and Jordan [1996], Tseng [2004], Chrétien and Hero [2008]. Further, for certain models and under various assumptions about the initialization, EM has been shown to converge to the MLE [Redner and Walker, 1984, Balakrishnan et al., 2017, Klusowski and Brinda, 2016, Yan et al., 2017]. Typically, the initialization is required to be sufficiently close to the true parameter values of the data-generating distribution. Much less is known about global convergence of EM, as the landscape of the log-likelihood function has not been well-studied. For GMMs, Xu et al. [2016] and [Daskalakis et al., 2017] study mixtures of two Gaussians with equal weights and show that the log-likelihood objective has only two global maxima and one saddle point; and if EM is randomly initialized in a natural way, the probability that EM converges to this saddle point is zero. (Our Theorem 2 generalizes these results.) It is known that for mixtures of three or more Gaussians, global convergence is not generally possible [Jin et al., 2016].

The value of over-parameterization for local or greedy search algorithms that aim to find a global minimizer of non-convex objectives has been rigorously established in other domains. Matrix completion is a concrete example: the goal is to recover of a rank $r \ll n$ matrix $M \in \mathbb{R}^{n \times n}$ from observations of randomly chosen entries [Candès and Recht, 2009]. A direct approach to this problem is to find the matrix $X \in \mathbb{R}^{n \times n}$ of minimum rank that is consistent with the observed entries of $M$. However, this optimization problem is NP-hard in general, despite the fact that there are only $2nr - r^2 \ll n^2$ degrees-of-freedom. An indirect approach to this matrix completion problem is to find a matrix $X$ of smallest nuclear norm, subject to the same constraints; this is a convex relaxation of the rank minimization problem. By considering all $n^2$ degrees-of-freedom, Candès and Tao [2010] show that the matrix $M$ is exactly recovered via nuclear norm minimization as soon as $\Omega(nr \log^6 n)$ entries are observed (with high probability). Notably, this combination of over-parameterization with convex relaxation works well in many other research problems such as sparse-PCA [d'Aspremont et al., 2005] and compressive sensing [Donoho, 2006]. However, many problems (like ours) do not have a straightforward convex relaxation. Therefore, it is important to understand how over-parameterization can help one solve a non-convex problem other than convex relaxation.

Another line of work in which the value of over-parameterization is observed is in deep learning. It is conjectured that the use of over-parameterization is the main reason for the success of local search algorithms in learning good parameters for neural nets [Livni et al., 2014, Safran and Shamir, 2017]. Recently, Haeffele and Vidal [2015], Nguyen and Hein [2017, 2018], Soltani and Hegde [2018], Du and Lee [2018] confirm this observation for many neural networks such as feedforward and convolutional neural networks.

## 2   Theoretical results

In this section, we present our main theoretical results concerning EM and two-component Gaussian mixture models.

## 2.1 Sample-based EM and Population EM

Without loss of generality, we assume $\boldsymbol{\Sigma}^* = \boldsymbol{I}$. We consider the following Gaussian mixture model:

$$\boldsymbol{y}_1, \ldots, \boldsymbol{y}_n \overset{\text{i.i.d.}}{\sim} w_1^* \mathcal{N}(\boldsymbol{\theta}^*, \boldsymbol{I}) + w_2^* \mathcal{N}(-\boldsymbol{\theta}^*, \boldsymbol{I}). \tag{2}$$

The mixing weights $w_1^*$ and $w_2^*$ are *fixed* (i.e., assumed to be known). Without loss of generality, we also assume that $w_1^* \geq w_2^* > 0$ (and, of course, $w_1^* + w_2^* = 1$). The only parameter to estimate is the mean vector $\boldsymbol{\theta}^*$. The EM algorithm for this model uses the following iterations:

$$\hat{\boldsymbol{\theta}}^{\langle t+1 \rangle} = \frac{1}{n} \sum_{i=1}^n \left[ \frac{w_1^* e^{\langle \boldsymbol{y}_i, \hat{\boldsymbol{\theta}}^{\langle t \rangle} \rangle} - w_2^* e^{-\langle \boldsymbol{y}_i, \hat{\boldsymbol{\theta}}^{\langle t \rangle} \rangle}}{w_1^* e^{\langle \boldsymbol{y}_i, \hat{\boldsymbol{\theta}}^{\langle t \rangle} \rangle} + w_2^* e^{-\langle \boldsymbol{y}_i, \hat{\boldsymbol{\theta}}^{\langle t \rangle} \rangle}} \boldsymbol{y}_i \right]. \tag{3}$$

We refer to this algorithm as Sample-based $EM_1$: it is the EM algorithm one would normally use when the mixing weights are known. In spite of this, we also consider an EM algorithm that pretends that the weights are not known, and estimates them alongside the mean parameters. We refer to this algorithm as Sample-based $EM_2$, which uses the following iterations:

$$\hat{w}_1^{\langle t+1 \rangle} = \frac{1}{n} \sum_{i=1}^n \left[ \frac{\hat{w}_1^{\langle t \rangle} e^{\langle \boldsymbol{y}_i, \hat{\boldsymbol{\theta}}^{\langle t \rangle} \rangle}}{\hat{w}_1^{\langle t \rangle} e^{\langle \boldsymbol{y}_i, \hat{\boldsymbol{\theta}}^{\langle t \rangle} \rangle} + \hat{w}_2^{\langle t \rangle} e^{-\langle \boldsymbol{y}_i, \hat{\boldsymbol{\theta}}^{\langle t \rangle} \rangle}} \right] = 1 - \hat{w}_2^{\langle t+1 \rangle}.$$

$$\hat{\boldsymbol{\theta}}^{\langle t+1 \rangle} = \frac{1}{n} \sum_{i=1}^n \left[ \frac{\hat{w}_1^{\langle t \rangle} e^{\langle \boldsymbol{y}_i, \hat{\boldsymbol{\theta}}^{\langle t \rangle} \rangle} - \hat{w}_2^{\langle t \rangle} e^{-\langle \boldsymbol{y}_i, \hat{\boldsymbol{\theta}}^{\langle t \rangle} \rangle}}{\hat{w}_1^{\langle t \rangle} e^{\langle \boldsymbol{y}_i, \hat{\boldsymbol{\theta}}^{\langle t \rangle} \rangle} + \hat{w}_2^{\langle t \rangle} e^{-\langle \boldsymbol{y}_i, \hat{\boldsymbol{\theta}}^{\langle t \rangle} \rangle}} \boldsymbol{y}_i \right]. \tag{4}$$

This is the EM algorithm for a different Gaussian mixture model in which the weights $w_1^*$ and $w_2^*$ are not fixed (i.e., unknown), and hence must be estimated. Our goal is to study the global convergence properties of the above two EM algorithms on data from the *first* model, where the mixing weights are, in fact, known.

We study idealized executions of the EM algorithms in the large sample limit, where the algorithms are modified to be computed over an infinitely large i.i.d. sample drawn from the mixture distribution in (2). Specifically, we replace the empirical averages in (3) and (4) with the expectations with respect to the mixture distribution. We obtain the following two modified EM algorithms, which we refer to as Population $EM_1$ and Population $EM_2$:

- Population $EM_1$:

$$\boldsymbol{\theta}^{\langle t+1 \rangle} = \mathbb{E}_{\boldsymbol{y} \sim f^*} \left[ \frac{w_1^* e^{\langle \boldsymbol{y}, \boldsymbol{\theta}^{\langle t \rangle} \rangle} - w_2^* e^{-\langle \boldsymbol{y}, \boldsymbol{\theta}^{\langle t \rangle} \rangle}}{w_1^* e^{\langle \boldsymbol{y}, \boldsymbol{\theta}^{\langle t \rangle} \rangle} + w_2^* e^{-\langle \boldsymbol{y}, \boldsymbol{\theta}^{\langle t \rangle} \rangle}} \boldsymbol{y} \right] =: H(\boldsymbol{\theta}^{\langle t \rangle}; \boldsymbol{\theta}^*, w_1^*), \tag{5}$$

  where $f^* = f^*(\boldsymbol{\theta}^*, w_1^*)$ here denotes the true distribution of $\boldsymbol{y}_i$ given in (2).

- Population $EM_2$: Set $w_1^{\langle 0 \rangle} = w_2^{\langle 0 \rangle} = 0.5$[1], and run

$$w_1^{\langle t+1 \rangle} = \mathbb{E}_{\boldsymbol{y} \sim f^*} \left[ \frac{w_1^{\langle t \rangle} e^{\langle \boldsymbol{y}, \boldsymbol{\theta}^{\langle t \rangle} \rangle}}{w_1^{\langle t \rangle} e^{\langle \boldsymbol{y}, \boldsymbol{\theta}^{\langle t \rangle} \rangle} + w_2^{\langle t \rangle} e^{-\langle \boldsymbol{y}, \boldsymbol{\theta}^{\langle t \rangle} \rangle}} \right] =: G_w(\boldsymbol{\theta}^{\langle t \rangle}, w^{\langle t \rangle}; \boldsymbol{\theta}^*, w_1^*) \tag{6}$$

$$= 1 - w_2^{\langle t+1 \rangle}.$$

$$\boldsymbol{\theta}^{\langle t+1 \rangle} = \mathbb{E}_{\boldsymbol{y} \sim f^*} \left[ \frac{w_1^{\langle t \rangle} e^{\langle \boldsymbol{y}, \boldsymbol{\theta}^{\langle t \rangle} \rangle} - w_2^{\langle t \rangle} e^{-\langle \boldsymbol{y}, \boldsymbol{\theta}^{\langle t \rangle} \rangle}}{w_1^{\langle t \rangle} e^{\langle \boldsymbol{y}, \boldsymbol{\theta}^{\langle t \rangle} \rangle} + w_2^{\langle t \rangle} e^{-\langle \boldsymbol{y}, \boldsymbol{\theta}^{\langle t \rangle} \rangle}} \boldsymbol{y} \right] =: G_\theta(\boldsymbol{\theta}^{\langle t \rangle}, w^{\langle t \rangle}; \boldsymbol{\theta}^*, w_1^*). \tag{7}$$

As $n \rightarrow \infty$, we can show the performance of Sample-based $EM_\star$ converges to that of the Population $EM_\star$ in probability. This argument has been used rigorously in many previous works on EM [Balakrishnan et al., 2017, Xu et al., 2016, Klusowski and Brinda, 2016, Daskalakis et al., 2017]. The main goal of this section, however, is to study the dynamics of Population $EM_1$ and Population $EM_2$, and the landscape of the log-likelihood objectives of the two models.

## 2.2 Main theoretical results

Let us first consider the special case $w_1^* = w_2^* = 0.5$. Then, it is straightforward to show that $w_1^{\langle t \rangle} = w_2^{\langle t \rangle} = 0.5$ for all $t$ in Population $\text{EM}_2$. Hence, Population $\text{EM}_2$ is equivalent to Population $\text{EM}_1$. Global convergence of $\boldsymbol{\theta}^{\langle t \rangle}$ to $\boldsymbol{\theta}^*$ for this case was recently established by Xu et al. [2016, Theorem 1] for almost all initial $\boldsymbol{\theta}^{\langle 0 \rangle}$ (see also [Daskalakis et al., 2017]).

We first show that the same global convergence may *not* hold for Population $\text{EM}_1$ when $w_1^* \neq w_2^*$.

**Theorem 1.** *Consider Population $\text{EM}_1$ in dimension one (i.e., $\theta^* \in \mathbb{R}$). For any $\theta^* > 0$, there exists $\delta > 0$, such that given $w_1^* \in (0.5, 0.5 + \delta)$ and initialization $\theta^{\langle 0 \rangle} \leq -\theta^*$, the Population $\text{EM}_1$ estimate $\theta^{\langle t \rangle}$ converges to a fixed point $\theta_{\text{wrong}}$ inside $(-\theta^*, 0)$.*

This theorem, which is proved in Appendix A, implies that if we use random initialization, Population $\text{EM}_1$ may converge to the wrong fixed point with constant probability. We illustrate this in Figure 1. The iterates of Population $\text{EM}_1$ converge to a fixed point of the function $\theta \mapsto H(\theta; \theta^*, w_1^*)$ defined in (5). We have plotted this function for several different values of $w_1^*$ in the left panel of Figure 1. When $w_1^*$ is close to 1, $H(\theta; \theta^*, w_1^*)$ has only one fixed point and that is at $\theta = \theta^*$. Hence, in this case, the estimates produced by Population $\text{EM}_1$ converge to the true $\theta^*$. However, when we decrease the value of $w_1^*$ below a certain threshold (which is numerically found to be approximately 0.77 for $\theta^* = 1$), two other fixed points of $H(\theta; \theta^*, w_1^*)$ emerge. These new fixed points are foils for Population $\text{EM}_1$.

From the failure of Population $\text{EM}_1$, one may expect the over-parameterized Population $\text{EM}_2$ to fail as well. Yet, surprisingly, our second theorem proves the opposite is true: Population $\text{EM}_2$ has global convergence even when $w_1^* \neq w_2^*$.

**Theorem 2.** *For any $w_1^* \in [0.5, 1)$, the Population $\text{EM}_2$ estimate $(\boldsymbol{\theta}^t, w^{\langle t \rangle})$ converges to either $(\boldsymbol{\theta}^*, w_1^*)$ or $(-\boldsymbol{\theta}^*, w_2^*)$ with any initialization $\boldsymbol{\theta}^{\langle 0 \rangle}$ except on the hyperplane $\langle \boldsymbol{\theta}^{\langle 0 \rangle}, \boldsymbol{\theta}^* \rangle = 0$. Furthermore, the convergence speed is geometric after some finite number of iterations, i.e., there exists a finite number $T$ and constant $\rho \in (0, 1)$ such that the following hold.*

- *If $\langle \boldsymbol{\theta}^{\langle 0 \rangle}, \boldsymbol{\theta}^* \rangle > 0$, then for all $t > T$,*

$$\|\boldsymbol{\theta}^{\langle t+1 \rangle} - \boldsymbol{\theta}^*\|^2 + |w_1^{\langle t+1 \rangle} - w_1^*|^2 \leq \rho^{t-T} \left( \|\boldsymbol{\theta}^{\langle T \rangle} - \boldsymbol{\theta}^*\|^2 + (w_1^{\langle T \rangle} - w_1^*)^2 \right).$$

- *If $\langle \boldsymbol{\theta}^{\langle 0 \rangle}, \boldsymbol{\theta}^* \rangle < 0$, then for all $t > T$,*

$$\|\boldsymbol{\theta}^{\langle t+1 \rangle} + \boldsymbol{\theta}^*\|^2 + |w_1^{\langle t+1 \rangle} - w_2^*|^2 \leq \rho^{t-T} \left( \|\boldsymbol{\theta}^{\langle T \rangle} + \boldsymbol{\theta}^*\|^2 + (w_1^{\langle T \rangle} - w_2^*)^2 \right).$$

Theorem 2 implies that if we use random initialization for $\boldsymbol{\theta}^{\langle 0 \rangle}$, with probability one, the Population $\text{EM}_2$ estimates converge to the true parameters.

The failure of Population $\text{EM}_1$ and success of Population $\text{EM}_2$ can be explained intuitively. Let $C_1$ and $C_2$, respectively, denote the true mixture components with parameters $(w_1^*, \theta^*)$ and $(w_2^*, -\theta^*)$. Due to the symmetry in Population $\text{EM}_1$, we are assured that among the two estimated mixture components, one will have a positive mean, and the other will have a negative mean: call these $\hat{C}_+$ and $\hat{C}_-$, respectively. Assume $\theta^* > 0$ and $w_1^* > 0.5$, and consider initializing the Population $\text{EM}_1$ with $\theta^{\langle 0 \rangle} := -\theta^*$. This initialization incorrectly associates $\hat{C}_-$ with the larger weight $w_1^*$ instead of the smaller weight $w_2^*$. This causes, in the E-step of EM, the component $\hat{C}_-$ to become "responsible" for an overly large share of the overall probability mass, and in particular an overly large share of the mass from $C_1$ (which has a positive mean). Thus, in the M-step of EM, when the mean of the estimated component $\hat{C}_-$ is updated, it is pulled rightward towards $+\infty$. It is possible that this rightward pull would cause the estimated mean of $\hat{C}_-$ to become positive—in which case the roles of $\hat{C}_+$ and $\hat{C}_-$ would switch—but this will not happen as long as $w_1^*$ is sufficiently bounded away from 1 (but still $> 0.5$).[2] The result is a bias in the estimation of $\theta^*$, thus explaining why the Population $\text{EM}_1$ estimate converges to some $\theta_{\text{wrong}} \in (-\theta^*, 0)$ when $w_1^*$ is not too large.

Our discussion confirms that one way Population $\text{EM}_1$ may fail (in dimension one) is if it is initialized with $\theta^{\langle 0 \rangle}$ having the "incorrect" sign (e.g., $\theta^{\langle 0 \rangle} = -\theta^*$). On the other hand, the performance of Population $\text{EM}_2$ does not depend on the sign of the initial $\theta^{\langle 0 \rangle}$. Recall that the estimates of Population $\text{EM}_2$ converge to the fixed points of the mapping $\mathcal{M} : (\boldsymbol{\theta}, w_1) \mapsto (G_\theta(\boldsymbol{\theta}, w_1; \boldsymbol{\theta}^*, w_1^*), G_w(\boldsymbol{\theta}, w_1; \boldsymbol{\theta}^*, w_1^*))$, as defined in (6) and (7). One can check that for all $\boldsymbol{\theta}, w_1, \boldsymbol{\theta}^*, w_1^*$, we have

$$
\begin{aligned}
G_\theta(\boldsymbol{\theta}, w_1; \boldsymbol{\theta}^*, w_1^*) + G_\theta(-\boldsymbol{\theta}, w_2; \boldsymbol{\theta}^*, w_1^*) &= 0, \\
G_w(\boldsymbol{\theta}, w_1; \boldsymbol{\theta}^*, w_1^*) + G_w(-\boldsymbol{\theta}, w_2; \boldsymbol{\theta}^*, w_1^*) &= 1.
\end{aligned}
\tag{8}
$$

Hence, $(\boldsymbol{\theta}, w_1)$ is a fixed point of $\mathcal{M}$ if and only if $(-\boldsymbol{\theta}, w_2)$ is a fixed point of $\mathcal{M}$ as well. Therefore, Population $\text{EM}_2$ is insensitive to the sign of the initial $\theta^{\langle 0 \rangle}$. This property can be extended to mixtures of $k > 2$ Gaussians as well. In these cases, the performance of EM for Model 2 is insensitive to permutations of the component parameters. Hence, because of this nice property, as we will confirm in our simulations, when the mixture components are well-separated, EM for Model 2 performs well for most of the initializations, while EM for Model 1 fails in many cases.

One limitation of our permutation-free explanation is that the argument only holds when the weights in Population $\text{EM}_2$ are initialized to be uniform. However, the benefits of over-parameterization are not limited to this case. Indeed, when we compare the landscapes of the log-likelihood objective for (the mixture models corresponding to) Population $\text{EM}_1$ and Population $\text{EM}_2$, we find that over-parameterization eliminates spurious local maxima that were obstacles for Population $\text{EM}_1$.

**Theorem 3.** *For all $w_1^* \neq 0.5$, the log-likelihood objective optimized by Population $\text{EM}_2$ has only one saddle point $(\boldsymbol{\theta}, w_1) = (\mathbf{0}, 1/2)$ and no local maximizers besides the two global maximizers $(\boldsymbol{\theta}, w_1) = (\boldsymbol{\theta}^*, w_1^*)$ and $(\boldsymbol{\theta}, w_1) = (-\boldsymbol{\theta}^*, w_2^*)$.*

The proof of this theorem is presented in Appendix C.

**Remark 1.** *Consider the landscape of the log-likelihood objective for Population $\text{EM}_2$ and the point $(\theta_{\text{wrong}}, w_1^*)$, where $\theta_{\text{wrong}}$ is the local maximizer suggested by Theorem 1. Theorem 3 implies that we can still easily escape this point due to the non-zero gradient in the direction of $w_1$ and thus $(\theta_{\text{wrong}}, w_1^*)$ is not even a saddle point. We emphasize that this is exactly the mechanism that we have hoped for the purpose and benefit of over-parameterization (See the left panel in Figure 2).*

**Remark 2.** *Note that although $(\boldsymbol{\theta}, w_1) = ((w_1^* - w_2^*)\boldsymbol{\theta}^*, 1)$ or $((w_2^* - w_1^*)\boldsymbol{\theta}^*, 0)$ are the two fixed points for Population $\text{EM}_2$ as well, they are not the first order stationary points of the log-likelihood objective if $w_1^* \neq 0.5$.*

Finally, to complete the analysis of EM for the mixtures of two Gaussians, we present the following result that applies to Sample-based $\text{EM}_2$.

**Theorem 4.** *Let $(\hat{\boldsymbol{\theta}}^{\langle t \rangle}, \hat{w}_1^{\langle t \rangle})$ be the estimates of Sample-based $\text{EM}_2$. Suppose $\hat{\boldsymbol{\theta}}^{\langle 0 \rangle} = \boldsymbol{\theta}^{\langle 0 \rangle}, \hat{w}_1^{\langle 0 \rangle} = w_1^{\langle 0 \rangle} = \frac{1}{2}$ and $\langle \boldsymbol{\theta}^{\langle 0 \rangle}, \boldsymbol{\theta}^* \rangle \neq 0$. Then we have*

$$
\limsup_{t \to \infty} \|\hat{\boldsymbol{\theta}}^{\langle t \rangle} - \boldsymbol{\theta}^{\langle t \rangle}\| \to 0 \quad \text{and} \quad \limsup_{t \to \infty} |\hat{w}_1^{\langle t \rangle} - w_1^{\langle t \rangle}| \to 0 \quad \text{as } n \to \infty,
$$

*where convergence is in probability.*

The proof of this theorem uses the same approach as Xu et al. [2016] and is presented in Appendix D.

## 2.3  Roadmap of the proof for Theorem 2

Our first lemma, proved in Appendix B.1, confirms that if $\langle \boldsymbol{\theta}^{\langle 0 \rangle}, \boldsymbol{\theta}^* \rangle > 0$, then $\langle \boldsymbol{\theta}^{\langle t \rangle}, \boldsymbol{\theta}^* \rangle > 0$ for every $t$ and $w_1^{\langle t \rangle} \in (0.5, 1)$. In other words, the estimates of the Population $\text{EM}_2$ remain in the correct hyperplane, and the weight moves in the right direction, too.

---

be so strong that the updated mean estimate becomes positive. Since the model enforces that the mean estimates of $\hat{C}_+$ and $\hat{C}_-$ be negations of each other, the roles of $\hat{C}_+$ and $\hat{C}_-$ switch, and now it is $\hat{C}_+$ that becomes associated with the larger mixing weight $w_1^*$. In this case, owing to the symmetry assumption, Population $\text{EM}_1$ may be able to successfully converge to $\theta^*$. We revisit this issue in the numerical study, where the symmetry assumption is removed.

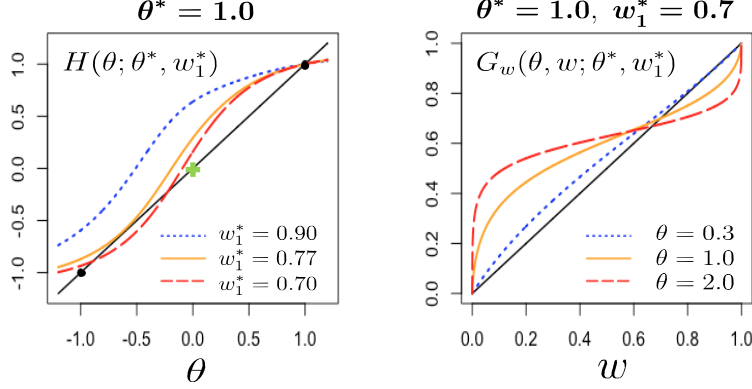

Figure 1: Left panel: we show the shape of iterative function $H(\theta; \theta^*, w_1^*)$ with $\theta^* = 1$ and different values of $w_1^* \in \{0.9, 0.77, 0.7\}$. The green plus $+$ indicates the origin $(0, 0)$ and the black points indicate the correct values $(\theta^*, \theta^*)$ and $(-\theta^*, -\theta^*)$. We observe that as $w_1^*$ increases, the number of fixed points goes down from 3 to 2 and finally to 1. Further, when there exists more than one fixed point, there is one stable incorrect fixed point in $(-\theta^*, 0)$. Right panel: we show the shape of iterative function $G_w(\theta, w_1; \theta^*, w_1^*)$ with $\theta^* = 1, w_1^* = 0.7$ and different values of $\theta \in \{0.3, 1, 2\}$. We observe that as $\theta$ increases, $G_w$ becomes from a concave function to a concave-convex function. Further, there are at most three fixed points and there is only one stable fixed point.

**Lemma 1.** *If $\langle \boldsymbol{\theta}^{\langle 0 \rangle}, \boldsymbol{\theta}^* \rangle > 0$, we have $\langle \boldsymbol{\theta}^{\langle t \rangle}, \boldsymbol{\theta}^* \rangle > 0, w_1^{\langle t \rangle} \in (0.5, 1)$ for all $t \geq 1$. Otherwise, if $\langle \boldsymbol{\theta}^{\langle 0 \rangle}, \boldsymbol{\theta}^* \rangle < 0$, we have $\langle \boldsymbol{\theta}^{\langle t \rangle}, \boldsymbol{\theta}^* \rangle < 0, w_1^{\langle t \rangle} \in (0, 0.5)$ for all $t \geq 1$.*

On account of Lemma 1 and the invariance in (8), we can assume without loss of generality that $\langle \boldsymbol{\theta}^{\langle t \rangle}, \boldsymbol{\theta}^* \rangle > 0$ and $w_1^{\langle t \rangle} \in (0.5, 1)$ for all $t \geq 1$.

Let $d$ be the dimension of $\boldsymbol{\theta}^*$. We reduce the $d > 1$ case to the $d = 1$ case. This achieved by proving that the angle between the two vectors $\boldsymbol{\theta}^{\langle t \rangle}$ and $\boldsymbol{\theta}^*$ is a decreasing function of $t$ and converges to 0. The details appear in Appendix B.4. Hence, in the rest of this section we focus on the proof of Theorem 2 for $d = 1$.

Let $g_\theta(\theta, w_1)$ and $g_w(\theta, w_1)$ be the shorthand for the two update functions $G_\theta$ and $G_w$ defined in (6) and (7) for a fixed $(\theta^*, w_1^*)$. To prove that $\{(\theta^{\langle t \rangle}, w^{\langle t \rangle})\}$ converges to the fixed point $(\theta_\star, w_\star)$, we establish the following claims:

C.1  There exists a set $S = (a_\theta, b_\theta) \times (a_w, b_w) \in \mathbb{R}^2$, where $a_\theta, b_\theta \in \mathbb{R} \cup \{\pm\infty\}$ and $a_w, b_w \in \mathbb{R}$, such that $S$ contains point $(\theta_\star, w_\star)$ and point $(g_\theta(\theta, w_1), g_w(\theta, w_1)) \in S$ for all $(\theta, w_1) \in S$. Further, $g_\theta(\theta, w_1)$ is a non-decreasing function of $\theta$ for a given $w_1 \in (a_w, b_w)$ and $g_w(\theta, w_1)$ is a non-decreasing function of $w$ for a given $\theta \in (a_\theta, b_\theta)$,

C.2  There is a *reference curve* $r \colon [a_w, b_w] \to [a_\theta, b_\theta]$ defined on $\bar{S}$ (the closure of $S$) such that:

&emsp;C.2a  $r$ is continuous, decreasing, and passes through point $(\theta_\star, w_\star)$, i.e., $r(w_\star) = \theta_\star$.

&emsp;C.2b  Given $\theta \in (a_\theta, b_\theta)$, function $w \mapsto g_w(\theta, w)$ has a stable fixed point in $[a_w, b_w]$. Further, any stable fixed point $w_s$ in $[a_w, b_w]$ or fixed point $w_s$ in $(a_w, b_w)$ satisfies the following:

&emsp;&emsp;$*$  If $\theta < \theta_\star$ and $\theta \geq r(b_w)$, then $r^{-1}(\theta) > w_s > w_\star$.

&emsp;&emsp;$*$  If $\theta = \theta_\star$, then $r^{-1}(\theta) = w_s = w_\star$.

&emsp;&emsp;$*$  If $\theta > \theta_\star$ and $\theta \leq r(a_w)$, then $r^{-1}(\theta) < w_s < w_\star$.

&emsp;C.2c  Given $w \in [a_w, b_w]$, function $\theta \mapsto g_\theta(\theta, w)$ has a stable fixed point in $[a_\theta, b_\theta]$. Further, any stable fixed point $\theta_s$ in $[a_\theta, b_\theta]$ or fixed point $\theta_s$ in $(a_\theta, b_\theta)$ satisfies the following:

&emsp;&emsp;$*$  If $w_1 < w_\star$, then $r(w) > \theta_s > \theta_\star$.

&emsp;&emsp;$*$  If $w_1 = w_\star$, then $r(w) = \theta_s = \theta_\star$.

&emsp;&emsp;$*$  If $w_1 > w_\star$, then $r(w) < \theta_s < \theta_\star$.

We explain C.1 and C.2 in the right panel of Figure 2. Heuristically, we expect $(\theta^*, w_1^*)$ to be the only fixed point of the mapping $(\theta, w) \mapsto (g_\theta(\theta, w), g_w(\theta, w))$, and that $(\theta^{\langle t \rangle}, w^{\langle t \rangle})$ move toward this fixed point. Hence, we can prove the convergence of the iterates by showing certain geometric

relationships between the curves of fixed points of the two functions. Hence, C.1 helps us to bound the iterates on the area that such nice geometric relations exist, and the reference curve $r$ and C.2 are the tools to help us mathematically characterizing the geometric relations shown in the figure. Indeed, the next lemma implies that $C.1$ and $C.2$ are sufficient to show the convergence to the right point $(\theta_\star, w_\star)$:

**Lemma 2** (Proved in Appendix B.2.1). *Suppose continuous functions $g_\theta(\theta, w), g_w(\theta, w)$ satisfy $C.1$ and $C.2$, then there exists a continuous mapping $m : \bar{S} \to [0, \infty)$ such that $(\theta_\star, w_\star)$ is the only solution for $m(\theta, w) = 0$ on $\bar{S}$, the closure of $S$ . Further, if we initialize $(\theta^{\langle 0 \rangle}, w^{\langle 0 \rangle})$ in $S$, the sequence $\{(\theta^{\langle t \rangle}, w^{\langle t \rangle})\}_{t \geq 0}$ defined by*

$$\theta^{\langle t+1 \rangle} \;=\; g_\theta(\theta^{\langle t \rangle}, w^{\langle t \rangle}), \quad and \quad w^{\langle t+1 \rangle} \;=\; g_w(\theta^{\langle t \rangle}, w^{\langle t \rangle}),$$

*satisfies that $m(\theta^{\langle t \rangle}, w^{\langle t \rangle}) \downarrow 0$, and therefore $(\theta^{\langle t \rangle}, w^{\langle t \rangle})$ converges to $(\theta_\star, w_\star)$.*

In our problem, we set $a_w = 0.5, b_w = 1, a_\theta = 0, b_\theta = \infty$ and $(\theta_\star, w_\star) = (\theta^*, w_1^*)$. Then according to Lemma 1 and monotonic property of $g_\theta$ and $g_w$, C.1 is satisfied.

To show C.2, we first define the reference curve $r$ by

$$r(w_1) := \frac{w_1^* - w_2^*}{w_1 - w_2} \theta^* \;=\; \frac{2w_1^* - 1}{2w_1 - 1} \theta^*, \qquad \forall w_1 \in (0.5, 1], w_2 = 1 - w_1. \tag{9}$$

The claim C.2a holds by construction. To show C.2b, we establish an even stronger property of the weights update function $g_w(\theta, w)$: for any fixed $\theta > 0$, the function $w_1 \mapsto g_w(\theta, w_1)$ has at most one other fixed point besides $w_1 = 0$ and $w_1 = 1$, and most importantly, it has only one unique stable fixed point. This is formalized in the following lemma.

**Lemma 3** (Proved in Appendix B.2.2). *For all $\theta > 0$, there are at most three fixed points for $g_w(\theta, w_1)$ with respect to $w_1$. Further, there exists an unique stable fixed point $F_w(\theta) \in (0, 1]$, i.e., (i) $F_w(\theta) = g_w(\theta, F_w(\theta))$ and (ii) for all $w_1 \in (0, 1)$, we have*

$$g_w(\theta, w_1) < w_1 \Leftrightarrow w_1 < F_w(\theta) \quad and \quad g_w(\theta, w_1) > w_1 \Leftrightarrow w_1 > F_w(\theta). \tag{10}$$

We explain Lemma 3 in Figure 1. Note that, in the figure, we observe that $g_w$ is an increasing function with $g_w(\theta, 0) = 0$ and $g_w(\theta, 1) = 1$. Further, it is either a concave function, it is piecewise concave-then-convex[3]. Hence, we know if $\partial g_w(\theta, w_1)/\partial w_1|_{w_1=1}$ is at most 1, the only stable fixed point is $w_1 = 1$, else if the derivative is larger than 1, there exists only one fixed point in $(0,1)$ and it is the only stable fixed point. The complete proof for C.2b is shown in Appendix B.3.

The final step to apply Lemma 2 is to prove C2.c. However, $(\theta, w_1) = ((2w_1^* - 1)\theta^*, 1)$ is a point on the reference curve $r$ and $\theta = (2w_1^* - 1)\theta^*$ is a stable fixed point for $g_\theta(\theta, 1)$. This violates C.2c. To address this issue, since we can characterize the shape and the number of fixed points for $g_w$, by typical uniform continuity arguments, we can find $\delta, \epsilon > 0$ such that the adjusted reference curve $r_{adj}(w) := r(w) - \epsilon \cdot \max(0, w - 1 + \delta)$ satisfies C.2a and C.2b. Then we can prove that the adjusted reference curve $r_{adj}(w)$ satisfies C2.c; see Appendix B.3.1.

## 3 Numerical results

In this section, we present numerical results that show the value of over-parameterization in some mixture models not covered by our theoretical results.

### 3.1 Setup

Our goal is to analyze the effect of the sample size, mixing weights, and the number of mixture components on the success of the two EM algorithms described in Section 2.1.

We implement EM for both Model 1 (where the weights are assumed to be known) and Model 2 (where the weights are not known), and run the algorithm multiple times with random initial mean estimates. We compare the two versions of EM by their (empirical) success probabilities, which we denote by $P_1$ and $P_2$, respectively. Success is defined in two ways, depending on whether EM is run with a finite sample, or with an infinite-size sample (i.e., the population analogue of EM).

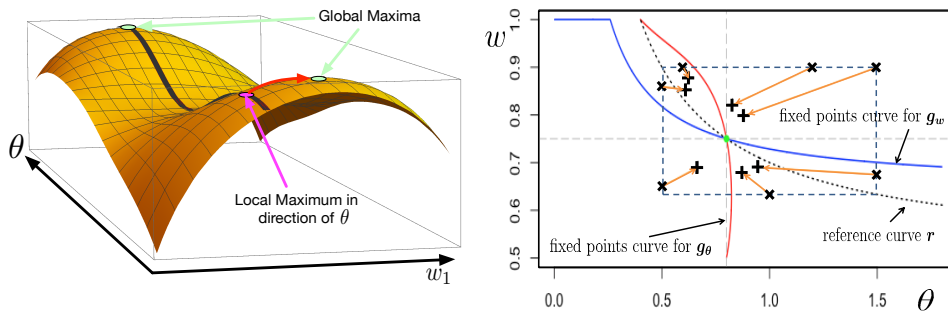

Figure 2: Left panel: The landscapes of log-likelihood objectives for Population $\text{EM}_1$ and Population $\text{EM}_2$ with $(\boldsymbol{\theta}^*, w_1^*) = (1, 0.4)$ are shown in the black belt and the yellow surface respectively. The two green points indicates the two global maxima of Population $\text{EM}_2$, one of which is also the global maximum of Population $\text{EM}_1$. The purple point indicates the local maximum of Population $\text{EM}_1$. Over-parameterization helps us to escape the local maximum through the direction of $w_1$. Right panel: The fixed point curves for functions $g_\theta$ and $g_w$ are shown with red and blue lines respectively. The green point at the intersections of the three curves is the correct convergence point $(\theta_\star, w_\star)$. The black dotted curve shows the reference curve $r$. The cross points $\times$ are the possible initializations and the plus points $+$ are the corresponding positions after the first iteration. By the geometric relations between the three curves, the iterations have to converge to $(\theta_\star, w_\star)$

When EM is run using a finite sample, we do not expect recover the $\boldsymbol{\theta}_i^* \in \mathbb{R}^d$ exactly. Hence, success is declared when the $\boldsymbol{\theta}_i^*$ are recovered up to some expected error, according to the following measure:

$$
\text{error} \;=\; \min_{\pi \in \Pi} \lim_{t \to \infty} \sum_{i=1}^k w_i^* \|\boldsymbol{\theta}_{\pi(i)}^{\langle t \rangle} - \boldsymbol{\theta}_i^*\|^2,
\tag{11}
$$

where $\Pi$ is the set of all possible permutations on $\{1, \dots, k\}$. We declare success if the error is at most $C_\epsilon / n$, where $C_\epsilon := 4 \cdot \text{Tr}(\boldsymbol{W}^* \mathcal{I}^{-1}(\boldsymbol{\Theta}^*))$. Here, $\boldsymbol{W}^*$ is the diagonal matrix whose diagonal is $(w_1^*, \dots, w_1^*, \dots, w_k^*, \dots, w_k^*) \in \mathbb{R}^{kd}$, where each $w_i^*$ is repeated $d$ times, and $\mathcal{I}(\boldsymbol{\Theta}^*)$ is the Fisher Information at the true value $\boldsymbol{\Theta}^* := (\boldsymbol{\theta}_1^*, \dots, \boldsymbol{\theta}_k^*)$. We adopt this criteria since it is well known that the MLE asymptotically converges to $\mathcal{N}(\theta^*, \mathcal{I}^{-1}(\boldsymbol{\Theta}^*)/n)$. Thus, constant $4 \approx 1.96^2$ indicates an approximately $95\%$ coverage.

When EM is run using an infinite-size sample, we declare EM successful when the error defined in (11) is at most $10^{-7}$.

## 3.2 Mixtures of two Gaussians

We first consider mixtures of two Gaussians in one dimension, i.e., $\theta_1^*, \theta_2^* \in \mathbb{R}$. Unlike in our theoretical analysis, the mixture components are not constrained to be symmetric about the origin. For simplicity, we always let $\theta_1^* = 0$, but this information is not used by EM. Further, we consider sample size $n \in \{1000, \infty\}$, separation $\theta_2^* = |\theta_2^* - \theta_1^*| \in \{1, 2, 4\}$, and mixing weight $w_1^* \in \{0.52, 0.7, 0.9\}$; this gives a total of 18 cases. For each case, we run EM with 2500 random initializations and compute the empirical probability of success. When $n = 1000$, the initial mean parameter is chosen uniformly at random from the sample. When $n = \infty$, the initial mean parameter is chosen uniformly at random from the rectangle $[-2, \theta_2^* + 2] \times [-2, \theta_2^* + 2]$.

A subset of the success probabilities are shown in Table 1; see Appendix F for the full set of results. Our simulations lead to the following empirical findings about the behavior of EM on data from well-separated mixtures ($|\theta_1^* - \theta_2^*| \geq 1$). First, for $n = \infty$, EM for Model 2 finds the MLE almost always ($P_2 = 1$), while EM for Model 1 only succeeds about half the time ($P_1 \approx 0.5$). Second, for smaller $n$, EM for Model 2 still has a higher chance of success than EM for Model 1, except when the weights $w_1^*$ and $w_2^*$ are almost equal. When $w_1^* \approx w_2^* \approx 1/2$, the bias in Model 1 is not big enough to stand out from the error due to the finite sample, and hence Model 1 is more preferable. Notably,

**Success probabilities for mixtures of two Gaussians** (Section 3.2)

| Separation | Sample size | $w_1^* = 0.52$ | $w_1^* = 0.7$ | $w_1^* = 0.9$ |
|---|---|---|---|---|
| $\theta_2^* - \theta_1^* = 2$ | $n = 1000$ | 0.799 / 0.500 | 0.497 / 0.800 | 0.499 / 0.899 |
| | $n = \infty$ | 0.504 / 1.000 | 0.514 / 1.000 | 0.506 / 1.000 |

**Success probabilities for mixtures of three or four Gaussians** (Section 3.3)

| Case 1 | Case 2 | Case 3 | Case 4 |
|---|---|---|---|
| 0.164 / 0.900 | 0.167 / 1.000 | 0.145 / 0.956 | 0.159 / 0.861 |

Table 1: Success probabilities for EM on Model 1 and Model 2 (denoted $P_1$ and $P_2$, respectively), reported as $P_1$ / $P_2$.

unlike the special model in (2), highly unbalanced weights do not help EM for Model 1 due to the lack of the symmetry of the component means (i.e., we may have $\theta_1^* + \theta_2^* \neq 0$).

We conclude that over-parameterization helps EM if the two mixture components are well-separated and the mixing weights are not too close.

### 3.3 Mixtures of three or four Gaussians

We now consider a setup with mixtures of three or four Gaussians. Specifically, we consider the following four cases, each using a larger sample size of $n = 2000$:

- Case 1, mixture of three Gaussians on a line: $\boldsymbol{\theta}_1^* = (-3, 0)$, $\boldsymbol{\theta}_2^* = (0, 0)$, $\boldsymbol{\theta}_3^* = (2, 0)$ with weights $w_1^* = 0.5, w_2^* = 0.3, w_3^* = 0.2$.
- Case 2, mixture of three Gaussians on a triangle: $\boldsymbol{\theta}_1^* = (-3, 0)$, $\boldsymbol{\theta}_2^* = (0, 2)$, $\boldsymbol{\theta}_3^* = (2, 0)$ with weights $w_1^* = 0.5, w_2^* = 0.3, w_3^* = 0.2$.
- Case 3, mixture of four Gaussians on a line: $\boldsymbol{\theta}_1^* = (-3, 0)$, $\boldsymbol{\theta}_2^* = (0, 0)$, $\boldsymbol{\theta}_3^* = (2, 0)$, $\boldsymbol{\theta}_4^* = (5, 0)$ with weights $w_1^* = 0.35, w_2^* = 0.3, w_3^* = 0.2, w_4^* = 0.15$.
- Case 4, mixture of four Gaussians on a trapezoid: $\boldsymbol{\theta}_1^* = (-3, 0)$, $\boldsymbol{\theta}_2^* = (-1, 2)$, $\boldsymbol{\theta}_3^* = (2, 0)$, $\boldsymbol{\theta}_4^* = (2, 2)$ with weights $w_1^* = 0.35, w_2^* = 0.3, w_3^* = 0.2, w_4^* = 0.15$.

The other aspects of the simulations are the same as in the previous subsection.

The results are presented in Table 1. From the table, we confirm that EM for Model 2 (with unknown weights) has a higher success probability than EM for Model 1 (with known weights). Therefore, over-parameterization helps in all four cases.

### 3.4 Explaining the disparity

As discussed in Section 2.2, the performance EM algorithm with unknown weights does not depend on the ordering of the initialization means. We conjuncture that in general, this property that is a consequence of over-parameterization leads to the boost that is observed in the performance of EM with unknown weights.

We support this conjecture by revisiting the previous simulations with a different way of running EM for Model 1. For each set of $k$ vectors selected to be used as initial component means, we run EM $k!$ times, each using a different one-to-one assignment of these vectors to initial component means. We measure the empirical success probability $P_3$ based on the *lowest* observed error among these $k!$ runs of EM. The results are presented in Table 3 in Appendix F. In general, we observe $P_3 \gtrsim P_2$ for all cases we have studied, which supports our conjecture. However, this procedure is generally more time-consuming than EM for Model 2 since $k!$ executions of EM are required.

### Acknowledgements

DH and JX were partially supported by NSF awards DMREF-1534910 and CCF-1740833, and JX was also partially supported by a Cheung-Kong Graduate School of Business Fellowship. We thank Jiantao Jiao for a helpful discussion about this problem.

## Footnotes

[1]Using equal initial weights is a natural way to initialize EM when the weights are unknown.

[2]When $w_1^*$ is indeed very close to 1, then almost all of the probability mass of the true distribution comes from $C_1$, which has positive mean. So, in the M-step discussed above, the rightward pull of the mean of $\hat{C}_-$ may

[3]There exists $\tilde{w} \in (0, 1)$ such that $g_w(\theta, w)$ is concave in $[0, \tilde{w}]$ and convex in $[\tilde{w}, 1]$.

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
