[Supplementary Material · supplementary.pdf]

## A   Proof of Theorem 1

Let us define $h(\theta, w_1^*) := H(\theta; \theta^*, w_1^*)$. First, it is straightforward to show that

$$h(0, 0.5) = 0,$$

and

- $h(\theta, 0.5)$ is concave for $\theta \geq 0$ and $h(\theta^*, 0.5) = \theta^*$.
- $h(\theta, 0.5)$ is convex for $\theta \leq 0$ and $h(-\theta^*, 0.5) = -\theta^*$.

Hence, we have

$$h(\theta, 0.5) - \theta \quad = \quad \begin{cases} > 0, & \theta \in (-\infty, -\theta^*) \bigcup (0, \theta^*) \\ = 0, & \theta = -\theta^*, 0, \theta^* \\ < 0, & \theta \in (-\theta^*, 0) \bigcup (\theta^*, \infty) \end{cases} \tag{12}$$

Therefore, if we can show that the curve of $h(\theta, w_1^*)$ is strictly above the curve $h(\theta, 0.5)$ for all $w_1^* > 0.5$ and $\theta < \theta^*$, i.e.,

$$h(\theta, w_1^*) \quad > \quad h(\theta, 0.5), \qquad \forall w_1^* > 0.5, \theta < \theta^*, \tag{13}$$

then by (12), we have

$$h(\theta, w_1^*) - \theta \quad > \quad h(\theta, 0.5) - \theta \geq 0, \qquad \forall w_1^* > 0.5, \theta \leq -\theta^*. \tag{14}$$

Further, since $h$ is continuous, we know there exists $\delta > 0$ and $\theta_\delta$, such that

$$h(\theta_\delta, w_1^*) \quad < \quad \theta_\delta, \quad \forall w_1^* \in [0.5, 0.5 + \delta].$$

Hence, with (14) and continuity of function $h(\theta, w_1^*) - \theta$, we know for each $w_1^* \in (0.5, 0.5 + \delta]$, there exists $\theta_w \in (-\theta^*, 0)$ (the smallest fixed point) such that

$$h(\theta_w, w_1^*) \quad = \quad \theta_w \quad \text{and} \quad h(\theta, w_1^*) > \theta, \qquad \forall \theta \in (-\infty, \theta_w).$$

Therefore, if we initialize $\theta^{\langle 0 \rangle} \leq -\theta^*$, the EM estimate will converge to $\theta_w$. Hence, our final step is to show (14) which is proved in the following lemma:

**Lemma 4** (Proved in Appendix E.1). *For all $w_1^* \neq 0.5$, we have*

$$h(\theta, w_1^*) \quad > \quad h(\theta, 0.5), \qquad \forall \theta < \theta^*, \tag{15}$$

*and for all $w_1^* \in [0, 1]$, we have*

$$0 \leq \frac{\partial h(\theta, w_1^*)}{\partial \theta} \leq e^{-\frac{(\theta^*)^2}{2}} < 1, \qquad \forall \theta \geq \theta^*. \tag{16}$$

In fact, by Lemma 4, (12) and the fact $h(\theta^*, w) \equiv \theta^*$, it is straightforward to show the following corollary

**Corollary 1.** *For all $w_1^* \in [0, 1]$, $h(\theta, w_1^*)$ has only one fixed point (a stable fixed point) in $(0, \infty)$, which is $\theta = \theta^*$.*

## B   Proof of Theorem 2

From the discussion in Section 2.2, we just need to prove Theorem 2 for $w_1^* > 0.5$. We use the following the strategy to prove Theorem 2.

1. Prove Lemma 1 (see Section 2.3) and therefore WLOG, we can safely assume $\langle \boldsymbol{\theta}^{\langle t \rangle}, \boldsymbol{\theta}^* \rangle > 0$ and $w^{\langle t \rangle} > 0.5$ for all $t > 0$.
2. Prove Theorem 2 when the mean parameters $\theta_i^*$ is in one dimension.
3. Show that we can reduce the multi-dimensional problem into the one dimensional one.
4. Show geometric convergence by proving an attraction basin around $(\boldsymbol{\theta}^*, w_1^*)$.

Each one of the steps is proved in the following subsections in order.

## B.1  Proof of Lemma 1

First it is clear that $w_1^{\langle t\rangle} \in (0,1)$. Hence, due to our initialization setting $w_1^{\langle 0\rangle} = w_2^{\langle 0\rangle} = 0.5$, we just need to show

- For all $\langle \boldsymbol{\theta}, \boldsymbol{\theta}^*\rangle > 0, w_1 \in [0.5, 1)$, we have

$$\langle G_\theta(\boldsymbol{\theta}, w_1; \boldsymbol{\theta}^*, w_1^*), \boldsymbol{\theta}^*\rangle \; > \; 0 \quad \text{and} \quad G_w(\boldsymbol{\theta}, w_1; \boldsymbol{\theta}^*, w_1^*) \; > \; 0.5. \tag{17}$$

- For all $\langle \boldsymbol{\theta}, \boldsymbol{\theta}^*\rangle < 0, w_1 \in (0, 0.5]$, we have

$$\langle G_\theta(\boldsymbol{\theta}, w_1; \boldsymbol{\theta}^*, w_1^*), \boldsymbol{\theta}^*\rangle \; < \; 0 \quad \text{and} \quad G_w(\boldsymbol{\theta}, w_1; \boldsymbol{\theta}^*, w_1^*) \; < \; 0.5. \tag{18}$$

and then by a simple induction argument, it is straightforward to show Lemma 1 holds. Moreover, let $w_2 = 1 - w_1$ and note that the symmetric property of $G_\theta$ and $G_w$, i.e.,

$$\begin{aligned}
G_\theta(\boldsymbol{\theta}, w_1; \boldsymbol{\theta}^*, w_1^*) + G_\theta(-\boldsymbol{\theta}, w_2; \boldsymbol{\theta}^*, w_1^*) &= 0 \\
G_w(\boldsymbol{\theta}, w_1; \boldsymbol{\theta}^*, w_1^*) + G_w(-\boldsymbol{\theta}, w_2; \boldsymbol{\theta}^*, w_1^*) &= 1.
\end{aligned}$$

Hence, we just need to show (17) holds. Since for any orthogonal matrices $V$, we have

$$\begin{aligned}
\langle G_\theta(\boldsymbol{\theta}, w_1; \boldsymbol{\theta}^*, w_1^*), \boldsymbol{\theta}^*\rangle &= \langle G_\theta(V\boldsymbol{\theta}, w_1; V\boldsymbol{\theta}^*, w_1^*), V\boldsymbol{\theta}^*\rangle \\
G_w(\boldsymbol{\theta}, w_1; \boldsymbol{\theta}^*, w_1^*) &= G_w(V\boldsymbol{\theta}, w_1; V\boldsymbol{\theta}^*, w_1^*)
\end{aligned}$$

Hence, the claim made in (17) and (18) is invariant to rotation of the coordinates. Hence, WLOG, we assume that $\boldsymbol{\theta} = (\|\boldsymbol{\theta}\|, 0, 0, \ldots, 0)$ and $\boldsymbol{\theta}^* = (\theta_\|^*, \theta_\perp^*, 0, \ldots, 0)$ with $\theta_\|^* > 0$. Let us first show $G_w(\boldsymbol{\theta}, w; \boldsymbol{\theta}^*, w_1^*) > 0.5$. It is straightforward to show that

$$\begin{aligned}
G_w(\boldsymbol{\theta}, w_1; \boldsymbol{\theta}^*, w_1^*) &= \int \frac{w_1 e^{y\|\theta\|}}{w_1 e^{y\|\theta\|} + w_2 e^{-y\|\theta\|}} \left( w_1^* \phi(y - \theta_\|^*) + w_2^* \phi(y + \theta_\|^*)\right) \mathrm{d}y \\
&=: \; g_w(\|\theta\|, w_1; \theta_\|^*, w_1^*),
\end{aligned}$$

where $\phi(x)$ denotes the pdf for $d'-$dimensional standard Gaussian if $x \in \mathbb{R}^{d'}$. Hence, we just need to show that

$$g_w(\theta, w_1; \theta^*, w_1^*) \; > \; 0.5, \quad \forall w_1 \in [0.5, 1), w_1^* \in (0.5, 1), \theta > 0, \theta^* > 0. \tag{19}$$

Note that

$$\frac{\partial g_w(\theta, w_1; \theta^*, w_1^*)}{\partial w_1} \; = \; \int \frac{1}{\left(w_1 e^{y\theta} + w_2 e^{-y\theta}\right)^2}\left(w_1^* \phi(y - \theta^*) + w_2^* \phi(y + \theta^*)\right)\mathrm{d}y \; > \; 0.$$

Hence, we just need to show $g_w(\theta, 0.5; \theta^*, w_1^*) > 0.5$. Note that

$$\begin{aligned}
g_w(\theta, 0.5; \theta^*, w_1^*) - 0.5 &= \int \frac{e^{y\theta}}{e^{y\theta} + e^{-y\theta}}\left(w_1^* \phi(y - \theta^*) + w_2^* \phi(y + \theta^*)\right)\mathrm{d}y - 0.5 \\
&= \int \frac{e^{y\theta} - e^{-y\theta}}{2(e^{y\theta} + e^{-y\theta})}\left(w_1^* \phi(y - \theta^*) + w_2^* \phi(y + \theta^*)\right)\mathrm{d}y \\
&= \int_{y \geq 0} \phi(y) e^{-\frac{(\theta^*)^2}{2}} \cdot \left(\frac{(2w_1^* - 1)\left(\cosh_y(\theta^* + \theta) - \cosh_y(\theta^* - \theta)\right)}{2\cosh_y(\theta)}\right)\mathrm{d}y \\
&> 0,
\end{aligned}$$

where $\cosh_y(x) = \frac{1}{2}(e^{yx} + e^{-yx})$. Hence, (19) holds. Now we just need to show $\langle G_\theta(\boldsymbol{\theta}, w_1; \boldsymbol{\theta}^*, w_1^*), \boldsymbol{\theta}^*\rangle > 0$. It is straightforward to show that all components of $G_\theta(\boldsymbol{\theta}, w_1; \boldsymbol{\theta}^*, w_1^*)$ are 0 except for the first two components denoted as $\tilde{\theta}_1$ and $\tilde{\theta}_2$. For the second component $\tilde{\theta}_2$, we have

$$\begin{aligned}
\tilde{\theta}_2 &= \theta_\perp^* \int \frac{w_1 e^{y\|\theta\|} - w_2 e^{-y\|\theta\|}}{w_1 e^{y\|\theta\|} + w_2 e^{-y\|\theta\|}}\left(w_1^* \phi(y - \theta_\|^*) - w_2^* \phi(y + \theta_\|^*)\right)\mathrm{d}y \\
&=: \; \theta_\perp^* \cdot s(\|\theta\|, w_1; \theta_\|^*, w_1^*),
\end{aligned} \tag{20}$$

and for the first component $\tilde{\theta}_1$, we have

$$
\begin{aligned}
\tilde{\theta}_1 &= \theta_\|^* \int \frac{w_1 e^{y\|\theta\|} - w_2 e^{-y\|\theta\|}}{w_1 e^{y\|\theta\|} + w_2 e^{-y\|\theta\|}} \left( w_1^* \phi(y - \theta_\|^*) - w_2^* \phi(y + \theta_\|^*) \right) \mathrm{d}y \\
&\quad + \int \frac{w_1 e^{y\|\theta\|} - w_2 e^{-y\|\theta\|}}{w_1 e^{y\|\theta\|} + w_2 e^{-y\|\theta\|}} \left( w_1^*(y - \theta_\|^*) \phi(y - \theta_\|^*) + w_2^*(y + \theta_\|^*) \phi(y + \theta_\|^*) \right) \mathrm{d}y \\
&\stackrel{(a)}{=} \theta_\|^* \cdot s(\|\theta\|, w_1; \theta_\|^*, w_1^*) + \|\theta\| \int \frac{4 w_1 w_2}{\left( w_1 e^{y\|\theta\|} + w_2 e^{-y\|\theta\|} \right)^2} \left( w_1^* \phi(y - \theta_\|^*) + w_2^* \phi(y + \theta_\|^*) \right) \mathrm{d}y \\
&> \theta_\|^* \cdot s(\|\theta\|, w_1; \theta_\|^*, w_1^*),
\end{aligned}
\tag{21}
$$

where equation (a) holds due to partial integration. Hence, by (20) and (21) and $\theta_\|^* > 0$, we have

$$
\langle G_\theta(\boldsymbol{\theta}, w_1; \boldsymbol{\theta}^*, w_1^*), \boldsymbol{\theta}^* \rangle > \|\boldsymbol{\theta}^*\|^2 \cdot s(\|\theta\|, w_1; \theta_\|^*, w_1^*).
$$

Hence, we just need to show

$$
s(\theta, w_1; \theta^*, w_1^*) > 0, \quad \forall \theta > 0, w_1 \in [0.5, 1], \theta^* > 0, w_1^* \in (0.5, 1).
\tag{22}
$$

For $w_1 = 0.5$, by (20), we have

$$
\begin{aligned}
s(\theta, 0.5; \theta^*, w_1^*) &= \int \frac{e^{y\theta} - e^{-y\theta}}{e^{y\theta} + e^{-y\theta}} \left( w_1^* \phi(y - \theta^*) - w_2^* \phi(y + \theta^*) \right) \mathrm{d}y \\
&= \int_{y \geq 0} \frac{e^{y\theta} - e^{-y\theta}}{e^{y\theta} + e^{-y\theta}} \phi(y) e^{-\frac{(\theta^*)^2}{2}} \left( e^{y\theta^*} - e^{-y\theta^*} \right) \mathrm{d}y > 0.
\end{aligned}
\tag{23}
$$

For $w_1 \in (0.5, 1]$, by (20) and taking derivative with respect to $w_1^*$, we have

$$
\begin{aligned}
\frac{\partial s(\theta, w_1; \theta^*, w_1^*)}{\partial w_1^*} &= \int \frac{w_1 e^{y\theta} - w_2 e^{-y\theta}}{w_1 e^{y\theta} + w_2 e^{-y\theta}} \left( \phi(y - \theta^*) + \phi(y + \theta^*) \right) \mathrm{d}y \\
&= \int_{y \geq 0} \frac{2(w_1^2 - w_2^2)}{\left( w_1 e^{y\theta} + w_2 e^{-y\theta} \right) \left( w_1 e^{-y\theta} + w_2 e^{y\theta} \right)} \left( \phi(y - \theta^*) + \phi(y + \theta^*) \right) \mathrm{d}y \\
&> 0.
\end{aligned}
$$

Hence, we just need to show

$$
s(\theta, w_1; \theta^*, 0.5) \geq 0, \quad \forall \theta > 0, w_1 \in (0.5, 1], \theta^* > 0.
\tag{24}
$$

Note that

$$
\begin{aligned}
2s(\theta, w_1; \theta^*, 0.5) &= \int \frac{w_1 e^{y\theta} - w_2 e^{-y\theta}}{w_1 e^{y\theta} + w_2 e^{-y\theta}} \left( \phi(y - \theta^*) - \phi(y + \theta^*) \right) \mathrm{d}y \\
&= \int_{y \geq 0} \frac{w_1 w_2 (e^{2y\theta} - e^{-2y\theta})}{\left( w_1 e^{y\theta} + w_2 e^{-y\theta} \right) \left( w_1 e^{-y\theta} + w_2 e^{y\theta} \right)} \left( \phi(y - \theta^*) - \phi(y + \theta^*) \right) \mathrm{d}y \\
&\geq 0.
\end{aligned}
$$

Hence, we have (24) holds. Combine with (23), we have (22) holds which completes the proof of this lemma.

## B.2 Proof of Theorem 2 in one dimension

We filled out the proofs that have left out in Section 2.3, namely Lemma 2, Lemma 3 and C.2c.

### B.2.1 Proof of Lemma 2

Based on $(\theta_\star, w_\star)$, we divide the region of $S - \{(\theta_\star, w_\star)\}$ into 8 pieces:

- $R_1 = \{(\theta, w) \in S : \theta \in [\theta_\star, \min\{r(a_w), b_\theta\}), w \in (a_w, w_\star]\} - \{(\theta_\star, w_\star)\}$.
- $R_2 = \{(\theta, w) \in S : \theta \in [\theta_\star, \min\{r(a_w), b_\theta\}), w \in [w_\star, b_w)\} - \{(\theta_\star, w_\star)\}$.
- $R_3 = \{(\theta, w) \in S : \theta \in (\max\{r(b_w), a_\theta\}, \theta_\star], w \in (a_w, w_\star]\} - \{(\theta_\star, w_\star)\}$.

- $R_4 = \{(\theta, w) \in S : \theta \in (\max\{r(b_w), a_\theta\}, \theta_\star], w \in [w_\star, b_w)\} - \{(\theta_\star, w_\star)\}$.
- $R_5 = \{(\theta, w) \in S : \theta \leq r(b_w), w \in (a_w, w_\star]\}$.
- $R_6 = \{(\theta, w) \in S : \theta \leq r(b_w), w \in [w_\star, b_w)\}$.
- $R_7 = \{(\theta, w) \in S : \theta \geq r(a_w), w \in (a_w, w_\star]\}$.
- $R_8 = \{(\theta, w) \in S : \theta \geq r(a_w), w \in [w_\star, b_w)\}$.

Note that region $R_5$ to $R_8$ may not exists depending on the range of $r(w)$. Next, due to C.2a, we know the reference curve only crosses region $R_1$ and $R_4$. Note that $r^{-1}(\theta)$ exists on the regions $R_1, R_2, R_3$ and $R_4$. Hence, based on the points are above or below the reference curve $r$, we can further divide the region $R_1$ and $R_4$ into 4 pieces:

- $R_{11} = \{(\theta, w) \in R_1 : r^{-1}(\theta) \leq w\}$.
- $R_{12} = \{(\theta, w) \in R_1 : r^{-1}(\theta) \geq w\}$.
- $R_{41} = \{(\theta, w) \in R_4 : w \leq r^{-1}(\theta)\}$.
- $R_{42} = \{(\theta, w) \in R_4 : w \geq r^{-1}(\theta)\}$.

Now let's define $m : S \to [0, \infty)$ based on the following 10 regions

$$\{R_{11}, R_{12}, R_2, R_3, R_{41}, R_{42}, R_5, R_6, R_7, R_8\} :$$

- If $(\theta, w) \in R_{11}$, $m(\theta, w) = (w_\star - w)(r(w) - \theta_\star)$, which is the area of the rectangle $D(\theta, w)$ given by $(\theta_\star, w_\star), (r(w), w)$.
- If $(\theta, w) \in R_{12}$, $m(\theta, w) = (w_\star - r^{-1}(\theta))(\theta - \theta_\star)$, which is the area of the rectangle $D(\theta, w)$ given by $(\theta_\star, w_\star), (\theta, r^{-1}(\theta))$.
- If $(\theta, w) \in R_2$, $m(\theta, w) = (w - r^{-1}(\theta))(\theta - r(w))$, which is the area of the rectangle $D(\theta, w)$ given by $(r(w), r^{-1}(\theta)), (\theta, w)$.
- If $(\theta, w) \in R_3$, $m(\theta, w) = (r^{-1}(\theta) - w)(r(w) - \theta)$, which is the area of the rectangle $D(\theta, w)$ given by $(r(w), r^{-1}(\theta)), (\theta, w)$.
- If $(\theta, w) \in R_{41}$, $m(\theta, w) = (r^{-1}(\theta) - w_\star)(\theta_\star - \theta)$, which is the area of the rectangle $D(\theta, w)$ given by $(\theta_\star, w_\star), (\theta, r^{-1}(\theta))$.
- If $(\theta, w) \in R_{42}$, $m(\theta, w) = (w - w_\star)(\theta_\star - r(w))$, which is the area of the rectangle $D(\theta, w)$ given by $(\theta_\star, w_\star), (r(w), w)$.
- If $(\theta, w) \in R_5$, $m(\theta, w) = (b_w - w)(r(w) - \theta)$, which is the area of the rectangle $D(\theta, w)$ given by $(r(w), b_w), (\theta, w)$.
- If $(\theta, w) \in R_6$, $m(\theta, w) = (b_w - w_\star)(\theta_\star - \theta)$, which is the area of the rectangle $D(\theta, w)$ given by $(\theta, b_w), (\theta_\star, w_\star)$.
- If $(\theta, w) \in R_7$, $m(\theta, w) = (w_\star - a_w)(\theta - \theta_\star)$, which is the area of the rectangle $D(\theta, w)$ given by $(\theta_\star, w_\star), (\theta, a_w)$.
- If $(\theta, w) \in R_8$, $m(\theta, w) = (w - a_w)(\theta - r(w))$, which is the area of the rectangle $D(\theta, w)$ given by $(r(w), a_w), (\theta, w)$.

It is straightforward to show that function $m$ is a continuous function by checking the boundary and continuity of the reference function $r$. Further, $(\theta_\star, w_\star)$ is indeed the only solution for $m(\theta, w) = 0$. Moreover, our construction of the rectangle $D$ makes sure that

$$\text{If } (\tilde{\theta}, \tilde{w}) \text{ is strictly inside } D(\theta, w), \text{ then } D(\tilde{\theta}, \tilde{w}) \subsetneq D(\theta, w). \tag{25}$$

Next, we shall discuss the movement of the iterates from point $(\theta^{\langle t \rangle}, w^{\langle t \rangle})$ to point $(\theta^{\langle t+1 \rangle}, w^{\langle t+1 \rangle})$. For a given $w^{\langle t \rangle} \in [a_w, b_w]$, consider all the fixed points $\mathcal{V}$ in $[a_\theta, b_\theta]$ for $g_\theta(\theta, w)$ with respect to $\theta$. Then, for any $\theta^{\langle t \rangle} \in (a_\theta, b_\theta)$, it should be inside an interval defined by $[q_1, q_2]$ where $q_1, q_2 \in \mathcal{V} \bigcup \{a_\theta, b_\theta\}$ and at least one of $q_1$ or $q_2$ is either a stable fixed point or one of $a_\theta, b_\theta$. Further, since $g_\theta(\theta, w)$ is a non-decreasing function of $\theta$ and $(\theta^{\langle t+1 \rangle}, w^{\langle t+1 \rangle}) \in S$, we know $\theta^{\langle t+1 \rangle} = g_\theta(\theta^{\langle t \rangle}, w^{\langle t \rangle}) \in [q_1, q_2]$ as well. Hence, comparing to the previous iteration $\theta^{\langle t \rangle}$, $\theta^{\langle t+1 \rangle} = g_\theta(\theta^{\langle t \rangle}, w^{\langle t \rangle})$ should (i) stay at a fixed point, i.e., $q_1$ or $q_2$ or (ii) move towards a stable fixed point $q_i$ or $a_\theta, b_\theta$. Further, if $\theta^{\langle t+1 \rangle}$ moves towards $a_\theta$ or $b_\theta$, then $a_\theta$ or $b_\theta$ has to be a stable fixed point as well. In other words, suppose $\theta^{\langle t+1 \rangle}$ move towards $a_\theta$ and $a_\theta$ is not a stable fixed point. Then $a_\theta$ is not a fixed point as well and there exists a constant $c > 0$ such that

$\lim_{\theta \to a_\theta} g_\theta(\theta, w^{\langle t \rangle}) \le a_\theta - c$. Hence by choosing $\theta$ close enough to $a_\theta$, we know $g_\theta(\theta, w) < a_\theta$ which contradicts C.1. Now, by C.2b, C.2c and discussing which region $(\theta, w)$ belongs to, we can prove

Point $(\theta^{\langle t+1 \rangle}, w^{\langle t+1 \rangle})$ is strictly inside $D(\theta^{\langle t \rangle}, w^{\langle t \rangle})$ and $m(\theta^{\langle t+1 \rangle}, w^{\langle t+1 \rangle}) < m(\theta^{\langle t \rangle}, w^{\langle t \rangle})$. (26)

and

If $(\theta^{\langle t \rangle}, w^{\langle t \rangle}) \in R_1 \bigcup R_2 \bigcup R_3 \bigcup R_4$, then $(\theta^{\langle t+1 \rangle}, w^{\langle t+1 \rangle}) \in R_1 \bigcup R_2 \bigcup R_3 \bigcup R_4$. (27)

Note that depending on the regions, there are total 10 cases. But for simplicity, we show the proof for two cases: $R_{11}$ and $R_6$ and leave the rest of the cases to the readers. For the first example, if point $(\theta^{\langle t \rangle}, w^{\langle t \rangle}) \in R_{11}$, then we know there exists a fixed point $\theta_s \in [\theta_\star, b_\theta]$ for $g_\theta$ and $w_s \in [a_w, w_\star]$ for $g_w$ such that $\theta^{\langle t+1 \rangle} = g_\theta(\theta^{\langle t \rangle}, w^{\langle t \rangle})$ lies in between $\theta^{\langle t \rangle}$ and $\theta_s$, and $w^{\langle t+1 \rangle} = g_w(\theta^{\langle t \rangle}, w^{\langle t \rangle})$ lies in between $w^{\langle t \rangle}$ and $w_s$. Hence $(\theta^{\langle t+1 \rangle}, w^{\langle t+1 \rangle})$ can only stay in $R_1$ which proves (27) for the case $(\theta^{\langle t \rangle}, w^{\langle t \rangle}) \in R_{11}$. Further, we have

$$|g_\theta(\theta^{\langle t \rangle}, w^{\langle t \rangle}) - \theta_s| \le |\theta^{\langle t \rangle} - \theta_s|, \qquad (28)$$
$$|g_w(\theta^{\langle t \rangle}, w^{\langle t \rangle}) - w_s| \le |w^{\langle t \rangle} - w_s|, \qquad (29)$$

where equality (28)/(29) holds if and only if $\theta^{\langle t \rangle} = \theta_s / w^{\langle t \rangle} = w_s$. Hence, by C.2, we have

- If $\theta^{\langle t \rangle} = \theta_\star$, then $w^{\langle t \rangle} < w_\star$. Hence we have $\theta_s \in (\theta_\star, r(w^{\langle t \rangle}))$ and $w_s = w_\star$. and therefore, (29) is strict inequality. Hence, $w^{\langle t \rangle} < w^{\langle t+1 \rangle}$.
- If $\theta^{\langle t \rangle} > \theta_\star$, then $\max(\theta_s, \theta^{\langle t \rangle}) \le r(w^{\langle t \rangle})$ and $w_s > r^{-1}(\theta^{\langle t \rangle}) \ge w^{\langle t \rangle}$, therefore,

$$\theta^{\langle t+1 \rangle} = g_\theta(\theta^{\langle t \rangle}, w^{\langle t \rangle}) \le r(w^{\langle t \rangle}), \quad \text{and} \quad w^{\langle t \rangle} < g_w(\theta^{\langle t \rangle}, w^{\langle t \rangle}) = w^{\langle t+1 \rangle}. \qquad (30)$$

Therefore point $(\theta^{\langle t+1 \rangle}, w^{\langle t+1 \rangle})$ lies in the rectangle $D(\theta^{\langle t \rangle}, w^{\langle t \rangle})$ no matter what. Further, due to monotonic property of function $r$, we have

$$r(w^{\langle t \rangle}) > r(g_w(\theta^{\langle t \rangle}, w^{\langle t \rangle})). \qquad (31)$$

Hence, by (30) and (31), no matter what region $R_{11}$ or $R_{12}$ contains the point $(\theta^{\langle t+1 \rangle}, w^{\langle t+1 \rangle})$, the rectangle $D(\theta^{\langle t+1 \rangle}, w^{\langle t+1 \rangle})$ is strictly smaller than the rectangle $D(\theta^{\langle t \rangle}, w^{\langle t \rangle})$. Hence, we have (26) holds for the case $(\theta^{\langle t \rangle}, w^{\langle t \rangle}) \in R_{11}$. For the second example that if $(\theta, w) \in R_6$, then by C.2, we know there exists a fixed point $\theta_s \in (r(b_w), \theta_\star]$ for $g_\theta$ and $w_s \in [w_\star, b_w]$ for $g_w$ such that $\theta^{\langle t+1 \rangle} = g_\theta(\theta^{\langle t \rangle}, w^{\langle t \rangle})$ lies in between $\theta^{\langle t \rangle}$ and $\theta_s$; and $w^{\langle t+1 \rangle} = g_w(\theta^{\langle t \rangle}, w^{\langle t \rangle})$ lies in between $w^{\langle t \rangle}$ and $w_s$. Hence, point $(\theta^{\langle t+1 \rangle}, w^{\langle t+1 \rangle})$ can only stay in the region $R_6$ or $R_4$. Further, we have

$$|g_\theta(\theta^{\langle t \rangle}, w^{\langle t \rangle}) - \theta_s| \le |\theta^{\langle t \rangle} - \theta_s|,$$

where equality holds if and only if $\theta^{\langle t \rangle} = \theta_s$. Therefore, we have

$$\theta^{\langle t+1 \rangle} = g_\theta(\theta^{\langle t \rangle}, w^{\langle t \rangle}) > \theta^{\langle t \rangle},$$

and hence, no matter what region $R_6$ or $R_4$ contains the point $(\theta^{\langle t+1 \rangle}, w^{\langle t+1 \rangle})$, the rectangle $D(\theta^{\langle t+1 \rangle}, w^{\langle t+1 \rangle})$ is strictly smaller than the rectangle $D(\theta^{\langle t \rangle}, w^{\langle t \rangle})$. Similarly, we can show (26) holds for all other cases. Next, we claim that if point $(\theta^{\langle 0 \rangle}, w^{\langle 0 \rangle}) \in R_5 \bigcup R_6 \bigcup R_7 \bigcup R_8$, then within finite steps $t_0$, the estimate $(\theta^{\langle t_0 \rangle}, w^{\langle t_0 \rangle})$ should lie in the region $R_1 \bigcup R_2 \bigcup R_3 \bigcup R_4$. Suppose point $(\theta^{\langle 0 \rangle}, w^{\langle 0 \rangle}) \in R_6$, $g_\theta(\theta, w)/\theta$ is continuous on $[\theta^{\langle 0 \rangle}, r(b_w)] \times [w_\star, b_w]$. Further, due to (26), we have

$$g_\theta(\theta, w)/\theta > 1, \quad \forall (\theta, w) \in [\theta^{\langle 0 \rangle}, r(b_w)] \times [w_\star, b_w].$$

Therefore, there exists a constant $\rho > 1$ such that $g_\theta(\theta, w) \geq \rho\theta$ on $[\theta^{\langle 0 \rangle}, r(b_w)] \times [w_\star, b_w]$. Hence, within finite steps, we have $(\theta^{\langle t_0 \rangle}, w^{\langle t_0 \rangle}) \in R_1 \bigcup R_2 \bigcup R_3 \bigcup R_4$. Similarly we can show for $(\theta^{\langle 0 \rangle}, w^{\langle 0 \rangle}) \in R_5, R_7, R_8$ as well. Hence, by (27), we just need to focus on $(\theta^{\langle 0 \rangle}, w^{\langle 0 \rangle}) \in R_1 \bigcup R_2 \bigcup R_3 \bigcup R_4$. Now we use contradiction to prove that $m(\theta^{\langle t \rangle}, w^{\langle t \rangle})$ converges to 0. Suppose $m(\theta^{\langle t \rangle}, w^{\langle t \rangle})$ does not converge to 0, then by definition of $m$, we know there exists some constant $c_\theta > 0$ and $c_w > 0$, such that

$$|\theta_\star - \theta^{\langle t \rangle}| \geq c_\theta \quad \text{and} \quad |w_\star - w^{\langle t \rangle}| \geq c_w, \quad \forall t \geq 0. \tag{32}$$

Further, since $S \supset D(\theta^{\langle 0 \rangle}, w^{\langle 0 \rangle}) \supset D(\theta^{\langle 1 \rangle}, w^{\langle 1 \rangle}) \supset \cdots$, we know all points $(\theta^{\langle t \rangle}, w^{\langle t \rangle})$ are bounded on a compact set $D(\theta^{\langle 0 \rangle}, w^{\langle 0 \rangle})$. Now consider function

$$U(\theta^{\langle t \rangle}, w^{\langle t \rangle}) := \frac{m(\theta^{\langle t+1 \rangle}, w^{\langle t+1 \rangle})}{m(\theta^{\langle t \rangle}, w^{\langle t \rangle})}$$

we know $U$ is continuous on $(\theta^{\langle t \rangle}, w^{\langle t \rangle}) \in Q = \{(\theta, w_1) \in D(\theta^{\langle 0 \rangle}, w^{\langle 0 \rangle}) : |\theta_\star - \theta| \geq c_\theta, |w_\star - w| \geq c_w\}$. Further, since $Q$ is a compact set and $U < 1$ on $Q$, we know there exists constant $\rho < 1$ such that $\sup_Q U(\theta, w) \leq \rho$. Hence, we have $m(\theta^{\langle t \rangle}, w^{\langle t \rangle})$ converges to 0. Therefore, $(\theta^{\langle t \rangle}, w^{\langle t \rangle})$ converges to $(\theta_\star, w_\star)$ since it is the only solution for $m = 0$ and $m$ is continuous.

### B.2.2 Proof of Lemma 3

We study the shape of $g_w$ by its first, second and third derivatives. Note that (with $w_2 = 1 - w_1$)

$$\frac{\partial g_w(\theta, w_1)}{\partial w_1} = \mathbb{E}_{y \sim f^*} \left[ \frac{1}{\left( w_1 e^{y\theta} + w_2 e^{-y\theta} \right)^2} \right] > 0 \tag{33}$$

$$\frac{\partial^2 g_w(\theta, w_1)}{\partial w_1^2} = \mathbb{E}_{y \sim f^*} \left[ \frac{e^{-y\theta} - e^{y\theta}}{\left( w_1 e^{y\theta} + w_2 e^{-y\theta} \right)^3} \right] \tag{34}$$

$$\frac{\partial^3 g_w(\theta, w_1)}{\partial w_1^3} = \mathbb{E}_{y \sim f^*} \left[ \frac{\left( e^{y\theta} - e^{-y\theta} \right)^2}{\left( w_1 e^{y\theta} + w_2 e^{-y\theta} \right)^4} \right] > 0 \tag{35}$$

Hence, by (35), we know the second derivative $\frac{\partial^2 g_w(\theta, w_1)}{\partial w_1^2}$ is a strictly increasing function of $w_1$ if $\theta \neq 0$. Hence, the second derivative can only change the sign at most once, the shape of $g_w$ can only be one of the following three cases: (i) concave (the second derivative is always negative), (ii) concave-convex (the second derivative is negative, then positive) and (iii) convex (the second derivative is always positive). Note that by Lemma 1, we know $g_w(\theta, 0.5) > 0.5$ if $\theta > 0$. Moreover, it is easy to check that $g(\theta, 0) = 0$ and $g(\theta, 1) = 1$. Hence, we know for $\theta > 0$, the shape of $g_w$ can only be either case (i) or case (ii). For case (i), it is clear that we have 1 is the only stable fixed point and

$$g_w(\theta, w_1) > w_1 \quad \text{is equivalent to} \quad w_1 \in (0, 1). \tag{36}$$

For case (ii), then depends on the value of the derivative at $w_1 = 1$ i.e., $\partial g_w(\theta, w_1)/\partial w_1|_{w_1=1}$, we have

- If $\partial g_w(\theta, w_1)/\partial w_1|_{w_1=1} \leq 1$, $w_1 = 1$ is the stable fixed point and (36) holds.
- If $\partial g_w(\theta, w_1)/\partial w_1|_{w_1=1} < 1$, then $w_1 = 1$ is only a fixed point and there exists a stable fixed point in $(0, 1)$ such that (10) holds.

### B.3 Proof of C.2b

According to (9), function $r$ is a one to one mapping between $w \in (0.5, 1]$ and $\theta \in [(w_1^* - w_2^*)\theta^*, \infty)$. Hence, we can simplify C.2b as

- If $w_1 \in (w_1^*, 1]$, then $w_1 > w_s > w_1^*$,
- If $w_1 = w_1^*$, then $w_1 = w_s = w_\star$,
- If $w_1 \in (0.5, w_1^*)$, then $w_1 < w_s < w_1^*$,

where $w_s$ is any stable fixed point in $[a_w, b_w]$ or fixed point in $(a_w, b_w)$ for $\theta = r(w_1)$. By (10) in Lemma 3, we can complete the proof for C.2b by showing the following technical lemma proved in Appendix E.2:

**Lemma 5.** *Let* $\gamma = \frac{2w_1^* - 1}{2w_1 - 1}$, *we have*

$$
\begin{aligned}
g_w(\gamma\theta^*, w_1) &< w_1 \quad \text{and} \quad g_w(\gamma\theta^*, w_1^*) > w_1^* \quad \forall w_1 \in (w_1^*, 1] \\
g_w(\gamma\theta^*, w_1) &> w_1 \quad \text{and} \quad g_w(\gamma\theta^*, w_1^*) < w_1^* \quad \forall w_1 \in (0.5, w_1^*)
\end{aligned}
$$

### B.3.1   Proof of C.2c

Recall our construction of the adjusted reference curve $r_{adj}$ in Section 2.3, we have

$$
r_{adj}(w) = r(w) - \epsilon \cdot \max(0, w - 1 + \delta) = \frac{2w_1^* - 1}{2w - 1}\theta^* - \epsilon \cdot \max(0, w - 1 + \delta),
$$

for some positive $\epsilon, \delta > 0$. Also, note that $g_\theta(\theta, 1) \equiv (2w_1^* - 1)\theta^*$. Hence, we just need to show the following

C.2c' Given $w_1 \in (a_w, b_w)$, any stable fixed point $\theta_s$ of $g_\theta(\theta, w)$ in $[a_\theta, b_\theta]$ or fixed point $\theta_s$ in $(a_\theta, b_\theta)$ satisfies that

  – If $w_1 < w_\star$, then $r(w) > \theta_s > \theta_\star$.
  – If $w_1 = w_\star$, then $r(w) = \theta_s = \theta_\star$.
  – If $w_1 > w_\star$, then $r(w) < \theta_s < \theta_\star$.

Like the proof for C.2b shown in Section 2.3, we first show that there exists stable fixed point for $g_\theta(\theta, w_1)$ with respect to $\theta$, i.e.,

Claim 1 If $w_1 \in (0.5, w_1^*]$, then there exists an unique non-negative fixed point for $g_\theta(\theta, w_1)$ denoted as $F_\theta(w_1)$. Further, $F_\theta(w_1) \geq \theta^*$.

Claim 2 If $w_1 \in (w_1^*, 1]$, then there exists positive stable fixed point for $g_\theta(\theta, w_1)$ and all non-negative fixed points are in $(0, \theta^*)$.

First, it is clear that $\theta = 0$ is not a fixed point for $w_1 > 0.5$ and $w_1^* > 0.5$, therefore, we just need to consider $\theta > 0$. Then, to prove Claim 1 and Claim 2, we should find out the shape of $g_\theta(\theta, w_1)$ for different true values $(\theta^*, w_1^*)$. Notice that, by Lemma 4, we know the shape of $H(\theta, w_1; \theta^*) = G_\theta(\theta, w_1; \theta^*, w_1)$, i.e., for $\theta > 0, w_1 \in [0.5, 1]$

$$
H(\theta, w_1; \theta^*) \gtreqless \theta \quad \text{is equivalent to} \quad \theta \lesseqgtr \theta^*. \tag{37}
$$

Hence, our next step to compare $G_\theta(\theta, w_1; \theta^*, w_1^*)$ with $H(\theta, w_1; \theta^*) = G_\theta(\theta, w_1; \theta^*, w_1)$. Note that, we have

$$
\begin{aligned}
\frac{\partial G_\theta(\theta, w_1; \theta^*, w_1^*)}{\partial w_1^*} &= \int y \frac{w_1 e^{y\theta} - w_2 e^{-y\theta}}{w_1 e^{y\theta} + w_2 e^{-y\theta}} \left(\phi(y - \theta^*) - \phi(y + \theta^*)\right) \mathrm{d}y \\
&= \int_{y \geq 0} \left( \frac{w_1 e^{y\theta} - w_2 e^{-y\theta}}{w_1 e^{y\theta} + w_2 e^{-y\theta}} + \frac{w_1 e^{-y\theta} - w_2 e^{y\theta}}{w_1 e^{-y\theta} + w_2 e^{y\theta}} \right) y \left(\phi(y - \theta^*) - \phi(y + \theta^*)\right) \mathrm{d}y \\
&= 2 \int_{y \geq 0} \frac{w_1 - w_2}{\left(w_1 e^{y\theta} + w_2 e^{-y\theta}\right)\left(w_1 e^{-y\theta} + w_2 e^{y\theta}\right)} y \left(\phi(y - \theta^*) - \phi(y + \theta^*)\right) \mathrm{d}y > 0.
\end{aligned}
\tag{38}
$$

Hence, if $w_1 \in (w_1^*, 1]$, we know $G_\theta$ will be strictly below $H$. Therefore

$$
G_\theta(\theta, w_1; \theta^*, w_1^*) < \theta, \quad \forall \theta \geq \theta^*.
$$

Hence, with $G_\theta(0, w_1; \theta^*, w_1^*) = (w_1 - w_2)(w_1^* - w_2^*)\theta^* > 0$ and continuity of the function, we know Claim 2 holds. Similarly, if $w_1 \in (0.5, w_1^*]$, we know $G_\theta$ will be strictly above $H$. Therefore

$$
G_\theta(\theta, w_1; \theta^*, w_1^*) > \theta, \quad \forall 0 < \theta \leq \theta^*.
$$

Hence, to prove Claim 1, we just need to show that $G_\theta(\theta, w_1; \theta^*, w_1^*)$ is bounded by some constant $C$ and

$$\frac{\partial G_\theta(\theta, w_1; \theta^*, w_1^*)}{\partial \theta} \quad < \quad 1, \quad \forall \theta \geq \theta^*, 0.5 < w_1 \leq w_1^*. \tag{39}$$

To prove boundedness, we have the following more general lemma:

**Lemma 6** (Proved in Appendix E.3). *Given any* $(\boldsymbol{\theta}, w_1, \boldsymbol{\theta}^*, w_1^*)$, *we have*

$$\|G_\theta(\boldsymbol{\theta}, w_1; \boldsymbol{\theta}^*, w_1^*)\|^2 \leq 1 + \|\boldsymbol{\theta}^*\|^2.$$

*Hence, for all* $t \geq 1$, $\|\boldsymbol{\theta}^{\langle t \rangle}\|^2 \leq \|\boldsymbol{\theta}^*\|^2 + 1$.

To prove (39), we have for $\theta \geq \theta^*$,

$$
\begin{aligned}
\frac{\partial G_\theta(\theta, w_1; \theta^*, w_1^*)}{\partial \theta} \quad &= \quad \int \frac{4 w_1 w_2}{\left(w_1 e^{y\theta} + w_2 e^{-y\theta}\right)^2} y^2 \left(w_1^* \phi(y - \theta^*) + w_2^* \phi(y + \theta^*)\right) \mathrm{d}y \\
&= \quad \frac{\partial H(\theta, w_1; \theta^*)}{\partial \theta} + (w_1^* - w_1) \int \frac{4 w_1 w_2}{\left(w_1 e^{y\theta} + w_2 e^{-y\theta}\right)^2} y^2 \left(\phi(y - \theta^*) - \phi(y + \theta^*)\right) \mathrm{d}y \\
&\overset{(i)}{\leq} \quad \frac{\partial H(\theta, w_1; \theta^*)}{\partial \theta} \\
&\overset{(ii)}{\leq} \quad e^{-\frac{(\theta^*)^2}{2}} < 1,
\end{aligned}
$$

where inequality (ii) holds due to Lemma 4 and inequality (i) holds due to

$$w_1 e^{y\theta} + w_2 e^{-y\theta} \quad \geq \quad w_1 e^{-y\theta} + w_2 e^{y\theta}, \quad \forall \theta > 0.$$

This completes the proof for Claim 1 and Claim 2. Finally, it is straightforward to show the rest of C.2c by Claim 1 and Claim 2 and the following lemma:

**Lemma 7** (Proved in Appendix E.4).

$$
\begin{aligned}
g_\theta(\gamma \theta^*, w_1) \quad &< \quad \gamma \theta^*, \quad \forall w_1 \in (\frac{1}{2}, w_1) \tag{40} \\
g_\theta(b \theta^*, w_1) \quad &> \quad b \theta^*, \quad \forall b \in (0, \gamma], w_1 \in (w_1, 1). \tag{41}
\end{aligned}
$$

## B.4 Reduction to one dimension

In this section, we show how to reduce multi-dimensional problem into one-dimensional problem by proving the angle between the two vectors $\boldsymbol{\theta}^*$ and $\boldsymbol{\theta}^{\langle t \rangle}$ is decreasing to 0. Define

$$\beta^{\langle t \rangle} := \arccos \frac{\langle \boldsymbol{\theta}^{\langle t \rangle}, \boldsymbol{\theta}^* \rangle}{\|\boldsymbol{\theta}^{\langle t \rangle}\| \|\boldsymbol{\theta}^*\|},$$

then given $\langle \boldsymbol{\theta}^{\langle 0 \rangle}, \boldsymbol{\theta}^* \rangle > 0$, we have

- If $\beta^{\langle 0 \rangle} = 0$, then for $t \geq 1$, we have $\beta^{\langle t \rangle} = 0$, i.e., it is an one-dimensional problem.
- If $\beta^{\langle 0 \rangle} \in (0, \frac{\pi}{2})$, then for $t \geq 1$, we have $\beta^{\langle t \rangle} \in (0, \beta^{\langle t-1 \rangle})$.

We use similar strategy shown in [Xu et al., 2016] to prove this. First let us define $\alpha^{\langle t \rangle} := \arccos \frac{\langle \boldsymbol{\theta}^{\langle t \rangle}, \boldsymbol{\theta}^{\langle t+1 \rangle} \rangle}{\|\boldsymbol{\theta}^{\langle t \rangle}\| \|\boldsymbol{\theta}^{\langle t+1 \rangle}\|}$, i.e., the angle between the two vectors $\boldsymbol{\theta}^{\langle t \rangle}$ and $\boldsymbol{\theta}^{\langle t+1 \rangle}$. Then since $\langle \boldsymbol{\theta}^{\langle 0 \rangle}, \boldsymbol{\theta}^* \rangle > 0$, we have $\beta^{\langle 0 \rangle} \in [0, \frac{\pi}{2})$. Further, it is straightforward to verify that if $\beta^{\langle 0 \rangle} = 0$, we have $\beta^{\langle t \rangle} = 0, \forall t \geq 0$. Hence, with Lemma 1, from now on, we assume $\beta^{\langle t \rangle} \in (0, \frac{\pi}{2})$ and $w_1^{\langle t \rangle} \in [0.5, 1)$ for all $t \geq 0$. Therefore, we just need to show $\beta^{\langle t \rangle} < \beta^{\langle t-1 \rangle}, \forall t > 0$. To prove this, we just need to to prove the following three statements hold for $\forall t \geq 0$:

(i) $\beta^{\langle t \rangle} \in (0, \frac{\pi}{2})$.
(ii) $\alpha^{\langle t \rangle} \in (0, \beta^{\langle t \rangle})$.

(iii) $\beta^{\langle t+1\rangle} = \beta^{\langle t\rangle} - \alpha^{\langle t\rangle} \in (0, \beta^{\langle t\rangle})$.

We use induction to show (i)-(iii) by proving the following chain of arguments:

**Claim 1** If (i) holds for $t$, then (ii) holds for $t$.
**Claim 2** If (i) and (ii) hold for $t$, then (iii) holds for $t$.
**Claim 3** If (i), (ii), and (iii) hold for $t$, then (i) holds for $t + 1$.

Since (i) holds for $t = 0$ and Claim 1 holds, it suffices to prove Claims 2-3. For simplicity, we drop $\langle t\rangle$ in the notation and use $\tilde{\cdot}$ to indicate the values for the next iteration $t + 1$, i.e., $\tilde{\boldsymbol{\theta}} = \boldsymbol{\theta}^{\langle t+1\rangle}$ and $\tilde{\beta} = \beta^{\langle t+1\rangle}$. Since for any orthogonal matrix $\boldsymbol{V}$, we have

$$\boldsymbol{V}G_\theta(\boldsymbol{\theta}, w_1; \boldsymbol{\theta}^*, w_1^*), \boldsymbol{\theta}^* = G_\theta(\boldsymbol{V\theta}, w_1; \boldsymbol{V\theta}^*, w_1^*)$$
$$G_w(\boldsymbol{\theta}, w_1; \boldsymbol{\theta}^*, w_1^*) = G_w(\boldsymbol{V\theta}, w_1; \boldsymbol{V\theta}^*, w_1^*) \tag{42}$$

Hence, it is straightforward to check that the Claims are invariant under any rotation of the coordinates. Hence, WLOG, we assume that $\boldsymbol{\theta} = (\|\boldsymbol{\theta}\|, 0, 0, \ldots, 0)$ and $\boldsymbol{\theta}^* = (\theta_\|^*, \theta_\perp^*, 0, \ldots, 0)$ with $\theta_\|^* > 0$ and $|\theta_\perp^*| > 0$. Then, it is straightforward to show that all components of $\tilde{\boldsymbol{\theta}}$ are 0 except for the first two components denoted as $\tilde{\theta}_1$ and $\tilde{\theta}_2$. Hence, we just need to focus on the two-dimensional space spanned by the first two components. From (20), (21) and (22), we have $\tan\alpha < \tan\beta = |\theta_\perp|/\theta_\|$ which implies Claim 2, and $\tilde{\theta}_2/\theta_\perp^* > 0$ which implies Claim 3. Next, we want to prove the angle $\beta^{\langle t\rangle}$ is decreasing to 0. Define $\theta_\|^{\langle t\rangle} = \frac{|\langle\boldsymbol{\theta}^{\langle t\rangle}, \boldsymbol{\theta}^*\rangle|}{\|\boldsymbol{\theta}^{\langle t\rangle}\|}$ and $\theta_\perp^{\langle t\rangle} = \|\boldsymbol{\theta}^* - \theta_\|^{\langle t\rangle}\|$. Hence, to show $\beta^{\langle t\rangle}$ decreases to 0, it is equivalent to show that $\theta_\|^{\langle t\rangle}$ converges to $\|\boldsymbol{\theta}^*\|$. WLOG, we assume that $\boldsymbol{\theta}^{\langle 0\rangle} = (\|\boldsymbol{\theta}^{\langle 0\rangle}\|, 0, 0, \ldots, 0)$ and $\boldsymbol{\theta}^* = (\theta_\|^{\langle 0\rangle}, \theta_\perp^{\langle 0\rangle}, 0, \ldots, 0)$ with $\theta_\|^{\langle 0\rangle} > 0$ and $|\theta_\perp^{\langle 0\rangle}| > 0$. It is straightforward to show that the only non-zero components of $\boldsymbol{\theta}^{\langle t\rangle}$ are the first two components. Hence, we just need to analyze a two dimensional problem. Then, since $\beta^{\langle t\rangle}$ is decreasing, we have $\theta_\|^{\langle t\rangle} = \|\boldsymbol{\theta}^*\| \cdot \beta^{\langle t\rangle}$ is increasing. Hence

$$\theta_\|^{\langle t\rangle} \in [\theta_\|^{\langle 1\rangle}, \|\boldsymbol{\theta}^*\|], \quad \forall t \geq 1. \tag{43}$$

To prove the increasing sequence $\theta_\|^{\langle t+1\rangle}$ converges to $\|\boldsymbol{\theta}^*\|$, we just need to show that for any $\hat{\theta} < \|\boldsymbol{\theta}^*\|$, we can find $\theta_\|^{\langle t+1\rangle}/\theta_\|^{\langle t\rangle} \geq \rho_{\hat{\theta}}$ for some constant $\rho_{\hat{\theta}} > 1$, then with a straightforward contradiction argument, within finite iterations, we should have $\theta_\|^{\langle t'\rangle} > \hat{\theta}$ for a certain $t'$, which implies $\theta_\|^{\langle t+1\rangle}$ converges to $\|\boldsymbol{\theta}^*\|$. To find such $\rho$, note that, since $\theta_\|^{\langle t\rangle}$ is a value invariant to coordinate rotations, by (20),(21) and (22), we have $U := \theta_\|^{\langle t+1\rangle}/\theta_\|^{\langle t\rangle}$ is a continuous function of $\|\boldsymbol{\theta}^{\langle t\rangle}\|, w_1^{\langle t\rangle}$ and $\theta_\|^{\langle t\rangle}$ and

$$\theta_\|^{\langle t+1\rangle}/\theta_\|^{\langle t\rangle} > 1, \quad \forall \|\boldsymbol{\theta}^{\langle t\rangle}\| > 0, w_1^{\langle t\rangle} \in (0.5, 1], \theta_\|^{\langle t\rangle} \in [\theta_\|^{\langle 1\rangle}, \|\boldsymbol{\theta}^*\|).$$

Hence, we just need to find some constants $0 < c_1 < c_2$ and $0.5 < c_3 < 1$ such that $\|\boldsymbol{\theta}^{\langle t\rangle}\| \in [c_1, c_2]$ and $w_1^{\langle t\rangle} \in [c_3, 1]$ for $t \geq 1$, then we can find $\rho$ by the uniform continuity argument. From Lemma 6, we have $c_2 = 1 + \|\boldsymbol{\theta}^*\|$. Since both $\|\boldsymbol{\theta}^{\langle t\rangle}\|$ and $w_1^{\langle t\rangle}$ is invariant to the coordinate rotations due to (42). WLOG, we assume that $\boldsymbol{\theta}^{\langle t\rangle} = (\|\boldsymbol{\theta}^{\langle t\rangle}\|, 0)$ and $\boldsymbol{\theta}^* = (\theta_\|^{\langle t\rangle}, \theta_\perp^{\langle t\rangle})$. Let us define the first coordinates of $\boldsymbol{\theta}^{\langle t+1\rangle}$ as $\tilde{\theta}_1^{\langle t+1\rangle}$, note that, we have

$$\begin{aligned}
\tilde{\theta}_1^{\langle t+1\rangle} &= \int y \frac{w_1^{\langle t\rangle} e^{y\|\boldsymbol{\theta}^{\langle t\rangle}\|} - w_2^{\langle t\rangle} e^{-y\|\boldsymbol{\theta}^{\langle t\rangle}\|}}{w_1^{\langle t\rangle} e^{y\|\boldsymbol{\theta}^{\langle t\rangle}\|} + w_2^{\langle t\rangle} e^{-y\|\boldsymbol{\theta}^{\langle t\rangle}\|}} \left(w_1^* \phi(y - \theta_\|^{\langle t\rangle}) + w_2^* \phi(y + \theta_\|^{\langle t\rangle})\right) \mathrm{d}y \\
&= G_\theta(\|\boldsymbol{\theta}^{\langle t\rangle}\|, w_1^{\langle t\rangle}; \theta_\|^{\langle t\rangle}, w_1^*) \\
w_1^{\langle t+1\rangle} &= \int \frac{w_1^{\langle t\rangle} e^{y\|\boldsymbol{\theta}^{\langle t\rangle}\|} - w_2^{\langle t\rangle} e^{-y\|\boldsymbol{\theta}^{\langle t\rangle}\|}}{w_1^{\langle t\rangle} e^{y\|\boldsymbol{\theta}^{\langle t\rangle}\|} + w_2^{\langle t\rangle} e^{-y\|\boldsymbol{\theta}^{\langle t\rangle}\|}} \left(w_1^* \phi(y - \theta_\|^{\langle t\rangle}) + w_2^* \phi(y + \theta_\|^{\langle t\rangle})\right) \mathrm{d}y \\
&= G_w(\|\boldsymbol{\theta}^{\langle t\rangle}\|, w_1^{\langle t\rangle}; \theta_\|^{\langle t\rangle}, w_1^*)
\end{aligned} \tag{44}$$

Hence, $(\tilde{\theta}_1^{\langle t+1\rangle}, w_1^{\langle t+1\rangle})$ is the next iteration of $(\|\boldsymbol{\theta}^{\langle t\rangle}\|, w_1^{\langle t\rangle})$ of the population-EM$_2$ under the true value $(\theta_\|^{\langle t\rangle}, w_1^*)$. Indeed, we can consider this two dimensional problem as a series of one dimensional problems that follows this procedure:

Step 1  Start with point $(\|\boldsymbol{\theta}^{\langle 1\rangle}\|, w_1^{\langle 1\rangle}) \in S$, where $S = (0, \infty) \times (0.5, 1)$.

Step 2  For iteration $t$, let point $(\|\boldsymbol{\theta}^{\langle t\rangle}\|, w_1^{\langle t\rangle})$ move towards the point $(\tilde{\theta}_1^{\langle t+1\rangle}, w_1^{\langle t+1\rangle})$ following the one dimensional update rule for the true value $\theta_\star = \theta_\|^{\langle t\rangle}$.

Step 3  Shift the true value $\theta_\star = \theta_\|^{\langle t\rangle}$ and the point $(\tilde{\theta}_1^{\langle t+1\rangle}, w_1^{\langle t+1\rangle})$ to the right to their new values: true value $\theta_\star = \theta_\|^{\langle t+1\rangle}$ and new point $(\|\boldsymbol{\theta}^{\langle t+1\rangle}\|, w_1^{\langle t+1\rangle})$.

Step 4  End iteration $t$ and go back to Step 2 for iteration $t+1$.

To analyze this, recall our analysis for the one dimension case in Section 2.3. Due to Lemma 3 holds for any non-zero true value $\theta^*$, by typical uniform continuity argument, we can find $\delta, \epsilon > 0$ such that the adjusted reference curve $r_{adj}(w_1; \theta_\star)$ defined by

$$r_{adj}(w_1; \theta_\star) \;=\; \frac{2w_1^* - 1}{2w_1 - 1}\theta_\star - \epsilon \cdot \max(0, w_1 + \delta - 1) > 0,$$

satisfies C.1,C.2 with $(a_\theta, b_\theta) = (0, \infty), (a_w, b_w) = (0.5, 1)$ for any true value $\theta_\star \in [\theta_\|^{\langle 1\rangle}, \|\boldsymbol{\theta}^*\|]$ and $w_\star = w_1^*$. Hence, on $S = (0, \infty) \times (0.5, 1)$, as $\theta_\star$ increases, the reference curve shifted to the right. Further, for any point $(\theta, w)$ in $S$, recall its corresponding area function $m(\theta, w)$ and rectangle $D(\theta, w)$ in the proof for Lemma 2 in Appendix B.2.1. We use $m(\theta, w; \theta_\star)$ and $D(\theta, w; \theta_\star)$ to denote their values under the true value $\theta_\star$. By their definitions, we note that the left side and down side of the rectangle $D(\theta, w; \theta_\star)$ is non-decreasing as $\theta_\star$ increases. Hence, by (26), we know as $\theta_\|^{\langle t\rangle}$ increases, $w_1^{\langle t\rangle}$ is always lower bounded by the down side of the rectangle $D(\|\boldsymbol{\theta}^{\langle 1\rangle}\|, w_1^{\langle 1\rangle}; \theta_\|^{\langle 1\rangle})$ due to the following chain of arguments:

$$w_1^{\langle t+1\rangle} \;\overset{(i)}{\geq}\; \text{lower side of } D(\|\boldsymbol{\theta}^{\langle t\rangle}\|, w_1^{\langle t\rangle}; \theta_\|^{\langle t\rangle}) \;\overset{(ii)}{\geq}\; \text{lower side of } D(\|\boldsymbol{\theta}^{\langle t\rangle}\|, w_1^{\langle t\rangle}; \theta_\|^{\langle t-1\rangle})$$

$$\overset{(iii)}{\geq}\; \text{lower side of } D(\tilde{\theta}_1^{\langle t\rangle}, w_1^{\langle t-1\rangle}; \theta_\|^{\langle t-1\rangle}) \;\overset{(iv)}{\geq}\; \text{lower side of } D(\|\boldsymbol{\theta}^{\langle t-1\rangle}\|, w_1^{\langle t-1\rangle}; \theta_\|^{\langle t-1\rangle})$$

$$\geq\; \cdots \geq \text{ lower side of } D(\|\boldsymbol{\theta}^{\langle 1\rangle}\|, w_1^{\langle 1\rangle}; \theta_\|^{\langle 1\rangle}) \;=\; c_3,$$

where inequality (i) holds due to (26), inequality (ii) and (iii) hold due to the shift of reference curve and definition of the rectangle $D$, and inequality (iv) holds due to (25). Also, we can show

$$\|\boldsymbol{\theta}^{\langle t\rangle}\| \geq \min\{\|\boldsymbol{\theta}^{\langle 1\rangle}\|, (w_1^* - w_2^*)\theta_\|^{\langle 1\rangle} - \epsilon\delta\} := c_1.$$

This is because,

- If $\|\boldsymbol{\theta}^{\langle t\rangle}\| \leq \theta_\|^{\langle t\rangle} - \epsilon\delta$, i.e., point $(\|\boldsymbol{\theta}^{\langle t\rangle}\|, w_1^{\langle t\rangle})$ is inside the region $R_5$ or $R_6$ defined by the true value $\theta_\star = \theta_\|^{\langle t\rangle}$, then we know $\|\boldsymbol{\theta}^{\langle t+1\rangle}\| \geq \tilde{\theta}_1^{\langle t+1\rangle} \geq \|\boldsymbol{\theta}^{\langle t\rangle}\|$.

- If $\|\boldsymbol{\theta}^{\langle t\rangle}\| \leq \theta_\|^{\langle t\rangle} - \epsilon\delta$, i.e., point $(\|\boldsymbol{\theta}^{\langle t\rangle}\|, w_1^{\langle t\rangle})$ is inside the regions $R_1$-$R_4$ (note that regions $R_7$ and $R_8$ doesn't exists here), we have $(\tilde{\theta}^{\langle t\rangle}, w_1^{\langle t+1\rangle})$ stay at $R_1$-$R_4$ and hence $\|\boldsymbol{\theta}^{\langle t+1\rangle}\| \geq \tilde{\theta}_1^{\langle t+1\rangle} \geq \theta_\|^{\langle t\rangle} - \epsilon\delta$.

Hence, this completes the proof of our claim that the angle $\beta^{\langle t\rangle}$ is decreasing to 0. Finally, we want to show that $(\|\boldsymbol{\theta}^{\langle t\rangle}\|, w_1^{\langle t\rangle})$ converges to $(\|\boldsymbol{\theta}^*\|, w_1^*)$ which implies $(\boldsymbol{\theta}^{\langle t\rangle}, w_1^{\langle t\rangle})$ converges to $(\boldsymbol{\theta}^*, w_1^*)$ due to $\beta^{\langle t\rangle} \to 0$. To prove this final step, we just need to bound $w_1^{\langle t\rangle}$ away from 1, i.e., there exists $c_4 \in (0, 1)$ such that

$$w_1^{\langle t\rangle} \;\leq\; c_4 \;<\; 1, \quad \forall t \geq 1. \tag{45}$$

Note that if (45) holds. Consider the following functions

$$
\begin{aligned}
U_1 &= m(\tilde{\theta}_1^{\langle t+1 \rangle}, w_1^{\langle t+1 \rangle}; \theta_\parallel^{\langle t \rangle}) / m(\|\boldsymbol{\theta}^{\langle t \rangle}\|, w_1^{\langle t \rangle}; \theta_\parallel^{\langle t \rangle}) \\
U_2 &= m(\|\boldsymbol{\theta}^{\langle t+1 \rangle}\|, w_1^{\langle t+1 \rangle}; \|\boldsymbol{\theta}^*\|) / m(\tilde{\theta}_1^{\langle t+1 \rangle}, w_1^{\langle t \rangle}; \theta_\parallel^{\langle t \rangle}) \\
U_3 &= m(\|\boldsymbol{\theta}^{\langle t \rangle}\|, w_1^{\langle t \rangle}; \theta_\parallel^{\langle t \rangle}) / m(\|\boldsymbol{\theta}^{\langle t \rangle}\|, w_1^{\langle t \rangle}; \|\boldsymbol{\theta}^*\|).
\end{aligned}
$$

For any $\delta_0 > 0$, we have after finite iterations $t_1$, $\theta_\parallel^{\langle t_1 \rangle}$ will stay in the $\delta_0$-neighborhood around $\|\boldsymbol{\theta}^*\|$. Hence, consider $t > t_1$, note that on the following compact set $S'$:

$$
\begin{aligned}
S' := \quad &\left\{ w^{\langle t \rangle} \in [c_3, c_4], \|\boldsymbol{\theta}^{\langle t \rangle}\| \in [c_1, c_2], \theta_\parallel^{\langle t \rangle} \in [\|\boldsymbol{\theta}^*\| - \delta_0, \|\boldsymbol{\theta}^*\|] \right\} \\
&- \left\{ (\|\boldsymbol{\theta}^{\langle t \rangle}\| - \|\boldsymbol{\theta}^*\|)^2 + (w^{\langle t \rangle} - w_1^*)^2 < 4\delta_0^2 \right\}.
\end{aligned}
\tag{46}
$$

we have $U_1 < 1$, therefore, we can find constant $\rho_1 < 1$ such that $U_1 \leq \rho_1$ on $S'$. Further, we know there exists a constant $c'$ such that $\max(U_2, U_3) \leq (1 + c \cdot \beta^{\langle t \rangle})$ on this compact set $S'$ since $\theta_\parallel^{\langle t \rangle} = \cos \beta^{\langle t \rangle} \cdot \|\boldsymbol{\theta}^*\|$ and $\tilde{\theta}^{\langle t \rangle} = \cos \beta^{\langle t \rangle} \cdot \|\boldsymbol{\theta}^{\langle t \rangle}\|$. Hence for large enough $t_2$, there exists $\rho_2 < 1$ such that for any $t > t_2$ and point $(\|\boldsymbol{\theta}^{\langle t \rangle}\|, w_1^{\langle t \rangle})$ in $S'$, we have

$$
\frac{m(\|\boldsymbol{\theta}^{\langle t+1 \rangle}\|, w_1^{\langle t+1 \rangle}; \|\boldsymbol{\theta}^*\|)}{m(\|\boldsymbol{\theta}^{\langle t \rangle}\|, w_1^{\langle t \rangle}; \|\boldsymbol{\theta}^*\|)} = U_1 \cdot U_2 \cdot U_3 \leq \rho_2 < 1.
$$

Hence, we have either $m(\|\boldsymbol{\theta}^{\langle t+1 \rangle}\|, w_1^{\langle t+1 \rangle}; \|\boldsymbol{\theta}^*\|)$ is strictly decreasing at rate $\rho_2$ or $(\|\boldsymbol{\theta}^{\langle t \rangle}\|, w_1^{\langle t \rangle})$ was in the $2\delta_0$-neighborhood around $(\|\boldsymbol{\theta}\|^*, w_1^*)$ and therefore by the analysis in Lemma 2, there exists constant $c'' > 0$ and $c''' > 0$ such that

$$
m(\|\boldsymbol{\theta}^{\langle t+1 \rangle}\|, w_1^{\langle t+1 \rangle}; \|\boldsymbol{\theta}^*\|) < (1 + c'' \cdot \beta^{\langle t \rangle}) \cdot c''' \delta_0^2.
$$

Either way, by arbitrary choice of $\delta_0$, we know $m(\|\boldsymbol{\theta}^{\langle t+1 \rangle}\|, w_1^{\langle t+1 \rangle}; \|\boldsymbol{\theta}^*\|)$ converges to 0 which implies $\boldsymbol{\theta}^{\langle t \rangle}$ converges to $\boldsymbol{\theta}^*$. Hence, finally, we just need to bound $w_1^{\langle t \rangle}$. Note that in the proof of Lemma 2, we used the following strategy to show that $w_1^{\langle t \rangle}$ is bounded away from 1:

- If $(\theta^{\langle 0 \rangle}, w_1^{\langle 0 \rangle}) \in R_5 \bigcup R_6$, within finite iterations $t_0$, $(\theta^{\langle t_0 \rangle}, w_1^{\langle t_0 \rangle})$ will reach the region $R_1 \bigcup R_2 \bigcup R_3 \bigcup R_4$.
- When $(\theta^{\langle t_0 \rangle}, w_1^{\langle t_0 \rangle}) \in R_1 \bigcup R_2 \bigcup R_3 \bigcup R_4$, by (25) and (26), we have for all $t \geq t_0$,

$$
(\theta^{\langle t+1 \rangle}, w_1^{\langle t+1 \rangle}) \in D(\theta^{\langle t+1 \rangle}, w_1^{\langle t+1 \rangle}) \overset{(a)}{\subseteq} D(\theta^{\langle t \rangle}, w_1^{\langle t \rangle}) \subseteq \cdots \subseteq D(\theta^{\langle t_0 \rangle}, w_1^{\langle t_0 \rangle}).
\tag{47}
$$

  Hence, $w^{\langle t \rangle} \leq \max(w_1^{\langle t_0 \rangle}, r^{-1}(\theta^{\langle t_0 \rangle}))$.

However, in multi-dimsnional case, since we changed the true values $\theta_\star$ from $\theta_\parallel^{\langle t \rangle}$ to $\theta_\parallel^{\langle t+1 \rangle}$ after each iteration, definition of $R_5$ and $R_6$ changes and relation (a) in (47) does not hold anymore, namely,

$$
D(\tilde{\theta}_1^{\langle t+1 \rangle}, w_1^{\langle t+1 \rangle}; \theta_\parallel^{\langle t+1 \rangle}) \not\subset D(\|\boldsymbol{\theta}^{\langle t \rangle}\|, w_1^{\langle t \rangle}; \theta_\parallel^{\langle t \rangle}).
$$

Yet, we can have a quick remedy for this strategy. Note that since $\theta_\parallel^{\langle t \rangle} \to \|\boldsymbol{\theta}^*\|$, our adjusted reference curve $r_{adj}(w_1; \theta_\parallel^{\langle t \rangle})$ also converges to $r_{adj}(w_1; \|\boldsymbol{\theta}^*\|)$ uniformly for $w_1 \in [w_1^*, 1]$. Hence, we can find $\delta' > 0$, $t' > 0$ such that we can perturb every $r_{adj}(w_1; \theta_\parallel^{\langle t \rangle})$ for $t > t'$ such that we have $\tilde{r}_{adj}(w_1; \theta_\parallel^{\langle t \rangle})$ satisfies C.1 and C.2 for true value $\theta_\star = \theta_\parallel^{\langle t \rangle}$ for all $t > t'$ with

$$
\tilde{r}_{adj}(w_1; \theta_\star) = r_{adj}(w_1; \theta_\parallel^{\langle t' \rangle}), \quad \forall w_1 \in [1 - \delta', 1], \theta_\star \in [\theta_\parallel^{\langle t' \rangle}, \|\boldsymbol{\theta}^*\|],
$$

and

$$
\tilde{r}_{adj}(w_1; \theta_\star) = r(w_1; \theta_\star), \quad \forall w_1 \leq w_1^*, \theta_\star \in [\theta_\parallel^{\langle t' \rangle}, \|\boldsymbol{\theta}^*\|].
$$

Hence, the region $R_5$ and $R_6$ are invariant for $\theta_\star \in [\theta_\|^{\langle t'\rangle}, \|\boldsymbol{\theta}^*\|]$, and therefore with the same arguments made in the proof of Lemma 2, within finite iterations $t''$, we have

$$\|\boldsymbol{\theta}^{\langle t''\rangle}\| > \theta_\|^{\langle t'\rangle}(w_1^* - w_2^*),$$

in other words, $(\|\boldsymbol{\theta}^{\langle t''\rangle}\|, w_1^{\langle t''\rangle})$ lies in $R_1 \bigcup R_2 \bigcup R_3 \bigcup R_4$ for any true value $\theta_\star \in [\theta_\|^{\langle t'\rangle}, \|\boldsymbol{\theta}\|^*]$. Once the point $(\|\boldsymbol{\theta}^{\langle t''\rangle}\|, w_1^{\langle t''\rangle})$ lies in the region $R_1 \bigcup R_2 \bigcup R_3 \bigcup R_4$, we can bound every $(\|\boldsymbol{\theta}^{\langle t+1\rangle}\|, w_1^{\langle t+1\rangle})$ for all $t \geq t''$ by

$$D\left(\min\left(\tilde{r}_{adj}(1-\delta'), \|\boldsymbol{\theta}^{\langle t\rangle}\|\right), \min\left(c_3, r^{-1}(c_2; \theta_\|^{\langle t\rangle})\right); \|\boldsymbol{\theta}^*\|\right) \bigcup D\left(c_2, \max(w_1^{\langle t\rangle}, 1-\delta'); \theta_\|^{\langle t\rangle}\right), \text{ (48)}$$

due to the fact that $(\tilde{\theta}_1^{\langle t+1\rangle}, w_1^{\langle t+1\rangle}) \in D(\|\boldsymbol{\theta}^{\langle t\rangle}\|, w_1^{\langle t\rangle}; \theta_\|^{\langle t\rangle})$ and $\|\boldsymbol{\theta}^{\langle t+1\rangle}\| \leq c_2$. Denote the set defined in (48) as $Q(\|\boldsymbol{\theta}^{\langle t\rangle}\|, w_1^{\langle t\rangle})$. Then, we can check that for any $(\theta, w_1) \in Q(\|\boldsymbol{\theta}^{\langle t\rangle}\|, w_1^{\langle t\rangle})$, we have $Q(\theta, w_1) \subseteq Q(\|\boldsymbol{\theta}^{\langle t\rangle}\|, w_1^{\langle t\rangle})$. Therefore, we have $Q(\|\boldsymbol{\theta}^{\langle t+1\rangle}\|, w_1^{\langle t+1\rangle}) \subseteq Q(\|\boldsymbol{\theta}^{\langle t\rangle}\|, w^{\langle t\rangle})$. Hence, by a chain of arguments starting from $t''$, we have

$$(\|\boldsymbol{\theta}^{\langle t+1\rangle}\|, w_1^{\langle t+1\rangle}) \in Q((\|\boldsymbol{\theta}^{\langle t''\rangle}\|, w_1^{\langle t''\rangle})).$$

Hence, we have

$$w_1^{\langle t\rangle} \leq \max\left(\tilde{r}_{adj}^{-1}(\|\boldsymbol{\theta}^{\langle t''\rangle}\|; \|\boldsymbol{\theta}^*\|), 1-\delta', w_1^{\langle t''\rangle}\right) < 1, \quad \forall t \geq t''.$$

## B.5 Geometric convergence

Since we have shown that $(\boldsymbol{\theta}^{\langle t\rangle}, w^{\langle t\rangle})$ converges to $(\boldsymbol{\theta}^*, w_1^*)$, we just need to show an attraction basin around $(\boldsymbol{\theta}^*, w_1^*)$, and therefore, combining both, we know after a finite iteration $T$, we have geometric convergence. To show an attraction basin, let us consider the following two terms $\|\boldsymbol{\theta}^{\langle t+1\rangle} - \boldsymbol{\theta}^*\|$ and $|w_1^{\langle t+1\rangle} - w_1^*|$. Note that, at iteration $t$, let us choose the coordinate such that $\boldsymbol{\theta}^{\langle t\rangle} = (\|\boldsymbol{\theta}^{\langle t\rangle}\|, 0, \ldots, 0)$ and $\boldsymbol{\theta}^* = (\theta_\|^{\langle t\rangle}, \theta_\perp^{\langle t\rangle}, 0, \ldots, 0)$, then by (44) and (20), we have

$$\|\boldsymbol{\theta}^{\langle t+1\rangle} - \boldsymbol{\theta}^*\|^2 = |\tilde{\theta}_1^{\langle t+1\rangle} - \theta_\|^{\langle t\rangle}|^2 + |\tilde{\theta}_2^{\langle t+1\rangle} - \theta_\perp^{\langle t\rangle}|^2$$

$$= |G_\theta(\|\boldsymbol{\theta}^{\langle t\rangle}\|, w_1^{\langle t\rangle}; \theta_\|^{\langle t\rangle}, w_1^*) - \theta_\|^{\langle t\rangle}|^2 + |\theta_\perp^{\langle t\rangle}|^2(1 - s(\|\boldsymbol{\theta}^{\langle t\rangle}\|, w_1^{\langle t\rangle}; \theta_\|^{\langle t\rangle}, w_1^*))^2,$$

$$|w_1^{\langle t+1\rangle} - w_1^*| = |G_w(\|\boldsymbol{\theta}^{\langle t\rangle}\|, w_1^{\langle t\rangle}; \theta_\|^{\langle t\rangle}, w_1^*) - w_1^*|. \tag{49}$$

Hence, we just need to show that for all $\theta_\|^* > 0$ and $w_1^* \in (0, 1)$, the eigenvalues of the Jacobian matrix of the following mapping:

$$(\theta, w_1) \mapsto (G_\theta(\theta, w_1; \theta_\|^*, w_1^*), G_w(\theta, w_1; \theta_\|^*, w_1^*)) \tag{50}$$

are in $[0, 1)$ at $(\theta, w_1) = (\theta_\|^*, w_1^*)$. Then, note that

$$G_\theta(\theta_\|^*, w_1^*; \theta_\|^*, w_1^*) = \theta_\|^* \quad \text{and} \quad G_w(\theta_\|^*, w_1^*; \theta_\|^*, w_1^*) = w_1^*.$$

Hence, by continuity of the Jacobian of the functions, there exists $\epsilon > 0$ and $\rho < 1$ such that as long as $\theta, \theta_\|^* \in [\|\boldsymbol{\theta}^*\| - \epsilon, \|\boldsymbol{\theta}^*\| + \epsilon]$ and $w_1 \in [w_1^* - \epsilon, w_1^* + \epsilon]$, we have

$$(G_\theta(\theta, w_1; \theta_\|^*, w_1^*) - \theta_\|^*)^2 + (G_w(\theta, w_1; \theta_\|^*, w_1^*) - w_1^*)^2 \leq \rho\left((\theta - \theta_\|^*)^2 + (w_1 - w_1^*)^2\right).$$

Further, by (22), we know function $s(\theta, w_1; \theta_\|^*, w_1^*)$ is positive on $\theta, \theta_\|^* \in [\|\boldsymbol{\theta}^*\| - \epsilon, \|\boldsymbol{\theta}^*\| + \epsilon]$ and $w_1 \in [w_1^* - \epsilon, w_1^* + \epsilon]$. Hence, there exists constant $\rho'$ such that

$$(1 - s(\theta, w_1; \theta_\|^*, w_1^*))^2 \leq \rho', \quad \forall \theta, \theta_\|^* \in [\|\boldsymbol{\theta}^*\| - \epsilon, \|\boldsymbol{\theta}^*\| + \epsilon], w_1 \in [w_1^* - \epsilon, w_1^* + \epsilon].$$

Hence, plug in (49), we have if $\|\boldsymbol{\theta}^{\langle t\rangle}\|, \theta_\|^{\langle t\rangle} \in [\|\boldsymbol{\theta}^*\| - \epsilon, \|\boldsymbol{\theta}^*\| + \epsilon]$ and $w_1^{\langle t\rangle} \in [w_1^* - \epsilon, w_1^* + \epsilon]$, then

$$\|\boldsymbol{\theta}^{\langle t+1\rangle} - \boldsymbol{\theta}^*\|^2 + |w_1^{\langle t+1\rangle} - w_1^*|^2 \leq \rho\left((\|\boldsymbol{\theta}^{\langle t\rangle}\| - \theta_\|^{\langle t\rangle})^2 + (w_1^{\langle t\rangle} - w_1^*)^2\right) + \rho'|\theta_\perp^{\langle t\rangle}|^2$$

$$\leq \max(\rho, \rho')\left(\|\boldsymbol{\theta}^{\langle t\rangle} - \boldsymbol{\theta}^*\|^2 + (w_1^{\langle t\rangle} - w_1^*)^2\right).$$

Hence, by triangle inequality, we know once $\|\boldsymbol{\theta}^{\langle t \rangle} - \boldsymbol{\theta}^*\| \leq \epsilon$ and $|w_1^{\langle t \rangle} - w_1^*| \leq \epsilon$, we have $(\boldsymbol{\theta}^{\langle t \rangle}, w_1^{\langle t \rangle})$ geometrically converges towards $(\boldsymbol{\theta}^*, w_1^*)$. Further, the first iteration to reach the attraction basin is guaranteed by the geometric convergence of the angle $\beta^{\langle t \rangle}$ and geometric convergence of the area function $m(\theta, w)$ on $S'$ defined in (46) for $\delta_0 = \epsilon/4$.

Next, we will show that for all $\theta_{\|}^* > 0$ and $w_1^* \in (0, 1)$, the eigenvalues of the Jacobian matrix of the mapping defined in (50) at $(\theta, w_1) = (\theta_{\|}^*, w_1^*)$ are in $[0, 1)$. Note that this Jacobian matrix at $(\theta, w_1) = (\theta_{\|}^*, w_1^*)$ is the following:

$$
J \;=\; \left[
\begin{array}{cc}
\underbrace{\int \dfrac{4 w_1^* w_2^* y^2}{w_1^* e^{y\theta_{\|}^*} + w_2^* e^{-y\theta_{\|}^*}} \phi(y) e^{-\frac{(\theta_{\|}^*)^2}{2}}\, \mathrm{d}y}_{J_{11}} & \underbrace{\int \dfrac{2y}{w_1^* e^{y\theta_{\|}^*} + w_2^* e^{-y\theta_{\|}^*}} \phi(y) e^{-\frac{(\theta_{\|}^*)^2}{2}}\, \mathrm{d}y}_{J_{12}} \\[3em]
\underbrace{\int \dfrac{2 w_1^* w_2^* y}{w_1^* e^{y\theta_{\|}^*} + w_2^* e^{-y\theta_{\|}^*}} \phi(y) e^{-\frac{(\theta_{\|}^*)^2}{2}}\, \mathrm{d}y}_{J_{21}} & \underbrace{\int \dfrac{1}{w_1^* e^{y\theta_{\|}^*} + w_2^* e^{-y\theta_{\|}^*}} \phi(y) e^{-\frac{(\theta_{\|}^*)^2}{2}}\, \mathrm{d}y}_{J_{22}}
\end{array}
\right].
$$

Then the two eigenvalues of $J$ should be the two solutions of the following equation:

$$
q(\lambda) \;=\; \lambda^2 - \lambda(J_{11} + J_{22}) + J_{11}J_{22} - J_{12}J_{21} \;=\; 0.
$$

Note that, by Cauchy inequality, we know $\det(J) = J_{11}J_{22} - J_{12}J_{21} \geq 0$ and therefore $q(0) \geq 0$. Also note that

$$
q(J_{22}) \;=\; -J_{22}^2 - J_{12}J_{21} \;\leq\; 0,
$$

and

$$
\begin{aligned}
0 \;<\; J_{22} \;&=\; \int_{y \geq 0} \frac{e^{y\theta_{\|}^*} + e^{-y\theta_{\|}^*}}{w_1^* w_2^* (e^{y\theta_{\|}^*} - e^{-y\theta_{\|}^*})^2 + 1} \phi(y) e^{-\frac{(\theta_{\|}^*)^2}{2}}\, \mathrm{d}y \\
&=\; \int_{y \geq 0} (e^{y\theta_{\|}^*} + e^{-y\theta_{\|}^*}) \phi(y) e^{-\frac{(\theta_{\|}^*)^2}{2}}\, \mathrm{d}y \\
&\qquad - \int_{y \geq 0} \frac{w_1^* w_2^* (e^{y\theta_{\|}^*} + e^{-y\theta_{\|}^*})(e^{y\theta_{\|}^*} - e^{-y\theta_{\|}^*})^2}{w_1^* w_2^* (e^{y\theta_{\|}^*} - e^{-y\theta_{\|}^*})^2 + 1} \phi(y) e^{-\frac{(\theta_{\|}^*)^2}{2}}\, \mathrm{d}y \\
&=\; 1 - \int_{y \geq 0} \frac{w_1^* w_2^* (e^{y\theta_{\|}^*} + e^{-y\theta_{\|}^*})(e^{y\theta_{\|}^*} - e^{-y\theta_{\|}^*})^2}{w_1^* w_2^* (e^{y\theta_{\|}^*} - e^{-y\theta_{\|}^*})^2 + 1} \phi(y) e^{-\frac{(\theta_{\|}^*)^2}{2}}\, \mathrm{d}y \\
&\leq\; 1. \qquad\qquad\qquad\qquad\qquad\qquad\qquad\qquad\qquad\qquad\qquad\quad (51)
\end{aligned}
$$

Hence, we just need to show $q(1) > 0$, then the two solutions of $q(\lambda) = 0$ should stay in $[0, 1)$. Note that

$$
\begin{aligned}
J_{11} \;&=\; \int_{y \geq 0} \frac{4 w_1^* w_2^* (e^{y\theta_{\|}^*} + e^{-y\theta_{\|}^*}) y^2}{w_1^* w_2^* (e^{y\theta_{\|}^*} - e^{-y\theta_{\|}^*})^2 + 1} \phi(y) e^{-\frac{(\theta_{\|}^*)^2}{2}}\, \mathrm{d}y \\
&=\; \int_{y \geq 0} \frac{4 y^2}{e^{y\theta_{\|}^*} + e^{-y\theta_{\|}^*}} \phi(y) e^{-\frac{(\theta_{\|}^*)^2}{2}}\, \mathrm{d}y \\
&\qquad - \int_{y \geq 0} \frac{4 (w_1^* - w_2^*)^2 y^2}{(e^{y\theta_{\|}^*} + e^{-y\theta_{\|}^*})(w_1^* w_2^* (e^{y\theta_{\|}^*} - e^{-y\theta_{\|}^*})^2 + 1)} \phi(y) e^{-\frac{(\theta_{\|}^*)^2}{2}}\, \mathrm{d}y \\
&<\; 1 - \int_{y \geq 0} \frac{4 (w_1^* - w_2^*)^2 y^2}{(e^{y\theta_{\|}^*} + e^{-y\theta_{\|}^*})(w_1^* w_2^* (e^{y\theta_{\|}^*} - e^{-y\theta_{\|}^*})^2 + 1)} \phi(y) e^{-\frac{(\theta_{\|}^*)^2}{2}}\, \mathrm{d}y, \qquad (52)
\end{aligned}
$$

where the last inequality holds due to the fact that

$$
\int_{y \geq 0} \frac{4 y^2}{e^{y\theta_{\|}^*} + e^{-y\theta_{\|}^*}} \phi(y) e^{-\frac{(\theta_{\|}^*)^2}{2}}\, \mathrm{d}y \;\leq\; \int_{y \geq 0} 2 y^2 \phi(y) e^{-\frac{(\theta_{\|}^*)^2}{2}}\, \mathrm{d}y \;=\; e^{-\frac{(\theta_{\|}^*)^2}{2}}.
$$

Combine (51) and (52), we have

$$
\begin{aligned}
q(1) \;=\;& (1 - J_{11})(1 - J_{22}) - J_{12}J_{21} \\
>\;& \int_{y\geq 0} \frac{4(w_1^* - w_2^*)^2 y^2}{(e^{y\theta_{\parallel}^*} + e^{-y\theta_{\parallel}^*})(w_1^* w_2^*(e^{y\theta_{\parallel}^*} - e^{-y\theta_{\parallel}^*})^2 + 1)} \phi(y) e^{-\frac{(\theta_{\parallel}^*)^2}{2}} \, \mathrm{d}y \\
&\times \int_{y\geq 0} \frac{w_1^* w_2^*(e^{y\theta_{\parallel}^*} + e^{-y\theta_{\parallel}^*})(e^{y\theta_{\parallel}^*} - e^{-y\theta_{\parallel}^*})^2}{w_1^* w_2^*(e^{y\theta_{\parallel}^*} - e^{-y\theta_{\parallel}^*})^2 + 1} \phi(y) e^{-\frac{(\theta_{\parallel}^*)^2}{2}} \, \mathrm{d}y \\
& -4w_1^* w_2^*(w_1^* - w_2^*)^2 \int_{y\geq 0} \left( \frac{(e^{y\theta_{\parallel}^*} - e^{-y\theta_{\parallel}^*})y}{w_1^* w_2^*(e^{y\theta_{\parallel}^*} - e^{-y\theta_{\parallel}^*})^2 + 1} \phi(y) e^{-\frac{(\theta_{\parallel}^*)^2}{2}} \, \mathrm{d}y \right)^2 \\
\geq\;& 0,
\end{aligned}
$$

where the last inequality holds due to Cauchy inequality. Hence, we have $q(1) > 0$ and this completes our proof for geometric convergence of the EM estimates.

## C   Proof of Theorem 3

The maximum log-likelihood objective for population-$\mathrm{EM}_2$ is the following optimization problem:

$$
\max_{\boldsymbol{\theta}\in\mathbb{R}^d, w_1\in[0,1]} \mathbb{E}_{\boldsymbol{y}\sim f^*} \log\left( w_1 e^{-\frac{\|\boldsymbol{y}-\boldsymbol{\theta}\|^2}{2}} + w_2 e^{-\frac{\|\boldsymbol{y}+\boldsymbol{\theta}\|^2}{2}} \right). \tag{53}
$$

Due to the symmetric property of the landscape, without loss of generality, we assume $w_1^* > 0.5$. Note that the first order stationary points of above optimization problem should satisfy the following equation.

$$
\mathbb{E}_{\boldsymbol{y}\sim f^*}\left[ \frac{w_1 e^{\langle \boldsymbol{y},\boldsymbol{\theta}\rangle} - w_2 e^{-\langle \boldsymbol{y},\boldsymbol{\theta}\rangle}}{w_1 e^{\langle \boldsymbol{y},\boldsymbol{\theta}\rangle} + w_2 e^{-\langle \boldsymbol{y},\boldsymbol{\theta}\rangle}} \boldsymbol{y} \right] - \boldsymbol{\theta} \;=\; \boldsymbol{0}, \tag{54}
$$

$$
\mathbb{E}_{\boldsymbol{y}\sim f^*}\left[ \frac{e^{\langle \boldsymbol{y},\boldsymbol{\theta}\rangle} - e^{-\langle \boldsymbol{y},\boldsymbol{\theta}\rangle}}{w_1 e^{\langle \boldsymbol{y},\boldsymbol{\theta}\rangle} + w_2 e^{-\langle \boldsymbol{y},\boldsymbol{\theta}\rangle}} \right] \;=\; 0. \tag{55}
$$

We first consider the two trivial cases when $w_1 = 1$ and $w_1 = 0$. Suppose $w_1 = 1$, then from (54), we have $\boldsymbol{\theta} = (w_1^* - w_2^*)\boldsymbol{\theta}^*$. Hence, plug it in (55), we have the following equation holds

$$
\int \left( 1 - e^{-2(w_1^* - w_2^*)y\|\boldsymbol{\theta}^*\|} \right) \left( w_1^* \phi(y - \|\boldsymbol{\theta}^*\|) + w_2^* \phi(y + \|\boldsymbol{\theta}^*\|) \right) \mathrm{d}y \;=\; 0,
$$

which is equivalent to

$$
1 - w_1^* e^{-4w_2^*(w_1^* - w_2^*)\|\boldsymbol{\theta}^*\|^2} - w_2^* e^{4w_1^*(w_1^* - w_2^*)\|\boldsymbol{\theta}^*\|^2} \;=\; 0.
$$

Taking the derivative with respect to $\|\boldsymbol{\theta}^*\|$, it is straightforward to show that when $w_1^* > 0.5$, the LHS is a strictly decreasing function of $\|\boldsymbol{\theta}^*\|$ and achieves its maximum 0 at $\|\boldsymbol{\theta}^*\| = 0$. Hence, it contradicts the RHS of the equation and therefore (54) and (55) can not hold simultaneously for $w_1 = 1$. Hence, there is no first order stationary point for the case $w_1 = 1$ and similarly for $w_1 = 0$.

Now we restrict $w_1 \in (0,1)$. Then it is straightforward to show that every first order stationary point of the optimization in (53) should be a fixed point for population-$\mathrm{EM}_2$. From the proof of Theorem 2, we know the two global maxima $(\boldsymbol{\theta}^*, w_1)$ and $(-\boldsymbol{\theta}^*, w_2)$ are the only fixed points of population-$\mathrm{EM}_2$ in the following region:

$$
\underbrace{\left\{ (\boldsymbol{\theta}, w_1) \mid w_1 \in [0.5, 1), \langle \boldsymbol{\theta}, \boldsymbol{\theta}^*\rangle > 0 \right\}}_{\text{Area}_1} \bigcup \underbrace{\left\{ (\boldsymbol{\theta}, w_1) \mid w_1 \in (0, 0.5], \langle \boldsymbol{\theta}, \boldsymbol{\theta}^*\rangle < 0 \right\}}_{\text{Area}_2}
$$

Furthermore, for any fixed point lies in the hyperplane $\mathcal{H}: \langle \boldsymbol{\theta}, \boldsymbol{\theta}^*\rangle = 0$, it is clear that its corresponding $w_1$ should be 0.5. Further, since $\langle \boldsymbol{\theta}, \boldsymbol{\theta}^*\rangle = 0$, from (54), it is clear that $\boldsymbol{\theta}$ should satisfy the following equation

$$
\int \frac{e^{y\|\boldsymbol{\theta}\|} - e^{-y\|\boldsymbol{\theta}\|}}{e^{y\|\boldsymbol{\theta}\|} + e^{-y\|\boldsymbol{\theta}\|}} y\phi(y) \, \mathrm{d}y \;=\; \|\boldsymbol{\theta}\|.
$$

Since the derivative with respect to $\|\boldsymbol{\theta}\|$ of the LHS is in $(0,1)$ for $\|\boldsymbol{\theta}\| > 0$, it is clear that $\|\boldsymbol{\theta}\| = 0$ is the only solution for the equation and therefore, $(\boldsymbol{\theta}, w_1) = (\mathbf{0}, \frac{1}{2})$ is the only fixed point in the hyperplane $\mathcal{H}$. Furthermore, the Hessian of the log-likelihood in (53) at $(\boldsymbol{\theta}, w_1) = (\mathbf{0}, \frac{1}{2})$ is the following matrix.

$$
\begin{bmatrix}
\boldsymbol{\theta}^*(\boldsymbol{\theta}^*)^\top & 2(w_1^* - w_2^*)\boldsymbol{\theta}^* \\
2(w_1^* - w_2^*)(\boldsymbol{\theta}^*)^\top & 0
\end{bmatrix}
\tag{56}
$$

It is clear that it has a positive eigenvalue, a negative eigenvalue and therefore $(\mathbf{0}, \frac{1}{2})$ is a saddle point.

Finally, we will show there is no fixed point in the rest of the region in $\mathbb{R}^2 \times [0,1]$, i.e.,

$$
\underbrace{\left\{(\boldsymbol{\theta}, w_1)|w_1 \in (0,0.5), \langle \boldsymbol{\theta}, \boldsymbol{\theta}^* \rangle > 0\right\}}_{\text{Area}_3} \bigcup \underbrace{\left\{(\boldsymbol{\theta}, w_1)|w_1 \in (0.5,1), \langle \boldsymbol{\theta}, \boldsymbol{\theta}^* \rangle < 0\right\}}_{\text{Area}_4}
$$

Due to the symmetric property, we will just prove the result for $\text{Area}_3$. Note that, by Lemma 3 and the fact that

$$
g_w(\theta, 0.5) \lessgtr 0.5, \quad \forall \theta \lessgtr 0.
\tag{57}
$$

We know for all $w_1 \in (0, 0.5)$,

$$
\begin{aligned}
0 \quad &< \quad g_w(\|\boldsymbol{\theta}\|, w_1; \theta_\|, w_1^*) - w_1 \\
&= \quad w_1 w_2 \int \left[ \frac{e^{y\|\boldsymbol{\theta}\|} - e^{-y\|\boldsymbol{\theta}\|}}{w_1 e^{y\|\boldsymbol{\theta}\|} + w_2 e^{-y\|\boldsymbol{\theta}\|}} \right] \left( w_1^* \phi(y - \theta_\|) + w_2^* \phi(y + \theta_\|) \right) \mathrm{d}y \\
&= \quad w_1 w_2 \cdot \mathbb{E}_{\boldsymbol{y} \sim f^*} \left[ \frac{e^{\langle \boldsymbol{y}, \boldsymbol{\theta} \rangle} - e^{-\langle \boldsymbol{y}, \boldsymbol{\theta} \rangle}}{w_1 e^{\langle \boldsymbol{y}, \boldsymbol{\theta} \rangle} + w_2 e^{-\langle \boldsymbol{y}, \boldsymbol{\theta} \rangle}} \right],
\end{aligned}
$$

where $\theta_\| = \langle \boldsymbol{\theta}^*, \boldsymbol{\theta} \rangle / \|\boldsymbol{\theta}\|$. Hence, there is no solution for (55) in $\text{Area}_3$. This completes the proof of this theorem.

## D  Proof of Theorem 4

Let $(\hat{\boldsymbol{\theta}}^{\langle t \rangle}, \hat{w}_1^{\langle t \rangle})$ denote the finite sample estimate. To show the convergence of the finite sample estimate, we want to argue that its behavior is close to the corresponding convergence behavior of the population estimate. Hence, let us first prove the following uniform concentration bounds that for any fixed constant $c > 0$, with probability at least $1 - \delta$, we have

$$
\Delta_w := \sup_{\|\boldsymbol{\theta}\| \in [0,c], w_1 \in [0,1]} \left| \frac{1}{n} \sum_{i=1}^n \left[ \frac{w_1 e^{\langle \boldsymbol{y}_i, \boldsymbol{\theta} \rangle}}{w_1 e^{\langle \boldsymbol{y}_i, \boldsymbol{\theta} \rangle} + w_2 e^{-\langle \boldsymbol{y}_i, \boldsymbol{\theta} \rangle}} \right] - \mathbb{E}_{\boldsymbol{y} \sim f^*} \left[ \frac{w_1 e^{\langle \boldsymbol{y}, \boldsymbol{\theta} \rangle}}{w_1 e^{\langle \boldsymbol{y}, \boldsymbol{\theta} \rangle} + w_2 e^{-\langle \boldsymbol{y}, \boldsymbol{\theta} \rangle}} \right] \right|
$$

$$
\leq O\left( (\|\boldsymbol{\theta}^*\| + 1)\sqrt{\frac{d + \ln(2/\delta)}{n}} \right)
\tag{58}
$$

$$
\Delta_\theta := \sup_{\|\boldsymbol{\theta}\| \in [0,c], w_1 \in [0,1]} \left\| \frac{1}{n} \sum_{i=1}^n \left[ \frac{w_1 e^{\langle \boldsymbol{y}_i, \boldsymbol{\theta} \rangle} - w_2 e^{-\langle \boldsymbol{y}_i, \boldsymbol{\theta} \rangle}}{w_1 e^{\langle \boldsymbol{y}_i, \boldsymbol{\theta} \rangle} + w_2 e^{-\langle \boldsymbol{y}_i, \boldsymbol{\theta} \rangle}} \boldsymbol{y}_i \right] - \mathbb{E}_{\boldsymbol{y} \sim f^*} \left[ \frac{w_1 e^{\langle \boldsymbol{y}, \boldsymbol{\theta} \rangle} - w_2 e^{-\langle \boldsymbol{y}, \boldsymbol{\theta} \rangle}}{w_1 e^{\langle \boldsymbol{y}, \boldsymbol{\theta} \rangle} + w_2 e^{-\langle \boldsymbol{y}, \boldsymbol{\theta} \rangle}} \boldsymbol{y} \right] \right\|
$$

$$
\leq O\left( (\|\boldsymbol{\theta}^*\| + 1)\sqrt{\frac{d + \ln(2/\delta)}{n}} \right).
\tag{59}
$$

To show (58), by Jensen's inequality, we have

$$
\mathbb{E}e^{\lambda \Delta_w} \quad \leq \quad \mathbb{E}_{y,y'} \exp\left( \lambda \sup_{\|\boldsymbol{\theta}\| \in [0,c], w_1 \in [0,1]} \left| \frac{1}{n} \sum_{i=1}^n \left( \frac{w_1 e^{\langle \boldsymbol{y}_i, \boldsymbol{\theta} \rangle}}{w_1 e^{\langle \boldsymbol{y}_i, \boldsymbol{\theta} \rangle} + w_2 e^{-\langle \boldsymbol{y}_i, \boldsymbol{\theta} \rangle}} - \frac{w_1 e^{\langle \boldsymbol{y}_i', \boldsymbol{\theta} \rangle}}{w_1 e^{\langle \boldsymbol{y}_i', \boldsymbol{\theta} \rangle} + w_2 e^{-\langle \boldsymbol{y}_i', \boldsymbol{\theta} \rangle}} \right) \right| \right).
$$

Then, we introduce i.i.d. Rademacher variables $\xi_i$ and obtain that

$$\mathbb{E}e^{\lambda\Delta_w} \leq \mathbb{E}_{y,\xi}\exp\left(2\lambda \sup_{\|\boldsymbol{\theta}\|\in[0,c],w_1\in[0,1]}\left|\frac{1}{n}\sum_{i=1}^{n}\xi_i\left(\frac{w_1 e^{\langle\boldsymbol{y}_i,\boldsymbol{\theta}\rangle}}{w_1 e^{\langle\boldsymbol{y}_i,\boldsymbol{\theta}\rangle}+w_2 e^{-\langle\boldsymbol{y}_i,\boldsymbol{\theta}\rangle}}-w_1\right)\right|\right).$$

Now apply the following lemma from Koltchinskii [2011]

**Lemma 8.** *Let $\mathcal{H}\in\mathbb{R}^n$ and let $\psi_i:\mathbb{R}\mapsto\mathbb{R}, i=1,\cdots,n$ be functions such that $\psi_i(0)=0$ and*

$$|\psi_i(u)-\psi_i(v)| \leq |u-v|\in\mathbb{R}.$$

*For all convex nondecreasing functions $\Psi:\mathbb{R}_+\mapsto\mathbb{R}_+$,*

$$\mathbb{E}\Psi(\frac{1}{2}\sup_{\boldsymbol{h}\in\mathcal{H}}|\sum_{i=1}^{n}\psi_i(h_i)\xi_i|) \leq \mathbb{E}\Psi(\sup_{\boldsymbol{h}\in\mathcal{H}}|\sum_{i=1}^{n}h_i\xi_i|),$$

*where $\xi_i$ are i.i.d. Rademacher random variables.*

We have

$$\begin{aligned}
\mathbb{E}e^{\lambda\Delta_w} &\leq \mathbb{E}_{y,\xi}\exp\left(2\lambda \sup_{\|\boldsymbol{\theta}\|\in[0,c],w_1\in[0,1]}\left|\frac{1}{n}\sum_{i=1}^{n}\xi_i\langle\boldsymbol{y}_i,\boldsymbol{\theta}\rangle\right|\right)\\
&\leq \mathbb{E}_{y,\xi}\exp\left(2\lambda c\left\|\frac{1}{n}\sum_{i=1}^{n}\xi_i\boldsymbol{y}_i\right\|\right)\\
&= \mathbb{E}_{\tilde{y}}\exp\left(2\lambda c\left\|\frac{1}{n}\sum_{i=1}^{n}\tilde{\boldsymbol{y}}_i\right\|\right),
\end{aligned}$$

where $\tilde{\boldsymbol{y}}_i$ are i.i.d. random variables following this symmetric distribution: $\frac{1}{2}\mathcal{N}(-\boldsymbol{\theta}^*,\boldsymbol{I})+\frac{1}{2}\mathcal{N}(\boldsymbol{\theta}^*,\boldsymbol{I})$. Then apply a typical argument of $1/2$-covering net over the $d$-dimensional unit sphere, it is straight forward to show that we have

$$\mathbb{E}e^{\lambda\Delta_w} \leq \exp\left(8\lambda^2 c^2\frac{\|\boldsymbol{\theta}^*\|^2+1}{n}+2d\right).$$

Apply Markov inequality and choose $\lambda$ properly, we have (58) holds. To prove (59), we follow the proof of corollary 2 in B.2 in Balakrishnan et al. [2017]. Let

$$\Delta_\theta^{\boldsymbol{u}} = \sup_{\|\boldsymbol{\theta}\|\in[0,c],w_1\in[0,1]}\frac{1}{n}\sum_{i=1}^{n}\left[\frac{w_1 e^{\langle\boldsymbol{y}_i,\boldsymbol{\theta}\rangle}-w_2 e^{-\langle\boldsymbol{y}_i,\boldsymbol{\theta}\rangle}}{w_1 e^{\langle\boldsymbol{y}_i,\boldsymbol{\theta}\rangle}+w_2 e^{-\langle\boldsymbol{y}_i,\boldsymbol{\theta}\rangle}}\right]\langle\boldsymbol{y}_i,\boldsymbol{u}\rangle-\mathbb{E}_{\boldsymbol{y}\sim f^*}\left[\frac{w_1 e^{\langle\boldsymbol{y},\boldsymbol{\theta}\rangle}-w_2 e^{-\langle\boldsymbol{y},\boldsymbol{\theta}\rangle}}{w_1 e^{\langle\boldsymbol{y},\boldsymbol{\theta}\rangle}+w_2 e^{-\langle\boldsymbol{y},\boldsymbol{\theta}\rangle}}\right]\langle\boldsymbol{y},\boldsymbol{u}\rangle.$$

Then, we have

$$\begin{aligned}
\mathbb{E}e^{\lambda\Delta_\theta} &= \mathbb{E}_y e^{\lambda\sup_{\|\boldsymbol{u}\|=1}\Delta_\theta^{\boldsymbol{u}}} \leq \mathbb{E}_y e^{2\lambda\max_{j\in[M]}\Delta_\theta^{\boldsymbol{u}_j}} \leq \sum_{j=1}^{M}\mathbb{E}_y e^{2\lambda\Delta_\theta^{\boldsymbol{u}_j}}\\
&\leq \sum_{j=1}^{M}\mathbb{E}_{y,\xi}\exp\left(4\lambda \sup_{\|\boldsymbol{\theta}\|\in[0,c],w_1\in[0,1]}\frac{1}{n}\sum_{i=1}^{n}\xi_i\left[\frac{w_1 e^{\langle\boldsymbol{y}_i,\boldsymbol{\theta}\rangle}-w_2 e^{-\langle\boldsymbol{y}_i,\boldsymbol{\theta}\rangle}}{w_1 e^{\langle\boldsymbol{y}_i,\boldsymbol{\theta}\rangle}+w_2 e^{-\langle\boldsymbol{y}_i,\boldsymbol{\theta}\rangle}}-(w_1-w_2)\right]\langle\boldsymbol{y}_i,\boldsymbol{u}_j\rangle\right),
\end{aligned}$$

where $\{\boldsymbol{u}_j\}_{j=1}^{M}$ is the $\frac{1}{2}$-covering net over the $d$ dimensional unit sphere and $\xi_i$ are i.i.d. Rademacher random variables and the last inequality holds for standard symmetrization result for empirical process. Apply Lemma 8 again, we have

$$\begin{aligned}
\mathbb{E}e^{\lambda\Delta_\theta} &\leq \sum_{j=1}^{M}\mathbb{E}_{y,\xi}\exp\left(4\lambda \sup_{\|\boldsymbol{\theta}\|\in[0,c],w_1\in[0,1]}\frac{1}{n}\sum_{i=1}^{n}\xi_i\langle\boldsymbol{y}_i,\boldsymbol{\theta}\rangle\langle\boldsymbol{y}_i,\boldsymbol{u}_j\rangle\right)\\
&\leq e^{2d}\cdot\mathbb{E}_{y,\xi}\exp\left(4\lambda c\left\|\frac{1}{n}\sum_{i=1}^{n}\xi_i\boldsymbol{y}_i\boldsymbol{y}_i^\top\right\|_{\mathrm{op}}\right),
\end{aligned}$$

where $\| \cdot \|_{\text{op}}$ is the $\ell_2$-operator norm of a matrix (the maximum singular value). Follow the result in B.2 in Balakrishnan et al. [2017], we have

$$
\begin{aligned}
\mathbb{E}_{y,\xi}\exp\left(4\lambda c\left\|\frac{1}{n}\sum_{i=1}^{n}\xi_i \boldsymbol{y}_i \boldsymbol{y}_i^\top\right\|_{\text{op}}\right) &\leq \sum_{j=1}^{M}\mathbb{E}_{y,\xi}\exp\left(8\lambda c\frac{1}{n}\sum_{i=1}^{n}\xi_i\langle \boldsymbol{y}_i, \boldsymbol{u}_j\rangle^2\right)\\
&= \sum_{j=1}^{M}\mathbb{E}_{y,\xi,\xi'}\exp\left(8\lambda c\frac{1}{n}\sum_{i=1}^{n}\xi_i\langle \xi_i'\boldsymbol{y}_i, \boldsymbol{u}_j\rangle^2\right)\\
&= \sum_{j=1}^{M}\mathbb{E}_{\tilde{y},\xi}\exp\left(8\lambda c\frac{1}{n}\sum_{i=1}^{n}\xi_i\langle \tilde{\boldsymbol{y}}_i, \boldsymbol{u}_j\rangle^2\right),
\end{aligned}
$$

where $\xi_i'$ are independent copies of Rademacher random variables. Hence, from Balakrishnan et al. [2017], we have

$$
\sum_{j=1}^{M}\mathbb{E}_{\tilde{y},\xi}\exp\left(8\lambda c\frac{1}{n}\sum_{i=1}^{n}\xi_i\langle \tilde{\boldsymbol{y}}_i, \boldsymbol{u}_j\rangle^2\right) \leq e^{\frac{32\lambda^2 c^2(\|\boldsymbol{\theta}^*\|^2+1)}{n}+2d}.
$$

Hence, combine all, we have

$$
\mathbb{E}e^{\lambda\Delta_\theta} \leq e^{\frac{32\lambda^2 c^2(\|\boldsymbol{\theta}^*\|^2+1)}{n}+4d}.
$$

Apply Markov inequality and choose $\lambda$ properly, we have (59) holds.

Next, by choosing $c = \max(\|\hat{\boldsymbol{\theta}}^{\langle 0\rangle}\|, 2(1+\|\boldsymbol{\theta}^*\|))$, it is straight forward to apply induction with Lemma 6 to show that for sufficiently large $n$, with probability at least $1-\delta$,

$$
\|\hat{\boldsymbol{\theta}}^{\langle t\rangle}\| \leq c, \quad \forall t \geq 0.
$$

Then, since the update functions are Lipchitz with constant at most $O(1+\|\boldsymbol{\theta}^*\|)$, it is straight forward to show the following via induction that for any finite $t$,

$$
\|\hat{\boldsymbol{\theta}}^{\langle t\rangle} - \boldsymbol{\theta}^{\langle t\rangle}\|^2 + |\hat{w}_1^{\langle t\rangle} - w_1^{\langle t\rangle}|^2 \leq O\left((1+\|\boldsymbol{\theta}^*\|)^{t+1}\sqrt{\frac{d+\ln(2/\delta)}{n}}\right).
$$

From Appendix B.5, we know there exists an attraction basin around $(\boldsymbol{\theta}^*, w_1^*)$. Suppose this attraction basin contains the $\delta_0$-neighborhood around $(\boldsymbol{\theta}^*, w_1^*)$, i.e., we have for some $\rho < 1$,

$$
\|\boldsymbol{\theta}^{\langle t+1\rangle} - \boldsymbol{\theta}^*\|^2 + |w_1^{\langle t+1\rangle} - w_1^*|^2 \leq \rho\left(\|\boldsymbol{\theta}^{\langle t\rangle} - \boldsymbol{\theta}^*\|^2 + (w_1^{\langle t\rangle} - w_1^*)^2\right), \quad \forall \|\boldsymbol{\theta}^{\langle t\rangle} - \boldsymbol{\theta}^*\|^2 + |w_1^{\langle t\rangle} - w_1^*|^2 \leq \delta_0^2
$$

Hence, from the proof in Appendix B, we know there exists a finite iteration $T$ such that

$$
\|\boldsymbol{\theta}^{\langle T\rangle} - \boldsymbol{\theta}^*\|^2 + |w_1^{\langle T\rangle} - w_1^*|^2 \leq \frac{\delta_0^2}{2},
$$

and therefore, for large enough $n$, with probability at least $1-\delta$, we have the finite sample estimate lies in the attraction basin after $T$ iteration, i.e.,

$$
\|\hat{\boldsymbol{\theta}}^{\langle T\rangle} - \boldsymbol{\theta}^*\|^2 + |\hat{w}_1^{\langle T\rangle} - w_1^*|^2 \leq \delta_0^2.
$$

Once the finite sample estimate lies in the attraction basin, we follow the proof in Balakrishnan et al. [2017] and it is straight forward to show that for all $t \geq T$, we have

$$
\|\hat{\boldsymbol{\theta}}^{\langle t\rangle} - \boldsymbol{\theta}^*\|^2 + |\hat{w}_1^{\langle t\rangle} - w_1^*|^2 \leq \rho^{t-T}\left(\|\hat{\boldsymbol{\theta}}^{\langle T\rangle} - \boldsymbol{\theta}^*\|^2 + |\hat{w}_1^{\langle T\rangle} - w_1^*|^2\right) + O\left((\|\boldsymbol{\theta}^*\|+1)\sqrt{\frac{d+\ln(2/\delta)}{n}}\right).
$$

This completes our analysis for the convergence of the finite sample estimate.

# E  Proof of Auxiliary Lemmas

## E.1  Proof of Lemma 4

In this proof, we have $w_1 = w_1^*$. To prove (15), we just need to show

$$\frac{\partial h(\theta, w_1)}{\partial w_1} \begin{cases} > 0, & w_1 > 0.5 \\ < 0, & w_1 < 0.5 \end{cases} \quad \forall \theta < \theta^*. \tag{60}$$

To prove this, we divide it into two cases (i) $\theta \leq 0$ and (ii) $\theta \in (0, \theta^*)$. To prove (i), by the definition of $h(\theta, w_1)$ in (5) (with $w_2 = 1 - w_1$), we have

$$\frac{\partial h(\theta, w_1)}{\partial w_1} = \underbrace{\int \frac{w_1 e^{y\theta} - w_2 e^{-y\theta}}{w_1 e^{y\theta} + w_2 e^{-y\theta}} y(\phi(y - \theta^*) - \phi(y + \theta^*)) \, dy}_{\text{part 1}}$$

$$+ 2 \underbrace{\int \frac{w_1 e^{y\theta^*} + w_2 e^{-y\theta^*}}{(w_1 e^{y\theta} + w_2 e^{-y\theta})^2} y\phi(y) e^{-\frac{(\theta^*)^2}{2}} \, dy}_{\text{part 2}}.$$

For part 1, we have

$$\text{part 1} = \int_{y \geq 0} \left\{ \frac{w_1 e^{y\theta} - w_2 e^{-y\theta}}{w_1 e^{y\theta} + w_2 e^{-y\theta}} + \frac{w_1 e^{-y\theta} - w_2 e^{y\theta}}{w_1 e^{-y\theta} + w_2 e^{y\theta}} \right\} y(\phi(y - \theta^*) - \phi(y + \theta^*)) \, dy$$

$$= 2 \int_{y \geq 0} \frac{w_1^2 - w_2^2}{w_1^2 + w_2^2 + w_1 w_2 (e^{-y\theta} + e^{y\theta})} y(\phi(y - \theta^*) - \phi(y + \theta^*)) \, dy$$

Hence, we have

$$\text{part 1} \begin{cases} > 0, & w_1 > 0.5 \\ < 0, & w_1 < 0.5 \end{cases}. \tag{61}$$

For part 2, we have

$$\text{part 2} = \int_{y \geq 0} \left\{ \frac{w_1 e^{y\theta^*} + w_2 e^{-y\theta^*}}{(w_1 e^{y\theta} + w_2 e^{-y\theta})^2} - \frac{w_1 e^{-y\theta^*} + w_2 e^{y\theta^*}}{(w_1 e^{-y\theta} + w_2 e^{y\theta})^2} \right\} y\phi(y) e^{-\frac{(\theta^*)^2}{2}} \, dy$$

$$= (w_1 - w_2) \int_{y \geq 0} \left\{ \frac{(w_1^2 + w_2^2 + w_1 w_2)(e^{y(\theta^* - 2\theta)} - e^{y(2\theta - \theta^*)}) + 2w_1 w_2 (e^{y\theta^*} - e^{-y\theta^*})}{(w_1 e^{y\theta} + w_2 e^{-y\theta})^2 (w_1 e^{-y\theta} + w_2 e^{y\theta})^2} \right.$$

$$\left. + \frac{w_1 w_2 (e^{-y(\theta^* + 2\theta)} - e^{y(\theta^* + 2\theta)})}{(w_1 e^{y\theta} + w_2 e^{-y\theta})^2 (w_1 e^{-y\theta} + w_2 e^{y\theta})^2} \right\} y\phi(y) e^{-\frac{(\theta^*)^2}{2}} \, dy.$$

Since $\theta \leq 0$, we have

$$e^{y(\theta^* - 2\theta)} - e^{y(2\theta - \theta^*)} \geq \max \left\{ \left| e^{y\theta^*} - e^{-y\theta^*} \right|, \left| e^{y(\theta^* + 2\theta)} - e^{-y(\theta^* + 2\theta)} \right| \right\}.$$

Hence, we have

$$\frac{\text{part 2}}{w_1 - w_2} \geq \int_{y \geq 0} \frac{(w_1 - w_2)^2 (e^{y(\theta^* - 2\theta)} - e^{y(2\theta - \theta^*)})}{(w_1 e^{y\theta} + w_2 e^{-y\theta})^2 (w_1 e^{-y\theta} + w_2 e^{y\theta})^2} y\phi(y) e^{-\frac{(\theta^*)^2}{2}} \, dy \geq 0.$$

Therefore, we have

$$\text{part 2} \begin{cases} \geq 0, & w_1 > 0.5 \\ \leq 0, & w_1 < 0.5 \end{cases}. \tag{62}$$

Combine (61) and (62), we have (60) holds for case (i). To prove case (ii), we use a different strategy. First note that $h(\theta^*, w) \equiv \theta^*$, hence,

$$\left. \frac{\partial h(\theta, w)}{\partial w} \right|_{\theta = \theta^*} = 0. \tag{63}$$

Therefore, to prove (60) for case (ii), we just need to show

$$\frac{\partial^2 h(\theta, w_1)}{\partial \theta \partial w_1} \begin{cases} < 0, & w_1 > 0.5 \\ > 0, & w_1 < 0.5 \end{cases} \quad \forall \theta \in (0, \theta^*). \tag{64}$$

By the definition of $h(\theta, w_1)$ in (5) (with $w_2 = 1 - w_1$), we have

$$\frac{1}{4}\frac{\partial^2 h(\theta, w_1)}{\partial \theta \partial w_1} = \underbrace{2w_1 w_2 \int \frac{e^{y(\theta^* - \theta)} - e^{y(\theta - \theta^*)}}{(w_1 e^{y\theta} + w_2 e^{-y\theta})^3} y^2 \phi(y) e^{-\frac{(\theta^*)^2}{2}} \, \mathrm{d}y}_{\text{part 3}}$$

$$+ \underbrace{\int \left( -\frac{w_1^2 e^{y\theta^*}}{(w_1 e^{y\theta} + w_2 e^{-y\theta})^2} + \frac{w_2^2 e^{-y\theta^*}}{(w_1 e^{y\theta} + w_2 e^{-y\theta})^2} \right) y^2 \phi(y) e^{-\frac{(\theta^*)^2}{2}} \, \mathrm{d}y}_{\text{part 4}}$$

For part 3, we have

$$\text{part 3} = 2w_1 w_2 \int_{y \geq 0} \frac{(w_1 - w_2)(e^{y(\theta^* - \theta)} - e^{y(\theta - \theta^*)})(e^{-y\theta} - e^{y\theta})(A^2 + B^2 - AB)}{(w_1 e^{y\theta} + w_2 e^{-y\theta})^3 (w_1 e^{-y\theta} + w_2 e^{y\theta})^3} y^2 \phi(y) e^{-\frac{(\theta^*)^2}{2}} \, \mathrm{d}y,$$

where $A = w_1 e^{y\theta} + w_2 e^{-y\theta}$ and $B = w_1 e^{-y\theta} + w_2 e^{y\theta}$. Hence, since $\theta \in (0, \theta^*)$, we have

$$\text{part 3} \begin{cases} < 0, & w_1 > 0.5 \\ > 0, & w_1 < 0.5 \end{cases}. \tag{65}$$

For part 4, we have

$$\text{part 4} = -\int_{y \geq 0} (w_1 - w_2) \frac{(w_1^2 + w_2^2)(e^{(2\theta - \theta^*)y} + e^{-(2\theta - \theta^*)y}) + 2w_1 w_2 (e^{y\theta^*} + e^{-y\theta^*})}{(w_1 e^{y\theta} + w_2 e^{-y\theta})^2 (w_1 e^{-y\theta} + w_2 e^{y\theta})^2} y^2 \phi(y) e^{-\frac{(\theta^*)^2}{2}} \, \mathrm{d}y.$$

Hence, we have

$$\text{part 4} \begin{cases} < 0, & w_1 > 0.5 \\ > 0, & w_1 < 0.5 \end{cases}. \tag{66}$$

Combine (65) and (66), we have (64) holds and therefore (60) holds for case (ii). This completes the proof for (15). To prove (16), note that

$$0 \leq \frac{\partial H(\theta, w_1)}{\partial \theta} = \int \frac{4w_1 w_2}{(w_1 e^{y\theta} + w_2 e^{-y\theta})^2} y^2 (w_1 \phi(y - \theta^*) + w_2 \phi(y + \theta^*)) \, \mathrm{d}y$$

$$= \underbrace{\int_{y \geq 0} \frac{4w_1 w_2}{(w_1 e^{y\theta} + w_2 e^{-y\theta})^2} y^2 (w_1 \phi(y - \theta^*) + w_2 \phi(y + \theta^*)) \, \mathrm{d}y}_{\text{part 5}}$$

$$+ \underbrace{\int_{y \geq 0} \frac{4w_1 w_2}{(w_2 e^{y\theta} + w_1 e^{-y\theta})^2} y^2 (w_2 \phi(y - \theta^*) + w_1 \phi(y + \theta^*)) \, \mathrm{d}y}_{\text{part 6}}.$$

Since part 5 and part 6 are symmetric with respect to $w_1, w_2$, WLOG, we assume $w_1 \geq 0.5$. Then for part 5, note that since $\theta \geq \theta^*$, we have $w_1 e^{y\theta^*} + w_2 e^{-y\theta^*} \leq w_1 e^{y\theta} + w_2 e^{-y\theta}$, and therefore,

$$\text{part 5} \leq \int_{y \geq 0} \frac{4w_1 w_2}{(w_1 e^{y\theta^*} + w_2 e^{-y\theta^*})^2} y^2 (w_1 \phi(y - \theta^*) + w_2 \phi(y + \theta^*)) \, \mathrm{d}y$$

$$= \int_{y \geq 0} \frac{4w_1 w_2}{w_1 e^{y\theta^*} + w_2 e^{-y\theta^*}} y^2 \phi(y) e^{-\frac{(\theta^*)^2}{2}} \, \mathrm{d}y$$

$$\leq \int_{y \geq 0} 2\sqrt{w_1 w_2} y^2 \phi(y) e^{-\frac{(\theta^*)^2}{2}} \, \mathrm{d}y \leq \frac{e^{-\frac{(\theta^*)^2}{2}}}{2}, \tag{67}$$

where last two inequalities hold due to AM-GM inequality. For part 6, we have if $\theta \geq \theta^*$,

$$\text{part 6} = \int_{y \geq 0} \frac{4}{\left(\sqrt{\frac{w_1}{w_2}}e^{-y\theta} + \sqrt{\frac{w_2}{w_1}}e^{y\theta}\right)^2} y^2(w_1\phi(y+\theta^*) + w_2\phi(y-\theta^*))\,\mathrm{d}y$$

$$\overset{(a)}{\leq} \int_{y \geq 0} \frac{2}{e^{(y-\frac{\ln(w_1/w_2)}{2\theta})\theta} + e^{-(y-\frac{\ln(w_1/w_2)}{2\theta})\theta}} y^2(w_1\phi(y+\theta^*) + w_2\phi(y-\theta^*))\,\mathrm{d}y$$

$$\overset{(b)}{\leq} \int_{y \geq 0} \frac{2}{e^{(y-\frac{\ln(w_1/w_2)}{2\theta})\theta^*} + e^{-(y-\frac{\ln(w_1/w_2)}{2\theta})\theta^*}} y^2(w_1\phi(y+\theta^*) + w_2\phi(y-\theta^*))\,\mathrm{d}y$$

$$= \int_{y \geq 0} \frac{2}{(\frac{w_2}{w_1})^{\frac{\theta^*}{2\theta}}e^{y\theta^*} + (\frac{w_1}{w_2})^{\frac{\theta^*}{2\theta}}e^{-y\theta^*}} y^2(w_1\phi(y+\theta^*) + w_2\phi(y-\theta^*))\,\mathrm{d}y, \qquad (68)$$

where inequality (a) holds due to AM-GM inequality, and inequality (b) holds due to the monotonic of hyperbolic cosine function. Our next step is to prove for all $y\theta^* \geq 0$ and $0 < \theta^* \leq \theta$, we have

$$(\frac{w_2}{w_1})^{\frac{\theta^*}{2\theta}}e^{y\theta^*} + (\frac{w_1}{w_2})^{\frac{\theta^*}{2\theta}}e^{-y\theta^*} \geq 2(w_1 e^{-y\theta^*} + w_2 e^{y\theta^*}), \qquad (69)$$

which, with (68), immediately implies that

$$\text{part 6} \leq \int_{y \geq 0} y^2 \phi(y) e^{-\frac{(\theta^*)^2}{2}}\,\mathrm{d}y = \frac{e^{-\frac{(\theta^*)^2}{2}}}{2},$$

and therefore, combine with (67), we have (16) holds. To prove (69), note that this is equivalent to prove

$$\left(\frac{w_2}{w_1}\right)^{\frac{\theta^*}{2\theta}}\left(1 - 2w_1^{\frac{\theta^*}{2\theta}}w_2^{1-\frac{\theta^*}{2\theta}}\right)e^{y\theta^*} \geq \left(\frac{w_1}{w_2}\right)^{\frac{\theta^*}{2\theta}}\left(2w_2^{\frac{\theta^*}{2\theta}}w_1^{1-\frac{\theta^*}{2\theta}} - 1\right)e^{-y\theta^*}. \qquad (70)$$

Note that

$$w_1^{\frac{\theta^*}{2\theta}}w_2^{1-\frac{\theta^*}{2\theta}} + w_1^{1-\frac{\theta^*}{2\theta}}w_2^{\frac{\theta^*}{2\theta}} = (w_1 w_2)^{\frac{\theta^*}{2\theta}}(w_1^{1-\frac{\theta^*}{\theta}} + w_2^{1-\frac{\theta^*}{\theta}})$$

$$\leq \frac{w_1^{1-\frac{\theta^*}{\theta}} + w_2^{1-\frac{\theta^*}{\theta}}}{2^{\frac{\theta^*}{\theta}}}$$

$$\leq (w_1 + w_2)^{1-\frac{\theta^*}{\theta}} = 1.$$

where the last two inequalities holds due to AM-GM inequality and Holder inequality respectively. Also, since $w_1 \geq w_2$, we have

$$w_1^{\frac{\theta^*}{2\theta}}w_2^{1-\frac{\theta^*}{2\theta}} \leq w_1^{1-\frac{\theta^*}{2\theta}}w_2^{\frac{\theta^*}{2\theta}}.$$

Hence, we have

$$1 - 2w^{\frac{\theta^*}{2\theta}}w_2^{1-\frac{\theta^*}{2\theta}} \geq 0.$$

Therefore, to prove (70), it is sufficient to prove

$$(\frac{w_2}{w_1})^{\frac{\theta^*}{2\theta}}(1 - 2w_1^{\frac{\theta^*}{2\theta}}w_2^{1-\frac{\theta^*}{2\theta}}) \geq (\frac{w_1}{w_2})^{\frac{\theta^*}{2\theta}}(2w_2^{\frac{\theta^*}{2\theta}}w_1^{1-\frac{\theta^*}{2\theta}} - 1),$$

which is equivalent to

$$(\frac{w_2}{w_1})^{\frac{\theta^*}{2\theta}} + (\frac{w_1}{w_2})^{\frac{\theta^*}{2\theta}} \geq 2(w_1 + w_2) = 2,$$

which holds due to AM-GM inequality. Hence, we have (69) holds.

## E.2 Proof of Lemma 5

We first analyze the condition that can determine the sign of $g(\theta, w_1) - w_1$. Note that (with $w_2 = 1 - w_1$)

$$\frac{g(\theta, w_1) - w_1}{w_1} = \int \frac{1}{\sqrt{2\pi}} e^{-\frac{y^2 + (\theta^*)^2}{2}} \cdot \left( \frac{e^{y\theta}\left(w_1^* e^{y\theta^*} + w_2^* e^{-y\theta^*}\right)}{w_1 e^{y\theta} + w_2 e^{-y\theta}} - e^{y\theta^*} \right) dy$$

$$= \int_{y \geq 0} \frac{1}{\sqrt{2\pi}} e^{-\frac{y^2 + (\theta^*)^2}{2}} \cdot \left( \frac{e^{y\theta}\left(w_1^* e^{y\theta^*} + w_2^* e^{-y\theta^*}\right)}{w_1 e^{y\theta} + w_2 e^{-y\theta}} - e^{y\theta^*} + \frac{e^{-y\theta}\left(w_1^* e^{-y\theta^*} + w_2^* e^{y\theta^*}\right)}{w_1 e^{-y\theta} + w_2 e^{y\theta}} - e^{-y\theta^*} \right) dy$$

Hence, to determine the sign of $g(\theta, w_1) - w_1 \gtreqless 0$, we just need to show $\forall y \geq 0$

$$\left( \frac{e^{y\theta}\left(w_1^* e^{y\theta^*} + w_2^* e^{-y\theta^*}\right)}{w_1 e^{y\theta} + w_2 e^{-y\theta}} - e^{y\theta^*} \right) + \left( \frac{e^{-y\theta}\left(w_1^* e^{-y\theta^*} + w_2^* e^{y\theta^*}\right)}{w_1 e^{-y\theta} + w_2 e^{y\theta}} - e^{-y\theta^*} \right) \gtreqless 0,$$

which is equivalent to

$$(2w_1 - 1)\cosh_y(\theta^*) + (w_1^* - w_1)\cosh_y(\theta^* + 2\theta) + (1 - w_1 - w_1^*)\cosh_y(\theta^* - 2\theta) \gtreqless 0, \quad (71)$$

where $\cosh_y(x) = (e^{yx} + e^{-yx})/2$. Let $\theta_\gamma = \gamma\theta^* = \frac{2w_1^* - 1}{2w_1 - 1}\theta^*$. Let us first show that for $w_1 \in (0.5, 1]$

$$g_w(\theta_\gamma, w_1^*) \gtreqless w_1^*, \quad \forall w_1 \gtreqless w_1^*. \quad (72)$$

By (71), we just need to show

$$\cosh_y(\theta^*) \gtreqless \cosh_y(\theta^* - 2\theta_\gamma), \quad \forall w_1 \gtreqless w_1^*,$$

which holds due to the monotonic of hyperbolic cosine function. Hence, we have proved (72). Next, we want to show

$$g_w(\theta_\gamma, w_1) \gtreqless w_1, \quad \forall w_1 \lesseqgtr w_1^*. \quad (73)$$

By (71), we just need to show that $\forall y > 0$,

$$(2w_1 - 1)\cosh_y(\theta^*) + (w_1^* - w_1)\cosh_y(\theta^* + 2\theta_\gamma) + (1 - w_1 - w_1^*)\cosh_y(\theta^* - 2\theta_\gamma) \gtreqless 0, \quad \forall w_1 \lesseqgtr w_1^*. \quad (74)$$

Note that, by Taylor expansion of $2\cosh_y(x) = \sum_{i=0}^\infty \frac{(xy)^{2i}}{(2i)!}$, we just need to show that given $\gamma = \frac{2w_1^* - 1}{2w_1 - 1}$, we have

$$(2w_1 - 1) + (w_1^* - w_1)(1 + 2\gamma)^{2k} + (1 - w_1^* - w_1)(2\gamma - 1)^{2k} > 0, \quad \forall w_1 \in (\frac{1}{2}, w_1^*), k > 0, \quad (75)$$

$$(w_1 - w_1^*)(1 + 2\gamma)^{2k} + (w_1^* + w_1 - 1)(2\gamma - 1)^{2k} - (2w_1 - 1) > 0, \quad \forall w_1 \in (w_1^*, 1], k > 1. \quad (76)$$

For (75), since $w_1 < w_1^*$, we have $\gamma > 1$ and

$$(2w_1 - 1) + (w_1^* - w_1)(1 + 2\gamma)^{2k} + (1 - w_1^* - w_1)(2\gamma - 1)^{2k}$$

$$= (w_1^* - w_1)\left((1 + 2\gamma)^{2k} - (2\gamma - 1)^{2k}\right) + (2w_1 - 1)\left(1 - (2\gamma - 1)^{2k}\right)$$

$$= (w_1^* - w_1) \cdot 2 \left( \sum_{i=0}^{2k-1} (1 + 2\gamma)^i (2\gamma - 1)^{2k-1-i} \right) + (2w_1 - 1) \cdot (2\gamma - 2) \left( \sum_{i=0}^{2k-1} (2\gamma - 1)^i \right)$$

$$= 2(w_1^* - w_1) \left( \sum_{i=0}^{2k-1} \left((1 + 2\gamma)^i - 2\right)(2\gamma - 1)^{2k-1-i} \right)$$

$$\geq 2(w_1^* - w_1) \left( \sum_{i=0}^{1} \left((1 + 2\gamma)^i - 2\right)(2\gamma - 1)^{2k-1-i} \right)$$

$$= 4(w_1^* - w_1)(\gamma - 1)(2\gamma - 1)^{2k-2} > 0.$$

For (76), we have

$$(w_1 - w_1^*)(2\gamma + 1)^{2k} + (w_1^* + w_1 - 1)(2\gamma - 1)^{2k} - (2w_1 - 1)$$

$$= (w_1 - w_1^*)\left((2\gamma + 1)^{2k} - (2\gamma - 1)^{2k}\right) + (2w_1 - 1)\left((2\gamma - 1)^{2k} - 1\right)$$

$$= (w_1 - w_1^*)\left((2\gamma + 1)^2 - (2\gamma - 1)^2\right)\left(\sum_{i=0}^{k-1}(2\gamma + 1)^{2i}(2\gamma - 1)^{2k-2i-2}\right)$$

$$+ (2w_1 - 1)\left((2\gamma - 1)^2 - 1\right)\left(\sum_{i=0}^{k-1}(2\gamma - 1)^{2i}\right)$$

$$= 8(w_1 - w_1^*)\gamma\left(\sum_{i=0}^{k-1}\left((2\gamma + 1)^{2i} - 1\right)(2\gamma - 1)^{2k-2i-2}\right) > 0.$$

Hence, this completes the proof for (73).

### E.3 Proof of Lemma 6

We just need to bound $\|G_\theta(\boldsymbol{\theta}, w_1; \boldsymbol{\theta}^*, w_1^*)\|^2$. Note that by (7) and Jensen's inequality, we have

$$\|G_\theta(\boldsymbol{\theta}, w_1; \boldsymbol{\theta}^*, w_1^*)\|^2 \leq \mathbb{E}_{\boldsymbol{y}}\left[\left(\frac{w_1 e^{\langle \boldsymbol{y}, \boldsymbol{\theta}\rangle} - w_2^{\langle t\rangle} e^{-\langle \boldsymbol{y}, \boldsymbol{\theta}\rangle}}{w_1 e^{\langle \boldsymbol{y}, \boldsymbol{\theta}\rangle} + w_2 e^{-\langle \boldsymbol{y}, \boldsymbol{\theta}\rangle}}\right)^2 \|\boldsymbol{y}\|^2\right]$$

$$\leq \mathbb{E}_{\boldsymbol{y}}\|\boldsymbol{y}\|^2 = 1 + \|\boldsymbol{\theta}^*\|^2.$$

### E.4 Proof of Lemma 7

To show (40), we first define $\theta_\gamma = \gamma\theta^*$, $\theta_b = b\theta^*$, and

$$A = \int y \frac{e^{y\theta_\gamma}}{w_1 e^{y\theta_\gamma} + (1 - w_1)e^{-y\theta_\gamma}}\left(w_1^*\phi(y - \theta^*) + w_2^*\phi(y + \theta^*)\right) \mathrm{d}y$$

$$B = \int y \frac{e^{-y\theta_\gamma}}{w_1 e^{y\theta_\gamma} + (1 - w_1)e^{-y\theta_\gamma}}\left(w_1^*\phi(y - \theta^*) + w_2^*\phi(y + \theta^*)\right) \mathrm{d}y.$$

Note that $\forall w_1$

$$(2w_1 - 1)\theta_\gamma \equiv w_1 A + w_2 B. \tag{77}$$

Hence, we have (40) is equivalent to show that

$$w_1 A - w_2 B < \frac{w_1 A + w_2 B}{2w_1 - 1}, \quad \forall w_1 \in (0.5, w_1^*),$$

which is equivalent to show

$$A + B > 0, \quad \forall w_1 \in (0.5, w_1^*). \tag{78}$$

Note that

$$A + B = \int \frac{1}{\sqrt{2\pi}} y(e^{y\theta_\gamma} + e^{-y\theta_\gamma})e^{-\frac{y^2 + (\theta^*)^2}{2}}\frac{w_1^* e^{y\theta^*} + w_2^* e^{-y\theta^*}}{w_1 e^{y\theta_\gamma} + w_2 e^{-y\theta_\gamma}} \mathrm{d}y$$

$$= \int_{y \geq 0} \frac{1}{\sqrt{2\pi}} y(e^{y\theta_\gamma} + e^{-y\theta_\gamma})e^{-\frac{y^2 + (\theta^*)^2}{2}}\left(\frac{w_1^* e^{y\theta^*} + w_2^* e^{-y\theta^*}}{w_1 e^{y\theta_\gamma} + w_2 e^{-y\theta_\gamma}} - \frac{w_1^* e^{-y\theta^*} + w_2^* e^{y\theta^*}}{w_1 e^{-y\theta_\gamma} + w_2 e^{y\theta_\gamma}}\right) \mathrm{d}y$$

$$= \int_{y \geq 0} \frac{1}{\sqrt{2\pi}} y(e^{y\theta_\gamma} + e^{-y\theta_\gamma})e^{-\frac{y^2 + (\theta^*)^2}{2}}$$

$$\times \frac{(w_1^* + w_1 - 1)\left(e^{y\theta^*(1-\gamma)} - e^{-y\theta^*(1-\gamma)}\right) + (w_1^* - w_1)\left(e^{y\theta^*(1+\gamma)} - e^{-y\theta^*(1+\gamma)}\right)}{\left(w_1 e^{y\theta_\gamma} + w_2 e^{-y\theta_\gamma}\right)\left(w_1 e^{-y\theta_\gamma} + w_2 e^{y\theta_\gamma}\right)} \mathrm{d}y.$$

Hence, we just need to show that for $\forall y > 0, w_1^*, w_1 \in (\frac{1}{2}, 1)$,

$$(w_1^* + w_1 - 1)\left(e^{y\theta^*(1-\gamma)} - e^{-y\theta^*(1-\gamma)}\right) + (w_1^* - w_1)\left(e^{y\theta^*(1+\gamma)} - e^{-y\theta^*(1+\gamma)}\right) \;>\; 0, \quad \forall w_1 \in (0.5, w_1^*)$$

By Taylor expansion of $e^x$, we just need to prove that for all $k \geq 0$, we have

$$(w_1^* + w_1 - 1)(1 - \gamma)^{2k+1} + (w_1^* - w_1)(1 + \gamma)^{2k+1} \;>\; 0, \quad \forall w_1 \in (0.5, w_1^*)$$

By definition of $\gamma$, we just need to show

$$(w_1^* + w_1 - 1)2^{2k+1}(w_1 - w_1^*)^{2k+1} + (w_1^* - w_1)2^{2k+1}(w_1^* + w_1 - 1)^{2k+1} \;>\; 0, \quad \forall w_1 \in (0.5, w_1^*)$$
$$\Leftrightarrow w_1 + w_1^* - 1 \;>\; w_1^* - w_1, \quad \forall w_1 \in (0.5, w_1^*),$$

which obviously holds. To show (41), we should analyze the condition for $g_\theta(\theta, w_1) - \theta > 0$. Note that

$$
\begin{aligned}
g_\theta(\theta_b, w_1) - \theta_b &= \int y \left( \frac{w_1 e^{y\theta_b} - w_2 e^{-y\theta_b}}{w_1 e^{y\theta_\gamma} + w_2 e^{-y\theta_\gamma}} - \frac{b}{w_1^* - w_2^*} \right) \left( w_1^* \phi(y - \theta^*) + w_2^* \phi(y + \theta^*) \right) \mathrm{d}y \\
&= \frac{1}{w_1^* - w_2^*} \int y \frac{w_1(2w_1^* - 1 - b)e^{y\theta_b} - w_2(2w_1^* - 1 + b)e^{-y\theta_b}}{w_1 e^{y\theta_\gamma} + w_2 e^{-y\theta_\gamma}} \left( w_1^* \phi(y - \theta^*) + w_2^* \phi(y + \theta^*) \right) \mathrm{d}y \\
&= \int_{y \geq 0} \frac{y}{\sqrt{2\pi}} e^{-\frac{y^2 + (\theta^*)^2}{2}} \left( \frac{w_1 w_2 \left( (1 - b) \cdot 2 \sinh_{y\theta^*}(2b + 1) + (1 + b) \cdot 2 \sinh_{y\theta^*}(2b - 1) \right)}{\left( w_1 e^{y\theta_\gamma} + w_2 e^{-y\theta_\gamma} \right) \left( w_1 e^{-y\theta_\gamma} + w_2 e^{y\theta_\gamma} \right)} \right. \\
&\quad \left. + \frac{\left( (2w_1 - 1)(2w_1^* - 1) - (1 - 2w_1 w_2) b \right) \cdot 2 \sinh_{y\theta^*}(1)}{\left( w_1 e^{y\theta_\gamma} + w_2 e^{-y\theta_\gamma} \right) \left( w_1 e^{-y\theta_\gamma} + w_2 e^{y\theta_\gamma} \right)} \right) \mathrm{d}y,
\end{aligned}
$$

where $\sinh_{y\theta^*}(x) = (e^{yx\theta^*} - e^{-yx\theta^*})/2$. Hence, we just need to show for all $y > 0$,

$$
\begin{aligned}
& w_1 w_2 \left( (1 - b) \sinh_{y\theta^*}(2b + 1) + (1 + b) \sinh_{y\theta^*}(2b - 1) \right) \\
& \quad + \left( (2w_1 - 1)(2w_1^* - 1) - (1 - 2w_1 w_2) b \right) \sinh_{y\theta^*}(1) \;>\; 0, \quad \forall b \in (0, \gamma], w_1 \in (w_1^*, 1).
\end{aligned}
$$

By Taylor expansion of $\sinh_{y\theta^*}(x)$, we just need to show for all $k \geq 0$, we have

$$
\begin{aligned}
& w_1 w_2 \left( (1 - b)(2b + 1)^{2k+1} + (1 + b)(2b - 1)^{2k+1} \right) \\
& \quad + \left( (2w_1 - 1)(2w_1^* - 1) - (1 - 2w_1 w_2) b \right) \;\geq\; 0, \quad \forall b \in (0, \gamma], w_1 \in (w_1^*, 1). \quad (79)
\end{aligned}
$$

where inequality is strict for $k \geq 2$. It is straight forward to check (79) holds for $k = 0$ due to $b \leq \gamma$. For $k \geq 1$, note that

$$
\begin{aligned}
& (1 - b)(2b + 1)^{2k+1} + (1 + b)(2b - 1)^{2k+1} \\
&= \left( (2b + 1)^{2k+1} + (2b - 1)^{2k+1} \right) - b \left( (2b + 1)^{2k+1} - (2b - 1)^{2k+1} \right) \\
&= 4b \sum_{i=0}^{2k} (-1)^i (2b + 1)^{2k-i}(2b - 1)^i - 2b \sum_{i=0}^{2k} (2b + 1)^{2k-i}(2b - 1)^i \\
&= 2b \left( \sum_{i=0}^{k-1} (2b + 1)^{2k-2i-1}(2b - 1)^{2i}(2b + 1 - 3(2b - 1)) + (2b - 1)^{2k} \right) \\
&= 2b + 2b \left( \sum_{i=0}^{k-1} (2b + 1)^{2k-2i-1}(2b - 1)^{2i}(4 - 4b) + (2b - 1)^{2k} - 1 \right) \\
&= 2b + 2b(4 - 4b) \left( \sum_{i=0}^{k-1} (2b + 1)^{2k-2i-1}(2b - 1)^{2i} - \sum_{i=0}^{k-1} (2b - 1)^{2i} b \right) \\
&\geq 2b + 2b(4 - 4b) \left( \sum_{i=0}^{k-1} (b + 1)(2b - 1)^{2i} \right) \\
&\geq 2b.
\end{aligned}
$$

where last two inequalities hold due to $b \leq \gamma < 1$ and last inequality is strict when $k \geq 2$. Hence, to show (79), we just need to show

$$2bw_1w_2 + (2w_1 - 1)(2w_1^* - 1) - (1 - 2w_1w_2) \, b \; \geq \; 0$$
$$\Leftrightarrow \quad b \; \leq \; \gamma,$$

which holds clearly. Hence, this completes the proof for this lemma.

## F  Additional numerical results

| Sample size | Separation | $w_1^* = 0.52$ | $w_1^* = 0.7$ | $w_1^* = 0.9$ |
|---|---|---|---|---|
| | $\theta_2^* = 1$ | 0.999 / 0.999 | 0.499 / 0.699 | 0.450 / 0.338 |
| $n = 1000$ | $\theta_2^* = 2$ | 0.799 / 0.500 | 0.497 / 0.800 | 0.499 / 0.899 |
| | $\theta_2^* = 4$ | 1.000 / 1.000 | 0.447 / 0.900 | 0.501 / 0.999 |
| | $\theta_2^* = 1$ | 0.497 / 1.000 | 0.493 / 1.000 | 0.501 / 0.000 |
| $n = \infty$ | $\theta_2^* = 2$ | 0.504 / 1.000 | 0.514 / 1.000 | 0.506 / 1.000 |
| | $\theta_2^* = 4$ | 0.495 / 1.000 | 0.490 / 1.000 | 0.514 / 1.000 |

Table 2: In this table, we consider mixture of two Gaussian in one dimension with $\theta_1^* = 0$. We present the probability of success $P_1$ and $P_2$ for EM to find the MLE for Model 1 and Model 2, respectively, reported as $P_1$ / $P_2$. We only keep the first 3 digits after the decimal for each probability.

| Sample size | Separation | $w_1^* = 0.52$ | $w_1^* = 0.7$ | $w_1^* = 0.9$ |
|---|---|---|---|---|
| | $\theta_2^* = 1$ | 0.999 | 0.999 | 0.800 |
| $n = 1000$ | $\theta_2^* = 2$ | 1.000 | 1.000 | 1.000 |
| | $\theta_2^* = 4$ | 1.000 | 1.000 | 1.000 |
| | $\theta_2^* = 1$ | 1.000 | 1.000 | 1.000 |
| $n = \infty$ | $\theta_2^* = 2$ | 1.000 | 1.000 | 1.000 |
| | $\theta_2^* = 4$ | 1.000 | 1.000 | 1.000 |

| Case 1 | Case 2 | Case 3 | Case 4 |
|---|---|---|---|
| 0.980 | 0.998 | 1.000 | 1.000 |

Table 3: We present the probabilities of success $P_3$ for EM to find the MLE for Model 1 under the new procedure described in Section 3.3. The first table is for mixture of two Gaussians in one dimension discussed in Section 3.2. The second table is for mixture of three or four Gaussians discussed in Section 3.3. We only keep the first 3 digits after the decimal for each probability.