[Reviews · NeurIPS 2018]

Reviewer 1



The paper studies the convergence of EM algorithms for learning mixture of Gaussians. ------------- Results and comparison to existing work. 1. Thm 1: Shows that for learning mixture of two Gaussians with well separated means at -\theta^* and +\theta^* (and covariance of identity), the EM algorithm on the population objective for learning mixture weights and mean parameters converge to the true parameters (\theta^*,w^*) or almost all initializations. --The above result generalizes part of the results the existing work of Xu et al. 2016 and Daskalakis et al. 2016 which analyze EM under the above conditions when the mixture weights are fixed to be 1/2. --On the other hand, the results in Xu et al. 2016 extend to general means (\theta^*_1,\theta^*_2) and not necessarily (\theta^*,-\theta^*). From a preliminary check, I could not extend the current proof to general means. --Daskalakis et al. 2016 also give more rigorous analysis with finite samples. 2. Thm 2: for non-uniform mixtures, when the weights are fixed to true parameters, a constant fraction of initializations will converge to a wrong solution. --This negative example essentially relies on picking a theta^(0) that correlates better with the smaller component (say C2=(-theta^*,w2)). If the weights are learnable, the model will simply change the ordering of learned components, while effectively learning the same model. E.g. learn C_1=(-theta^*,w_2) and C_2=(theta^*,w_1) instead of C_1=(theta^*,w_1) and C_2=(theta^*,w_2) --The example highlights interesting phenomenon on the optimization landscape wherein, in the presence of invariances, fixing a subset of parameters to its optimum value restricts the set of invariances the model can learn. E.g., in this problem, for almost all initializations, there is a “good” basin of attraction in the joint parameter space of (w,\theta) that globally optimizes optimization objective by learning the first component to be either C1=(theta^*,w_1^*) or C1=(-theta^*,w_2^*). However, with w_1^* fixed, a constant the fraction of initializations will have a non-global basin of attraction in the space of just \theta. --However, for the problem of learning MOG itself, this result is not that interesting since such bad initializations can be easily avoided by using some very reasonable initialization schemes such as initializing with overall the average of samples or choosing an initiation uniformly randomly but with norm close to 0. ---------------------- Summary of evaluation: --While Thm 1 is a non-trivial extension of Xu et al. 2016 and Daskalakis et al. 2016 for the case of symmetric means, the significance of the result is somewhat limited since the analysis does not seem to naturally extend to general means (\theta^*_1,\theta^*_2) as in Xu et al. 2016 or to finite samples as in Daskalakis et al. 2016. --Thm 2 in my opinion is more interesting for the insight into the optimization landscape the optimization parameters have invariances. I would suggest elaborating on the optimization landscape more in the paper --Finally, the mixture of two gaussians is a very special case where EM converges since the landscape does not have bad local optima. The paper misses discussions on the following relevant results: (a) Jin, Chi, et al. "Local maxima in the likelihood of gaussian mixture models: Structural results and algorithmic consequences." — show that local optima are common in GMMs with more than two components (b) Yan et al. “Convergence of Gradient EM on Multi-component Mixture of Gaussians” (2017)—show local convergence of EM for >2 component GMMs Overall there are some interesting results and the theorems are correct from my limited check, but I find the extensions somewhat incremental compared to existing work. ----- Read author response

Reviewer 2



Summary ------------- In this paper, the authors focus on the convergence of the celebrated EM (Expectation - Maximization) algorithm for mixtures of two Gaussians with unknown means. Departing from the recent results of Xu et. al. and Daskalakis et. al. the authors consider the case of unbalanced mixture, where the probability of each component is different. Let w_1 and w_2 = 1 - w_1 be the probabilities of the component 1 and 2 respectively. In has been observed before that when w_1, w_2 are known then the EM iteration has a spurious stable fixed point and hence does not globally converge to the true parameters of the model. The authors make this observation formal in their Theorem 2. The main result of this paper is to show that if instead we run the EM iteration with unknown w_1 and w_2 then the EM algorithm globally converges to the correct estimation of the parameters of the model. Theorem 1 of this paper generalizes the Theorem 1 of Xu et. al. and Theorem 2 of Daskalakis et. al. in the sense that it accommodates unbalanced mixture of Gaussians. It is not clear though from the statement of Theorem 1 whether it implies geometric convergence or not, as it was the case for the corresponding theorems in Xu et. al. and Daskalakis et. al.. Moreover Lemma 2 in Section 2.3 implies that the convergence is guaranteed from the existence of a potential function m(.) without any information on how fast convergence this potential function implies. The technique used to prove Theorem 1 has two parts (1) reduction from the multi-dimensional to the single-dimensional case, (2) proof of convergence in the single-dimensional case. Part (1) of the proof is almost the same as the one used in Xu et. al. for the balanced mixture, although it is interesting that it generalizes in this case. Part (2) of the proof is novel and more interesting. The authors succeed to carefully constructing a global potential function based on the local properties of the EM iteration in each region of tha space. Finally, the results of this paper give one of the few formally proved examples where the over-parameterization is indeed helpful. The fact that over-parameterization helps in machine learning tasks has been observed in practice in the recent year without a solid theoretical understanding of why. Summary of Recommendation --------------------------- I believe that this paper is a novel and important contribution in the direction of understanding the global convergence properties of non-convex methods in machine learning and hence I strongly recommend acceptance if the following comments are addressed. Comments to the authors --------------------------- 1. Does your proof of Theorem 1 imply geometric convergence? If not please rephrase Remark 1 since in this case Theorem 1 of Xu et. al. is a stronger statement. 2. What about the sample dependence of EM? Is it possible to generalize Theorem 6 of Xu et. al. or Theorem 3 of Daskalakis et. al.? If not, an explanation of the bottleneck towards such generalization would be very informative. 3. In the statement of Lemma 2 it is claimed that since m(.) is a continuous function with a single root m(\theta, w) = 0, and m(\theta^t, w^t) -> 0 this implies (\theta^t, w^t) -> (\theta, w). Im afraid that this is not correct, consider for example the single variable function f(x) = x*exp(-x). This function is a continuous function with a single root f(0) = 0, but the sequence x^t = t has f(x^t) -> 0 although definitely x^t does not converge to 0. Looking at the proof of Lemma 2 though it seems like this is not an important issue since the authors show something stronger, namely that (\theta^(t + 1), w^(t + 1)) is always strict inside a rectangle D that depends on (\theta^(t), w^(t)) and contains the global fixed point. I didn't have time to very carefully verify the proof of this lemma so a comment from the authors on this issue will be appreciated. 4. Continuing to comment 3. it seems like the construction of m(.) with the area of the rectangles could be transformed to a contraction map argument instead of a potential function argument. If this is possible then the statement becomes much clearer with respect to the convergence rate of EM.

Reviewer 3



The paper presents a convergence result for the Gaussian mixture model. In particular, for GMM with two components, each with an identity covariance matrix, the paper shows that the EM algorithm will converge to the optimal solution. It further demonstrates that parameterizing the mixture weights is helpful. Estimating mixture weights as unknown parameters in the EM prevents the algorithm from being trapped in certain local maximas. Comparing to existing results for the convergence of GMM: - The result extends that of Xu et al. and Daskalakis et al. in that the two mixture weights don't need to be equal. It is weaker than these results in the sense that only asymptotic convergence is guaranteed. - The result depends on a strong assumption that the covariance matrices must be the identity matrix. - With more than two mixture components, it is known that the EM algorithm is not guaranteed to converge to the optimal solution [1]. Therefore the conclusion of this paper doesn't generalize to more than two components. Overall, the paper presents an interesting result, but it is an incremental improvement over some already restrictive theoretical analyses. I am not sure if the result will make a broader impact to the NIPS community. [1] Local maxima in the likelihood of gaussian mixture models: Structural results and algorithmic consequences, C Jin, Y Zhang, S Balakrishnan, MJ Wainwright, MI Jordan